# A machine learning-based perspective on deep convective clouds and their organisation in 3D. Part I: Influence of deep convective cores on the cloud life-cycle

Sarah Brüning[1] and Holger Tost[1]

[1]Institute for Physics of the Atmosphere, Johannes Gutenberg University Mainz, Johann-Joachim-Becher-Weg 21, Mainz, 55128, Rhineland-Palatinate, Germany

**Correspondence:** Sarah Brüning (sbruenin@uni-mainz.de)

**Abstract.** In this two-part study, we examine spatio-temporal patterns of convective clouds, their properties, and organisation. We use a machine learning-based method to extrapolate a contiguous 3D cloud field of 2D satellite data. The predicted data are used to simultaneously track both the horizontal and vertical development of clouds. Our research focuses on West Africa, a region known for frequent convective events and severe weather. In Part 1, this study compares cloud and core properties and the cloud life-cycle over land and ocean during a six-month period from March to August 2019. Our analysis reveals that 65 % of tracked cloud systems contain only a single core and persist for less than three hours. Despite their shorter lifespan compared to multi-core clusters, single-core clouds exhibit stronger changes in the radar reflectivity and a higher vertical growth. In contrast, multi-core clouds show greater horizontal growth, encompassing larger cloud and core areas, higher cloud-top heights (CTH), and higher average reflectivity at 10 km altitude. We also find that, in systems with more cores, both the maximum number of cores and the peak core area occur later during the cloud life-cycle. Notably, the differences in cloud characteristics between land and ocean are smaller than those associated with the number of convective cores. However, the results may not fully capture climatological differences. Further research using longer time series is needed to quantify the observed variability of tropical convection.

## 1 Introduction

Convective clouds play a vital role in the hydrological cycle of the Earth through their radiative forcing and feedback mechanisms (Wielicki et al., 1995). Despite growing evidence for the connection between clouds and climate warming, they remain one of the greatest sources of uncertainty in climate sensitivity assessments (e.g., Bony et al. (2015); Sherwood et al. (2020)). Additionally, convective clouds are key drivers of severe weather, particularly large-scale systems like mesoscale convective systems (MCSs), which are linked to extreme events such as hailstorms, damaging winds, and intense rainfall (e.g., Houze

and Hobbs (1982); Leary and Houze (1980); Maddox (1980)). Because of their societal and environmental impacts, accurately representing convective clouds remains of particular interest.

MCSs are typically defined as convective storm complexes with an axis length of at least 100 km (Houze Jr., 2004). These systems often feature a broad cold cloud shield, one or more deep convective cores (hereafter, "cores"), strong vertical updrafts that connect these cores at higher altitudes, and widespread anvils extending from the convective region (Zipser and LeMone, 1980). While cores drive intense precipitation, the stratiform anvil and cirrus canopy generally produce lighter rain (e.g., Houze Jr. (1989); Hartmann et al. (1984)). Core sizes typically range from 10 to 100 km, with lifespans of 1–3 hours, whereas anvils can persist for up to 10–20 hours. The idealised MCS life-cycle includes three stages: development, maturity, and dissipation (Futyan and Genio, 2007). During development, deep convective cells form and transport condensate upward. In the maturity stage, the anvil and associated mesoscale circulation develop while convection continues. In the dissipation stage, deep convection ceases, and the system gradually fades (e.g., Houze and Hobbs (1982); Machado et al. (1998)). The MCS life-cycle is influenced by location, time of day, and surface type. For instance, small to medium MCSs commonly form over land in the afternoon due to local thermal instability and, potentially, sea breeze circulations (Chen and Houze, 1997). In contrast, oceanic MCSs experience weaker diurnal variability because of the surface's stable thermal properties (Nesbitt and Zipser, 2003).

Our current understanding of convective clouds largely stems from satellite observations (Haynes et al., 2009). We may identify convective clouds from satellite data by distinguishing core regions and the surrounding cloud field (Steiner et al., 1995). When derived from passive satellite observations, cores are typically characterised by cold peaks in brightness temperature, surrounded by warmer anvil regions. Morphological features such as aspect ratio, length, width, and area may further classify convective systems (Ganetis et al., 2018). For instance, passive and active sensors provide valuable insights into the temporal evolution of clouds. Passive sensors, especially those measuring infrared (IR) radiation, help identify cloud-top features (Mecikalski et al., 2010). However, they lack vertical resolution, making it difficult to distinguish between deep convection, stratiform clouds, and cirrus (Liu and Zipser, 2008). In contrast, active sensors like radar can resolve vertical cloud structures and hydrometeor distributions (e.g., Bacmeister and Stephens (2011); Oreopoulos et al. (2017)). Still, both sensor types offer only limited spatial or temporal coverage. As Masunaga and Luo (2016) point out, a global, continuous 3D view of convective clouds remains unavailable from current satellite missions.

Early studies relied on manual tracking, but automated detection algorithms now enable the processing of large datasets. Most of these algorithms are centroid-based, linking cloud objects across time steps (Prein et al., 2024). One of the earliest and most influential tools is *TITAN* Dixon and Wiener (1993), later adapted into *TINT* Raut et al. (2021), which is optimized for tracking fast-evolving storm cells. *TOOCAN*, developed by Fiolleau and Roca (2013), focuses specifically on convective cores and associated anvils in MCSs. More recently, general-purpose tools such as *PyFLEXTRKR* (Feng et al., 2023) and *tobac* (Heikenfeld et al., 2019) have emerged. *PyFLEXTRKR* offers flexible 2D tracking, while *tobac* supports 4D analysis, enabling a more comprehensive view of convective systems.

Despite decades of research, knowledge of the 3D structure of convective cores remains limited. In the absence of high-resolution, global 3D observational data, our understanding of the relationship between cores and overall cloud evolution relies

heavily on 2D observations and simulations. Active and passive sensors contain important vertical or horizontal information, but are limited in their spatial and temporal coverage (active) or offer only an approximation of the vertical column (passive) (Masunaga and Luo, 2016; Taylor et al., 2017). To address this gap, we apply a machine learning (ML) framework to reconstruct contiguous 3D radar reflectivity fields from 2D satellite data (Brüning et al., 2024). Our goal is to simultaneously capture the

horizontal and vertical evolution of convective clouds and their cores. We use imagery from the Meteosat-11 SEVIRI sensor as input to the ML model, which is trained to reconstruct vertical cross sections based on CloudSat Cloud Profiling Radar (CPR) observations. This approach allows us to extrapolate a continuous 3D cloud field between 2.4 and 24 km altitude. The resulting dataset combines the spatial and temporal characteristics of the SEVIRI input with the vertical structure from CPR. We then use the *tobac* package to identify convective clouds and track them over time in 15-minute intervals. This enables the analysis

of 3D convective cloud and core properties over both land and ocean. A focus is comparing the life-cycles of clouds with single versus multiple core regions, offering insights into the spatial clustering and organization of convection.

We organise the article as follows. In Sect. 2, we present the data used in this study. Section 3 provides details on details the ML-based 3D reconstruction and tracking methodology. Section 4 presents the results focusing on the temporal variability of convective cloud and core characteristics and the connection between the cores and cloud life-cycle. Section 5 compares our

findings to known tropical convection characteristics and outlines limitations. Finally, Sect. 6 contains the concluding remarks.

## 2    Data

The area of interest (AOI) for this study spans a tropical region over central and western Africa, extending from 30° N to 30° S and 30° W to 30° E. This region is characterised by environmental conditions that contribute to the development of convective clouds (Takahashi et al., 2023). Our objective is to detect and analyse convective clouds and their life-cycles by a six month

period between March and August 2019. This period was selected to highlight key characteristics of 3D cloud structures across different surface types within the AOI. Particular attention is given to the seasonal northward migration of the Inter-Tropical Convergence Zone (ITCZ) and the onset of the West African Monsoon (WAM). Since the WAM plays a critical role in shaping West Africa's climate and is responsible for a significant portion of the annual rainfall in the AOI, its arrival is expected to enhance the frequency of convective cloud formation (Andrews et al., 2024; Kniffka et al., 2019).

To investigate these phenomena, we employ a ML algorithm that generates time series of 3D radar reflectivity fields based on 2D satellite observations, as described in Brüning et al. (2024). The input data are derived from the Spinning Enhanced Visible and Infrared Imager (SEVIRI) onboard the Meteosat-11 (MSG) satellite (Schmetz et al., 2002). The AOI is situated near the nadir of SEVIRI, which is positioned above the Equator at 0° longitude. SEVIRI captures multispectral imagery across 12 channels in the visible, near-infrared, and thermal-infrared ranges. Eleven of these channels offer a temporal resolution of

15 minutes and a spatial resolution of 3 km, while one high-resolution visible channel provides 1 km resolution at nadir (Table 1).

To validate our ML-based predictions, we use vertical cross-sections of radar reflectivity from the 94-GHz Cloud Profiling Radar (CPR) onboard the polar-orbiting CloudSat satellite. This active radar instrument transmits microwave pulses toward

**Table 1.** Overview of Meteosat SEVIRI channels (Schmetz et al., 2002).

| Channel | Wavelength ($\mu m$) | Description | Spatial resolution at nadir | Retrieval at nighttime |
|---------|----------------------|-------------|------------------------------|-------------------------|
| VIS0.6 | 0.56-0.71 | Visible channel | 3 km | No |
| VIS0.8 | 0.74-0.88 | Visible channel | 3 km | No |
| NIR1.6 | 1.5-1.78 | Near infrared window | 3 km | No |
| IR3.9 | 3.48-4.36 | Near infrared window | 3 km | Yes |
| WV6.2 | 5.35-7.15 | Upper-troposphere water vapour | 3 km | Yes |
| WV7.3 | 6.85-7.85 | Lower-troposphere water vapour | 3 km | Yes |
| IR8.7 | 8.30-9.10 | Mid infrared window | 3 km | Yes |
| IR9.7 | 9.38-9.94 | Ozone sensitivity | 3 km | Yes |
| IR10.8 | 9.80-11.80 | Clean longwave window | 3 km | Yes |
| IR12.0 | 11.00-13.00 | Dirty longwave window | 3 km | Yes |
| IR 13.4 | 12.40-14.40 | $CO_2$ sensitivity | 3 km | Yes |
| HRV | 0.5-0.9 | High-resolution visible | 1 km | No |

Earth to detect vertical profiles of cloud hydrometeors. The CPR achieves a vertical resolution of 240 m (distributed across 125 bins) and a horizontal resolution of 1.4 km across-track and 1.8 km along-track (Stephens et al., 2008). For this study, we use the level-2 2B-GEOPROF product. To address signal attenuation at lower altitudes, we limit the vertical analysis to 90 height levels ranging from 2.4 km to 24 km (Sassen and Wang, 2008). Additionally, due to reduced sensor sensitivity at high altitudes, the CPR may underrepresent certain cloud types, particularly thin ice clouds like cirrus. To mitigate noise, we filter the 2B-GEOPROF dataset using the CloudSat cloud mask quality flag (Marchand et al., 2008).

## 3  Method

### 3.1  Machine learning-based reconstruction of a 3D cloud field

In the following section, we briefly outline the method used to reconstruct a 3D cloud field, based on the framework developed by Brüning et al. (2024). Our approach employs a ML algorithm built on a 2D Res-UNet architecture — a modified version of a convolutional neural network specifically designed for image segmentation tasks (Ronneberger et al., 2015). While the model is primarily trained to reconstruct vertical cross-sections of the CloudSat CPR radar reflectivity using imagery from the MSG SEVIRI satellite, its output represents full 3D radar reflectivity volumes rather than just 2D slices.

The reconstructed 3D cloud field spans an area from 60° W to 60° E and from 60° S to 60° N, corresponding to 2400 × 2400 pixels in the horizontal dimensions. SEVIRI satellite imagery serves as input to the Res-UNet model, hence the horizontal resolution of the 3D data is 3 km x 3 km. Initially, 11 channels covering the visible, near-infrared, and thermal-infrared spectra

**Table 2.** Modifications to the Res-UNet applied in this study originally proposed in Brüning et al. (2024)

| Parameter | Original configuration | Modification |
|---|---|---|
| Number of input channels | 11 | 8 |
| Loss function | L2 | L1 |
| Nighttime predictions | No | Yes |
| Average RMSE | 3.05 | 2.99 |

were used (Table 1). For this study, we exclude the visible channels to ensure the model can make predictions independent of daylight conditions (Table 2).

Training data consist of $128 \times 128$ pixel patches of SEVIRI imagery that are spatially and temporally aligned with CloudSat overpasses. Each training sample includes a diagonal CPR cross-section. Due to the spatial resolution mismatch between MSG SEVIRI and CloudSat, we downsample the SEVIRI data to match the CPR's horizontal resolution. To address the strong class

imbalance between cloudy and cloud-free conditions, we limit cloud-free samples to a maximum of 10 % of the training data. The model is trained on nine months of data and validated on a separate three-month period. The Res-UNet is trained to reconstruct CloudSat-like 3D reflectivity volumes with a horizontal size of $100 \times 100$ pixels and a vertical size of 90 levels. The predicted radar reflectivity values range from –25 to 20 dBZ and retain the 15-minute temporal resolution of the original SEVIRI input. We use an L1 loss function (mean absolute error) during training to evaluate the model's performance. Notably,

direct validation is possible only for the diagonal cross-section, which accounts for about 10 % of each training sample. For the three-month test period, the modified daylight-independent model achieves a root mean square error (RMSE) of 2.99 dBZ — an improvement over the original model (Table 2). This level of accuracy is comparable to the 5 dBZ precision reported for other CloudSat products (Tomkins et al., 2024).

To generate complete coverage of the domain between 60° W to 60° E and 60° S to 60° N, the individual 3D output patches

are stitched together, producing a unified output volume of with a size $2400 \times 2400 \times 90$ pixels. This method may help to obtain a consistent spatial coverage, particularly over remote oceanic regions where active sensors are sparse (Prein et al., 2024). Visual inspection confirms the absence of artifacts at tile boundaries, indicating a seamless reconstruction of the 3D cloud field in different parts of the domain. To further assess model performance, we compute cloud top heights (CTH) from the predicted radar reflectivity and compare them to CTH values from the CMSAF CLAAS-V002E1 dataset (Finkensieper

et al., 2020). The comparison reveals that the model captures realistic spatial patterns of CTH in both tropical and mid-latitude regions. However, model accuracy tends to decline with increasing distance from the MSG SEVIRI nadir. Finally, the time-series of 3D radar reflectivity volumes is merged along the temporal axis to generate a 4D cloud field, which is used to detect and track convective clouds. For the purposes of this study, we crop the domain to consist of $1200 \times 1200$ pixels, covering the region between 30° W–30° E and 30° N–30° S — effectively focusing on the area between the Tropic of Cancer and the Tropic

of Capricorn.

## 3.2 Tracking convective clouds in 4D

In this study, we analyse the development and properties of convective clouds by employing the *tobac* package, a modular Python-based package for tracking atmospheric objects in 4D time series (Heikenfeld et al., 2019). In this study, we use the recently released version 1.5 of the software package (Sokolowsky et al., 2024). We merge the predicted 3D radar reflectivity fields along the temporal dimension and feed the 4D time series into the tracking algorithm to create continuous trajectories. The workflow to identify possibly convective trajectories consists of three steps: detecting cloud features by their centroid's position, segmenting the associated cloud field for each centroid, and linking segmented objects through time (Figure 1, a–c). Moreover, we aim to separate cloud clusters that are only connected by a few pixels in the horizontal and vertical dimensions (Oreopoulos et al., 2017). The workflow of this object-based approach is depicted in Fig. 1 and will be explained in the following paragraphs.

The framework, while enabling detailed analysis of convective cores, has limitations. The predicted 3D cloud fields represent model-based approximations rather than direct observations, reflecting patterns learned by the ML model. Additionally, using fixed thresholds in the object-based detection may oversimplify complex structures associated to clouds in the atmosphere. Nonetheless, we may employ the data to enable a large-scale, high-resolution tracking of convective systems over the tropical Atlantic and continental Africa.

### 3.2.1 Identifying cloud features

Radar reflectivity does not directly measure vertical air velocity, but it can serve as a valuable proxy for detecting hydrometeors associated with convective cloud development (Luo et al., 2008). To identify potential cloud structures, we apply a fixed threshold of –15 dBZ to distinguish signals of hydrometeors from background noise in the radar reflectivity data (Marchand et al., 2008). While this threshold is only moderately restrictive — allowing for the inclusion of short-lived or weak features — it is intentionally chosen to capture the full spatio-temporal evolution of convective clouds between development and dissipation stage (Esmaili et al., 2016).

The detection process begins by applying a Gaussian filter with a sigma value of 0.5 to smooth the input data and reduce noise (Kukulies et al., 2021). We then compute the centroid of each potential cloud using a weighted center-of-mass approach. Here, each point's weight is defined by its reflectivity value above the –15 dBZ threshold (Heikenfeld et al., 2019). These centroid positions are each assigned a unique identifier, which is maintained throughout the subsequent tracking and segmentation steps.

Next, we apply a 3D watershed segmentation algorithm to delineate the spatial extent of individual cloud structures associated with each centroid. In this approach, the 3D radar reflectivity field is interpreted as a topographic surface, where higher reflectivity values represent peaks and surrounding areas are segmented like catchment basins divided by ridges (Meyer, 1994). We initialise the algorithm by placing markers at the detected centroids in a binary 3D volume, where all other grid points are set to zero. From each marker, the algorithm expands through the volume, assigning reflectivity-based pixels to the corresponding cloud until the threshold of –15 dBZ is reached. The result is a labeled 3D cloud mask, where each pixel is either zero (indicating no cloud) or an integer label corresponding to a specific cloud object (Fiolleau and Roca, 2013). This mask allows

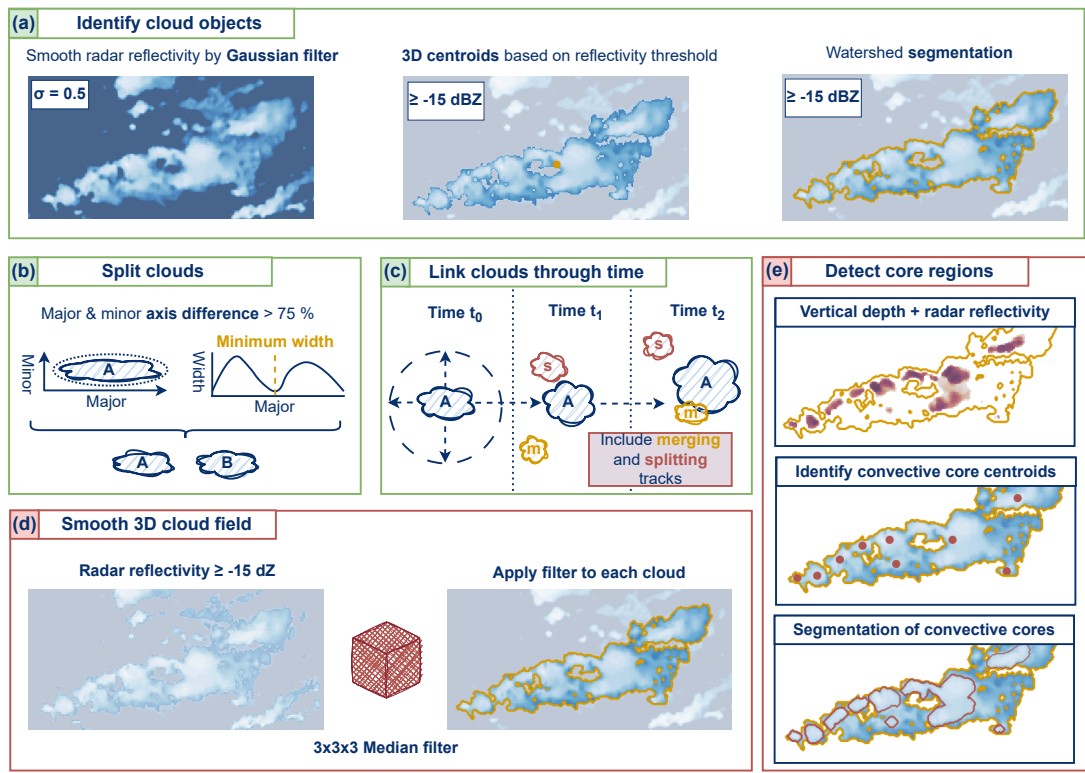

**Figure 1.** Workflow for tracking convective clouds **(a)**–**(c)** and their convective cores **(d)**–**(e)** using an object-based algorithm and ML-based 4D radar reflectivities. The routine consists of **(a)** the identification and segmentation of convective cloud centroids and volumes using a radar reflectivity threshold of -15 dBZ, **(b)** splitting shallow connected objects along their major axis, and **(c)** linking the labelled cloud objects through time. For detecting cores, we apply a Median filter with a kernel size of 3x3x3 pixels to smooth the 3D radar reflectivities associated to each cloud label **(d)**. We aim to identify convective core regions by adding **(e)** the number of pixels higher than 0 dBZ and the average cloud radar reflectivity in each vertical cloud column. This step provides a 2D layer of the combined radar reflectivity and potential core depth, which is used to search for local maxima displaying the centroid locations of convective cores. Subsequently, we apply a watershed segmentation to algorithm to derive the core area.

us to quantify the volumetric structure of clouds: the total number of labeled pixels per centroid corresponds to the cloud's volume rather than its area. For analyses requiring horizontal cloud coverage, we compute the 2D projected area by taking the column-wise maximum across vertical levels.

### 3.2.2 Split shallow connected clouds

After identifying the cloud centroids and their associated areas, we analyse the morphology of each cloud object to determine whether it may represent a merger of multiple cloud systems (Figure 1, b). To do this, we examine the labeled cloud mask to locate local minima in the cloud area, which may indicate potential split points between merged cloud structures. The shape of each cloud is characterised using the best-fitting ellipse (Ganetis et al., 2018). We then calculate the aspect ratio — i.e., the ratio of the major to minor axis lengths. If the major axis is more than 75 % longer than the minor axis, we classify the cloud as elongated (Cui et al., 2021). The orientation of the major axis provides the direction of elongation, which guides the search for potential split locations. Next, we examine the aggregated 2D cloud area along this direction and analyse the area distribution to detect change points. If the distribution is unimodal — featuring a single peak — we perform no split. However, if the distribution is multimodal, we apply a split at the local minimum, provided that this minimum deviates by more than 75 % from the mean size of the cloud shield. Then, we update the segmentation results by assigning a unique label to each newly separated cloud object.

### 3.2.3 Spatio-temporal linking

We track 3D cloud objects over time by linking them based on their estimated movement speed, following the method of Heikenfeld et al. (2019). The 3D perspective allows for detailed analysis of both horizontal and vertical cloud evolution—an essential aspect for understanding core development. At each 15-minute interval, we predict the expected location of a cloud object using its velocity from previous time steps (Figure 1, c). To streamline the linking process, we define a maximum search radius between time steps. Only cloud objects within this radius are considered potential matches, significantly reducing computational effort. When new clouds form, we assign them the average velocity of nearby clouds to estimate their likely movement (Sokolowsky et al., 2024). Due to computational limitations, we apply the linking algorithm to only two consecutive time steps at a time. We assume a successful link is established when a cloud object maintains a 15-minute temporal overlap and shares a consistent identifier across steps. For instance, we compare the cloud areas of linked objects, assuming that genuinely connected trajectories may exhibit more similar area changes than unrelated clouds (Prein et al., 2024). Finally, we evaluate the movement patterns before and after each time step to infer whether cloud objects may be merging or splitting and include the information in the final cloud trajectories.

### 3.3 Detect convective core regions

Convective clouds often contain one or more core regions, which are typically associated with stronger updrafts and intense precipitation that can penetrate above the freezing level. Because the formation and dissipation of these cores are closely linked to severe weather events, analysing their behavior is of particular interest (Takahashi et al., 2017).

To detect convective cores, we use the previously generated labelled 3D cloud mask (Section 3.2.1), derived from the ML–based radar reflectivity data. There are different approaches to identify convective cores from radar reflectivities. These methods may comprise the detection of convective precipitation, which may be associated to core regions in hydrometeors (Haynes et al., 2009; Pilewskie and L'Ecuyer, 2022) or the analysis of the radar reflectivity employing fixed thresholds along the vertical column (Luo et al., 2008; Bacmeister and Stephens, 2011; Igel et al., 2014). In this study, we focus on combining the latter with an object-based detection algorithm to identify centroids of convective cores in the predicted 3D radar reflectivity field. The approach is applied to each labelled cloud for each time step along the cloud trajectory (Figure 1, d,e). We begin by smoothing the radar reflectivity data associated with each cloud label using a 3×3×3 median filter. Core centroids are identified by locating local maxima in a combined metric that incorporates both smoothed radar reflectivity and the vertical extent of a contiguous potential core layer. Specifically, we calculate the mean radar reflectivity for each vertical cloud column, then determine the height of this core layer by counting the number of pixels with reflectivity values higher than 0 dBZ located at more than 5 km height. In our study, we do not include cores occurring lower than cloud base layer, where our model predictions may be less robust. We aim to fill isolated gaps for otherwise vertical continuous cores by expanding the threshold from 0 dBZ to –5 dBZ in columns that contain at least one pixel higher than 0 dBZ (Igel et al., 2014; Luo et al., 2008). We apply a minimum vertical extent of 5 km for a column to be considered part of a core; otherwise, its value is set to zero. The approach is visualised in Figure 1 (e). We add both indicators (average reflectivity and potential core vertical depth) for each pixel associated to a cloud label, resulting in a 2D layer in which we search for local maxima to serve as candidate core centroids. If no local maxima are found - e.g., in case no columns contain pixels higher than 0 dBZ at more than 5 km height - the cloud is recorded as having zero cores for that time step (Feng et al., 2023). When one or more core centroids are identified, we use a 3D watershed segmentation to delineate the core volumes. This process is repeated for every cloud object at each time step throughout its life-cycle, whereas a cloud may contain multiple cores at the same time.

### 3.4 Extract cloud and core properties

For each detected cloud trajectory, we extract both horizontal and vertical characteristics to describe the cloud and its internal structure (Table 3). Cloud properties include cloud area, CTH, CBH, duration, eccentricity, and the ratio of core to total cloud area. The cloud lifetime displays the total lifetime of the trajectory in hours, beginning from the first detection. Cloud area is calculated from the column-wise maximum of the 3D cloud mask, while vertical (CTH, CBH) metrics come from the number of pixels in the vertical column associated to each cloud label. Eccentricity is derived from the best-fitting ellipse, with values closer to 1 indicating a more circular shape (Cui et al., 2021). We also record the cloud's travel distance and assign a surface type using a binary land-sea mask and the modal value for the locations of the cloud trajectory within this land-sea mask.

The ratio of core to cloud area may provide a measure of convective compactness and intensity (Haberlie and Ashley, 2018). For clouds with one or more cores, we calculate the number, mean area, height, lifetime, eccentricity, and average distance between cores. The core area and height are derived from the column-wise maximum horizontal extent and vertical extent of the previously identified cores, similar to the cloud area and CTH. These metrics may help characterise the structural properties of detected cloud systems.

**Table 3.** Features used to describe the properties and life-cycle statistics of detected convective clouds and cores.

| Feature type | Feature name | Definition |
| --- | --- | --- |
| Cloud | Cloud area | Area of the cloud ($km^2$) |
| | Cloud top height (CTH) | Maximum height of the cloud (km) |
| | Cloud base height (CBH) | Minimum height of the cloud (km) |
| | Area ratio | Ratio between cloud area & core area |
| | Eccentricity | Roundness of the best fitting ellipse (cloud) |
| | Reflectivity | Average radar reflectivity of the cloud at 10 km height (dBZ) |
| | Location | Longitude and latitude of the cloud centroid (°) |
| | Travel distance | Euclidean distance for coordinates at initiation and dissipation (°) |
| | Cloud lifetime | Lifetime of the cloud trajectory (h) |
| | Surface type | Modal value from a binary land-sea mask for the cloud trajectory |
| Core | Number of cores | Number of identified convective core regions |
| | Core area | Average area of convective cores ($km^2$) |
| | Core vertical depth | Depth of the core in the vertical column (km) |
| | Mean distance | Average distance between cores in a cloud cluster (km) |
| | Core lifetime | Average lifetime of the cores (h) |
| | Core eccentricity | Roundness of the best fitting ellipse (core) |
| Life-cycle | Reflectivity gradient | Reflectivity change rate at 10 km height (dBZ) |
| | Area growth | Relative cloud area expansion (%) |
| | Vertical growth | Vertical growth of the cloud (km) |

## 3.5    Filter convective cloud trajectories


We filter the cloud trajectories to exclude possibly non-convective tracks from the analysis (Figure 2). For that purpose, we require the cloud tracks to have at least one core and a radar reflectivity of higher than 0 dBZ at 10 km height for at least 15 min along the trajectory. Additionally, we apply a minimum CTH of 10 km and a maximum CBH of lower than 5 km for the cloud during at least one time step (Igel et al., 2014; Luo et al., 2008). While we do not require the convective clouds to have a CTH

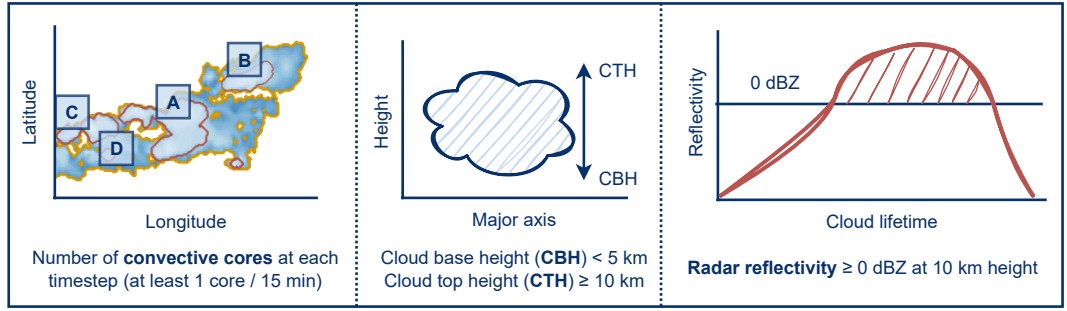

**Figure 2.** Criteria for filtering potentially convective cloud trajectories. The criteria consist of counting the number of convective cores (A–D) for each labelled cloud along the cloud lifetime. Moreover, we check the cloud base height (CBH) and cloud top height (CTH) of the cloud, and the radar reflectivity at 10 km height along the cloud trajectory. Following, we estimate the trajectory to belong to a convective cloud if we detect at least one core, a CBH lower than 5 km, a CTH higher 5 km, and a radar reflectivity higher than 0 dBZ at 10 km height for at least 15 minutes along the cloud lifetime.

higher than 10 km at every time step during their trajectory, we discard the trajectories that never reach the CTH threshold. The criteria may help to identify convective clouds with an evolved cloud base and vertical height that may be typically associated to tropical convection (Li et al., 2021; Takahashi et al., 2023).

## 3.6 Investigating the cloud life-cycle

We analyse the temporal evolution of detected clouds to explore how variations in the cloud life-cycle relate to the number of
convective cores. For this purpose, we divide each cloud's life-cycle into three idealised stages, following the framework proposed by Futyan and Genio (2007). Each stage corresponds to distinct spatio-temporal changes in cloud structure, as simplified illustrated in Figure 3. The first time step of each trajectory marks the beginning of the development stage. Unlike methods that assess cloud stages using a cooling induced by temperature changes, the ML-based radar reflectivity does not provide information on temperatures. As an alternative, we approximate the life-cycle using temporal changes in radar reflectivity at
10 km height and the resulting vertical and horizontal cloud characteristics. For estimating the vertical growth of the cloud, we compute the difference between CTH and CBH (i.e., to display the height of the cloud layer) for every point in time. For the horizontal growth of the cloud, we calculate changes of the cloud area derived as proportional differences to the cloud area at the first timestep of detection.

– *Development stage*: Building on the approach by Luo et al. (2008), we use a radar reflectivity threshold of 0 dBZ at 10
km altitude as a proxy for potential cloud-top cooling, which may be indicative of convective growth. We calculate the temporal gradient of radar reflectivity at 10 km for each cloud trajectory, identifying the time of maximum increase to mark the cloud development stage. This stage may be associated with a high cloud vertical layer and strong updrafts

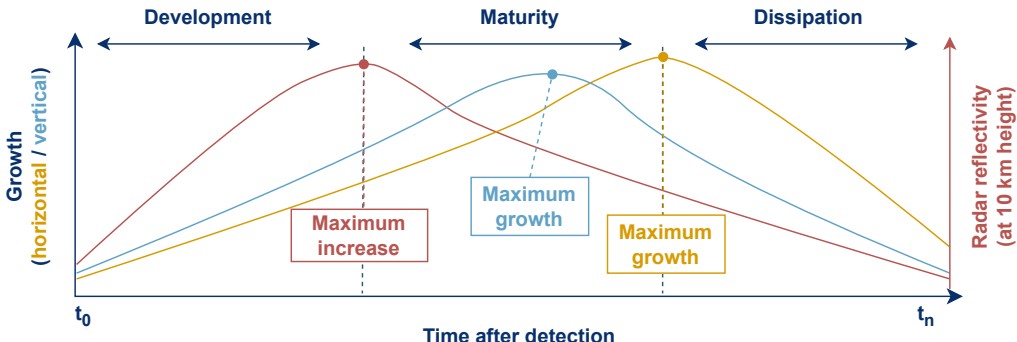

**Figure 3.** Schematic visualisation of the three stages (development, maturity, and dissipation) of the idealised convective life-cycle. Here, we show how changes of the radar reflectivity at 10 km height, the horizontal growth of the cloud area, and the vertical growth of the cloud vertical layer may be connected to the transition between life-cycle stages.

that support continued vertical growth (Kikuchi and Suzuki, 2019; Chen et al., 2021).The transition from development to maturity is defined by the time of maximum radar reflectivity increase (Takahashi et al., 2023; Hu et al., 2021).

– *Maturity stage*: Following the time of maximum radar reflectivity, the reflectivity gradient at 10 km height may gradually decrease. Instead, both the vertical thickness and horizontal extent of the cloud typically may increase in the maturity stage, indicating a sustained growth of the cloud (Gupta et al., 2024).

– *Dissipation stage*: Dissipation begins when vertical growth slows and the cloud reaches its maximum horizontal size. We continue tracking the cloud until the reflectivity falls below the -15 dBZ threshold and no centroids are identified
during feature detection, indicating cloud decay (Crook et al., 2019).

For each trajectory, we determine key time points that may approximate changes in the cloud life-cycle: the moment of maximum reflectivity gradient at 10 km, peak area (horizontal) growth, maximum vertical extent, and the onset of dissipation (Table 3). We also record when the highest number of cores is detected and when cores reach their maximum area. These markers are used to compare life-cycle characteristics between clouds with a single core and convective systems containing
multiple cores (i.e., more than one core). We note that the life-cycle statistics derived for this study are based on the ML-based radar reflectivities and inherit the uncertainties connected to these predictions. Hence, they may only provide an approximation of distinct changes occurring within the cloud trajectories over time.

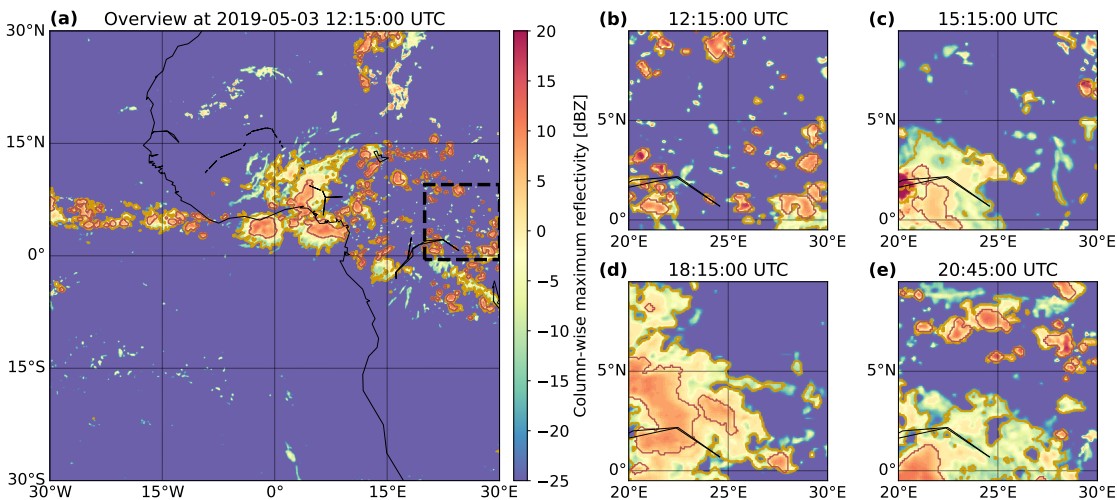

**Figure 4.** An example of convective clouds (orange outline) and their cores (red outline) detected in the ML-based 3D cloud field on the 03.05.2019, 12:15:00 UTC. The cloud mask is plotted over the 3D radar reflectivity, which shows the aggregated column-wise maximum. All times are given in UTC. In **(a)**, we see an overview of the AOI, **(b)** to **(e)** show a zoomed perspective in 3 h intervals (black square).

## 4 Results

### 4.1 Distribution of convective cloud and core properties

Between March and August 2019, we detected approximately 375,000 convective clouds using ML-based 3D radar reflectivity data. After excluding tracks with a lifetime of only one timestep, 338,142 cloud trajectories remained for analysis. Figure 4 shows an example from May 3, 2019, at 12:15 UTC, highlighting tracked convective clouds and cores. While regions over Morocco and Mauritania showed radar reflectivity higher than 0 dBZ, no vertically continuous convective systems were identified there. Instead, numerous convective clouds appeared over the Gulf of Guinea, the equatorial rainforest, and the

Atlantic Ocean. Figures 4 (b)–(e) illustrate the development and dissipation of cores, often lasting only a short time. Some clusters contained multiple cores, potentially indicating mesoscale convective systems (MCSs) (Takahashi et al., 2017).

Our 3D framework allows us to simultaneously track horizontal and vertical cloud development. For core statistics, we separate the core region from the anvil cloud, as shown in Figure 5. Clouds are grouped by core count to distinguish potentially more isolated systems (one core) from clustered systems (multiple cores). For statistical purposes, clouds with 6–9 cores and

those with 10 or more cores are combined into respective categories (Jones et al., 2024).

Single-core clouds make up roughly 65 % of all trajectories (Figure 6, a), with the frequency decreasing as core count increases. Only about 5 % of clouds have 10 or more cores. Most clouds (80 %) have lifespans between 0–6 hours (Figure 6, b). Surface type distribution reveals that 65 % of clouds form over the ocean and 35 % over land—about a 10 % shift toward ocean compared to land-sea coverage (Figure 6, c). Among single-core clouds, 70 % occur over the ocean, while for multi-

core clouds, the figure is 75 %. This imbalance — 249,484 oceanic clouds vs. 88,658 continental — may reflect differences

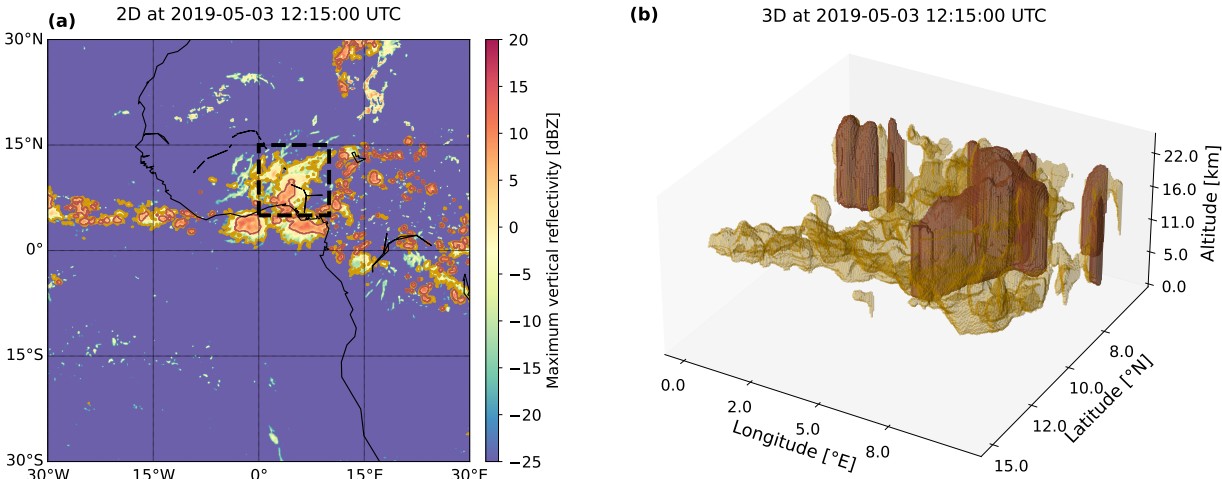

**Figure 5.** An example of the convective clouds (orange outline) and their cores (red outline) detected in the predicted 3D cloud field on the 03.05.2019, 12:15:00 UTC. The cloud mask is plotted over the 3D radar reflectivity, which shows the aggregated column-wise maximum. In **(a)**, we see an overview of the AOI, **(b)** shows the zoomed perspective (black square) in 3D for the cloud volume (orange) and core volume (red).

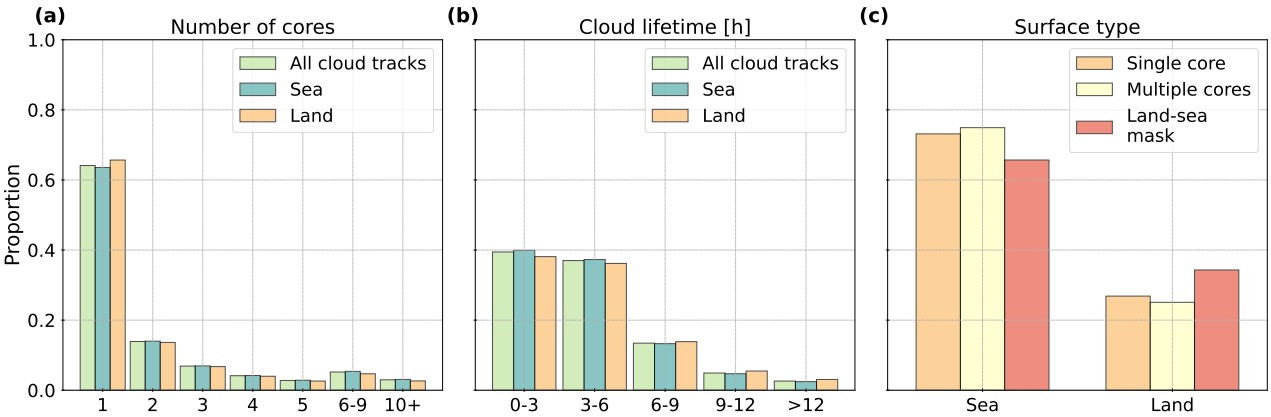

**Figure 6.** Distribution of **(a)** the number of associated cores, **(b)** the average cloud lifetime, and **(c)** the surface type derived from a land-sea mask compared to the modal locations of detected clouds with a single core or multiple cores. We show the distribution in **(a)** and **(b)** for all cloud tracks (n = 338,142), clouds over the ocean (n = 249,484), and clouds over land (n = 88,658).

in tropical landmass distribution and the eastward propagation of convective systems. Oceans may also offer more favorable conditions for multi-core development (Cui et al., 2021).

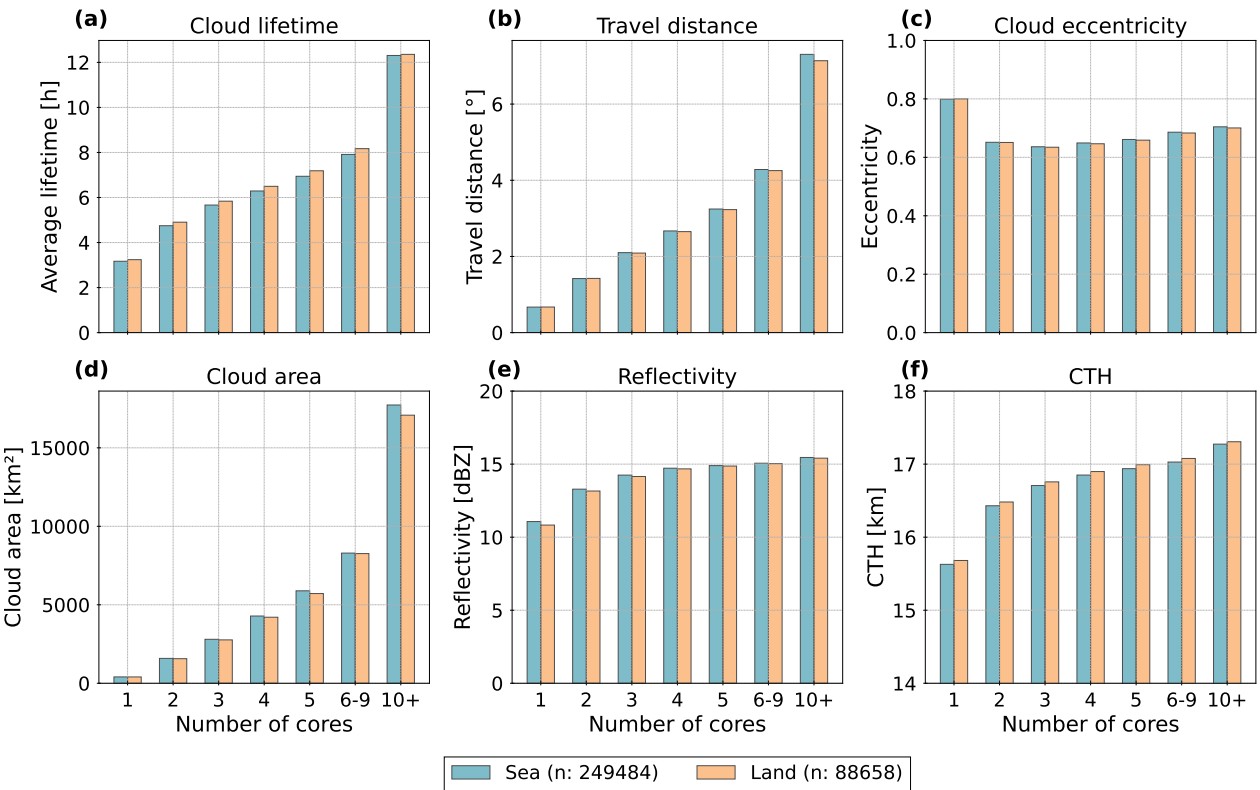

**Figure 7.** Distribution of cloud statistics grouped by the number of associated cores for **(a)** the cloud lifetime, **(b)** the travel distance between first and last detection, **(c)** the cloud eccentricity, **(d)** the cloud area, **(e)** the radar reflectivity at 10 km height, and **(f)** the CTH.

We assess how the 3D cloud properties described in Table 3 may vary with core count and surface type. Our findings show that single-core clouds have shorter lifetimes and travel distances than multi-core systems (Figure 7, a–b). Eccentricity exhibits a weak variation across all groups, mostly ranging between 0.6–0.7 (Figure 7, c). Cloud area increases significantly with core count, especially for clouds with 10 and more cores (Figure 7, d). CTH is 10–20 % greater over land, yet radar reflectivity at 10 km height and cloud area are slightly higher over the ocean (Figure 7, d–f). CTH increases from 15.5 km for single-core clouds to 17.25 km for multi-core ones (Figure 7, f). Land–sea differences are more pronounced for single-core clouds. Despite expectations based on previous tropical studies (Deng et al., 2016; Takahashi et al., 2017), oceanic clouds often show stronger reflectivity and larger areas — though overall surface-related differences remain small. The lower number of land-based clouds may exaggerate statistical noise.

The analysis of core properties (Figure 8) shows average core lifetimes of 0.3–0.4 hours for single-core clouds, increasing to about 0.8 hours for clouds with more than 10 cores (Figure 8, a). Core eccentricity shows little variability and ranges from 0.5–0.6 (Figure 8, b). Core area is slightly larger for single-core clouds than for those with 2–9 cores but increases considerably

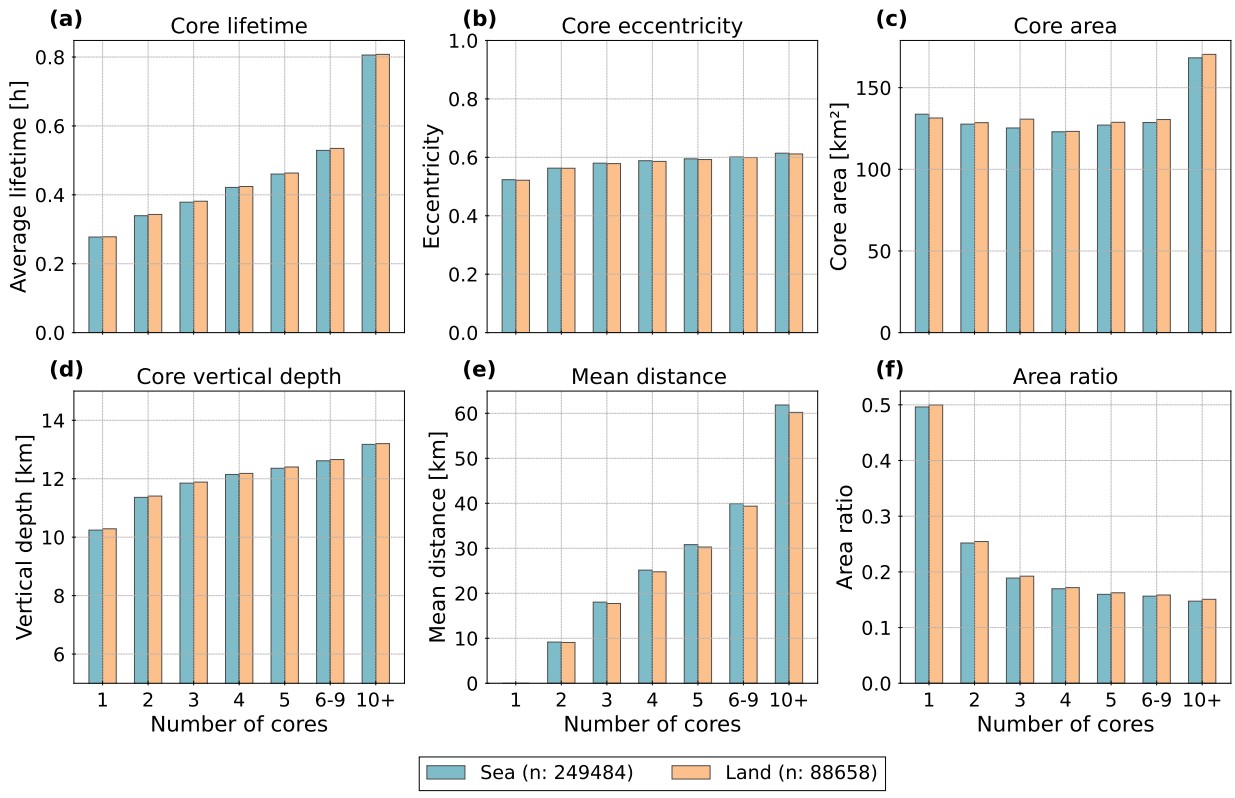

**Figure 8.** Distribution of core statistics grouped by the number of associated cores for **(a)** the core lifetime, **(b)** the core eccentricity, **(c)** the core area, **(d)** the vertical depth of the core, **(e)** the mean distance between individual cores, and **(f)** the area ratio between the cloud and the core.

for clouds with 10 and more cores. For single-core clouds, we detect a larger core area over the ocean, while cores for multi-core clouds are larger over land (Figure 8, c). Core height and distance between cores both increase with core count (Figure 8, d–e). The largest distances, which may indicate the least compact core morphology, occur for clouds with 10 and more cores (Figure 8, e). The area ratio between clouds and cores is highest for single-core clouds and declines with more cores. The sharp decrease may be connected to a faster increase of the cloud area compared to core area for multi-core clouds (Figure 7, d, Figure 8, f).

In summary, clouds with more cores exhibit longer lifetimes, larger areas, greater heights, and increased core size and distances between cores. Core properties are broadly consistent across surface types, except for core area and core distances (Figures 7a, 8c). However, these findings may be influenced by the land–sea imbalance in our dataset and inter-annual variability.

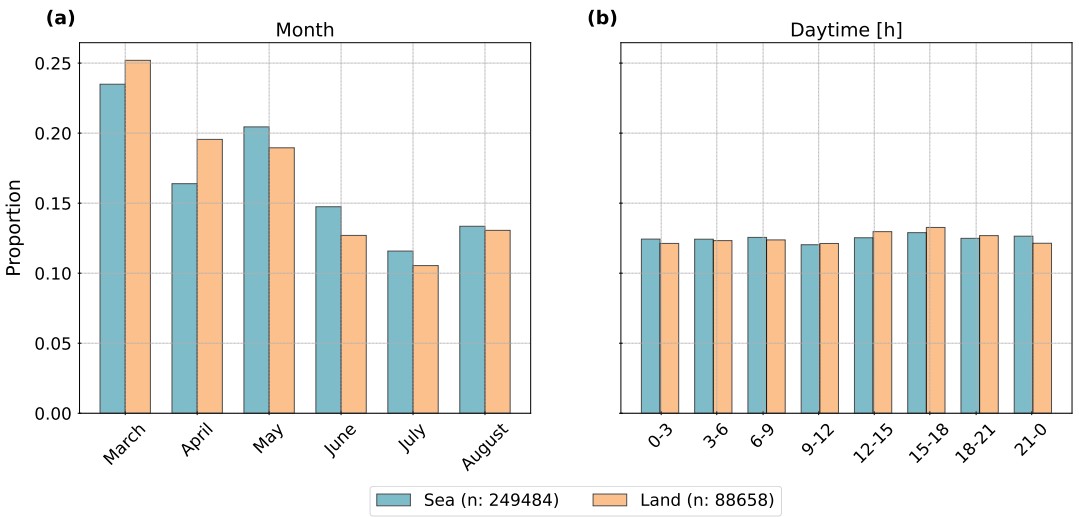

**Figure 9.** Distribution of cloud tracks over land and sea for **(a)** the months between March–August, and **(b)** the daytime of the first detection.

## 4.2 Temporal characteristics of convective clouds

We observe a higher proportion of cloud tracks between March to May, and notably less tracks between June to August. Over land, we find more clouds in March and April. Over the ocean, the proportion of cloud tracks is higher between May and August (Figure 9, a). For the diurnal distribution, we observe slightly more tracks between 12:00 to 18:00 UTC than at other time intervals. During nighttime (21:00–06:00 UTC), more cloud tracks occur over the ocean. In contrast, we detect a higher proportion of cloud occurrences over land during the day. However, the variability connected to the diurnal cycle and surface type remain weak (Figure 9, b).

### 4.2.1 Diurnal cycle over land and sea

We analyse the diurnal cycle of cloud properties for single-core and multi-core clouds over land and ocean by computing a 2D density distribution displaying hourly changes of the cloud properties. Figure 10 illustrates these variations in cloud lifetime (a–d), cloud area (e–h), and radar reflectivity at 10 km height (i–l). Over land, single-core clouds show an afternoon peak (12:00–16:00 UTC) in both radar reflectivity and cloud area, while cloud lifetime displays two peaks: one at night and one in the morning (Figure 10, b, d). Over the ocean, the diurnal cycle is weaker or less distinct. Cloud lifetime lacks a clear diurnal peak (Figure 10, a, c), whereas cloud area and reflectivity show nocturnal and daytime peaks (Figure 10, e, i, k). Despite similar diurnal patterns for the cloud lifetime and radar reflectivity, multi-core clouds consistently exhibit higher mean values than single-core clouds. These differences may reflect environmental contrasts between land and ocean. As suggested by Cui et al. (2021), local circulations over land in the tropics often trigger afternoon convection, producing the observed peaks in

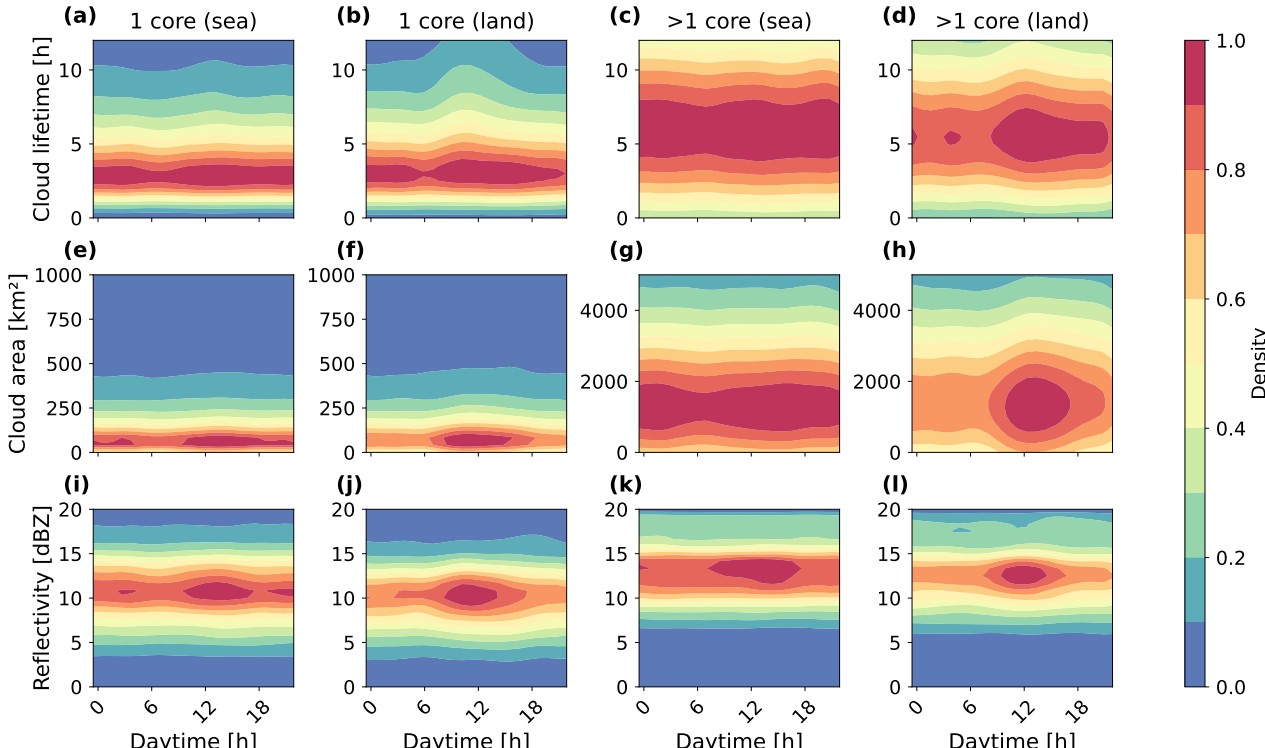

**Figure 10.** Diurnal cycle for cloud properties grouped by the number of associated cores and surface type. We display the hourly changes (in UTC) regarding **(a)**–**(d)** the cloud lifetime, **(e)**–**(h)** the cloud area, and **(i)**–**(l)** the radar reflectivity at 10 km height for single- (1 core) and multi-core (more than one core) clouds over sea and land. The values show the density distribution normalized between 0 and 1.

Figure 10 (f), (h), and (j). In contrast, more constant ocean temperatures may suppress strong diurnal variations (Figure 10, a, c, g).

The diurnal patterns of core properties (Figure 11) largely mirror those of the cloud properties. Over land, core area peaks
between 12:00–18:00 UTC for both single- and multi-core clouds. Over the ocean, single-core clouds show two peaks between 00:00–06:00 and 14:00–20:00 UTC (Figure 11, a–d). The core lifetime follows a similar pattern for single-core clouds. For multi-core clouds, cores over land show two peaks, while oceanic cores point out no clear diurnal variation for multi-core clouds (Figure 11, e–h). For single-core clouds, peaks of the core lifetime resemble the core area (Figure 11, a, e). The distribution of the core height follows those of the core area (Figure 11, m–p). On average, clouds with multiple cores have higher and more
variable values for core area, lifetime, and height. In contrast, the area ratio is lower and has a weaker variability for multi-core systems. For single-core clouds, we observe an afternoon peak over land and nocturnal and afternoon peaks over the ocean. Multi-core clouds show a weak diurnal variation, particularly over the ocean (Figure 11, i–l).

Overall, the diurnal cycle highlights a pronounced afternoon peak over land and a two peak, at nighttime and in the afternoon, over the ocean. These patterns align with observed differences in tropical convective behavior over land and sea (Vondou, 2012).

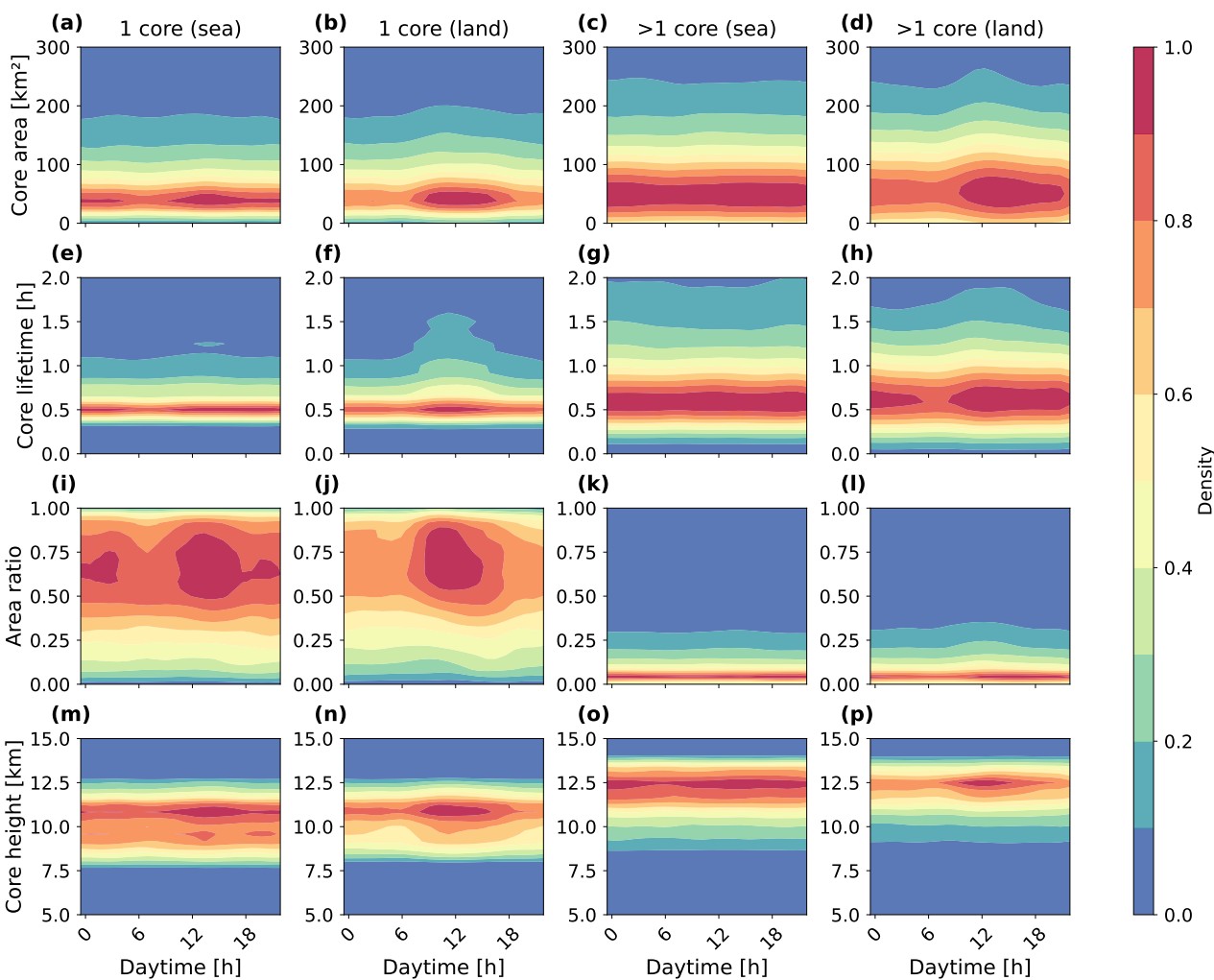

**Figure 11.** Diurnal cycle for core properties grouped by the number of associated cores and surface type. We display the hourly changes (in UTC) regarding **(a)**–**(d)** the core area, **(e)**–**(h)** the core lifetime, **(i)**–**(l)** the core eccentricity, and **(m)**-**(p)** the area ratio between the cloud core and anvil area for single- (1 core) and multi-core (more than one core) clouds over sea and land. The values show the density distribution normalized between 0 and 1.

While we find those patterns mostly for single-core clouds, results for multi-core clouds — especially over the ocean - are less distinct.

### 4.2.2 Monthly variability of convective properties

For different months, the value variability may considerably influence the development of convective clouds and their core structures within the tropics (Andrews et al., 2024). We explore these changes in Figure 12 by comparing monthly averages of

the cloud area, CTH, cloud lifetime, number of cores, core area, and area ratio over land and sea for single-core and multi-core clouds.

From March to August, the cloud area shows a gradual increase for single- and multi-core cloud systems over the ocean. For clouds over land, the cloud area slightly decreases (Figure 12, a). In contrast, CTH generally decreases, though month-to-month fluctuations appear to be higher than a consistent decrease (Figure 12, b). Cloud lifetime shows a higher variability between
months with increases in April for single- and multi-core clouds and in April and June for single-core clouds. Overall, clouds with multiple cores exhibit a slight decline in lifetime over land. Over the ocean, lifetime rises from March to April, decreases in May, and increases again in June — returning to near-initial values by August (Figure 12, c). The number of cores per cloud increases over the ocean from March to July, followed by a sharp drop in August. Initially higher over land, core counts shift in favor of oceanic clouds after April. Convective systems over land show a decrease of core numbers from March to June, a
peak in July, and another decline in August (Figure 12, d). The core area steadily increases over the ocean but fluctuates more over land (Figure 12, e). The area ratio shows a slight decrease for multi-core clouds throughout the period, while remaining higher for single-core clouds. We observe a high monthly variability over land and ocean, whereas the area ratio remains to be higher over continental Africa (Figure 12, f).

To quantify the effect of these changes, we compare average values across two periods: March–May (MAM) and June–
August (JJA). Metrics include the cloud area, CTH, cloud lifetime, number of cores, core area, and area ratio (Table 4). We calculate Cohen's D to measure effect sizes, with thresholds defined as small (< 0.2), medium (0.2–0.5), and large (> 0.8) (Cohen, 2013). Over the ocean, cloud area, number of cores, and core area are higher in JJA, while CTH, cloud lifetime, and area ratio are greater in MAM. A similar pattern emerges over land, except cloud area and number of cores are higher in MAM. Overall, observed differences between the two seasons and over land and sea remain weak. Most effect sizes are small,
indicating high internal variability rather than distinct temporal trends within the period. These results highlight the importance of analysing longer time periods to account for the inherent variability and imbalance between cloud tracks over land and sea (Figures 4 and 9), which may influence the representativeness of the findings.

### 4.3   Impact of convective cores on the cloud life-cycle

#### 4.3.1   Relationship between life-cycle statistics and cloud properties

To analyse the cloud life-cycle (as outlined in Section 3.6), we check the point of time when three key events occur in each cloud trajectory: the maximum radar reflectivity gradient at 10 km altitude ("reflectivity gradient"), the maximum cloud area growth ("area growth"), and the maximum vertical growth ("vertical growth"). Figure 13 shows the distribution of these indicators grouped by the surface type and number of cores. The average maximum reflectivity gradient ranges from 10 to 16 dBZ. Clouds with 2–3 cores show the highest gradients (14.5–16 dBZ), while the gradient for single-core clouds averages around
14 dBZ. It decreases with further increasing core count, dropping to around 10 dBZ for clouds with 10 or more cores. Surface type has little impact overall, though values are slightly higher over the ocean for clouds with 1–3 cores (Figure 13, a). In contrast, cloud area growth is slightly higher over land. More important, clouds with multiple cores grow considerably more

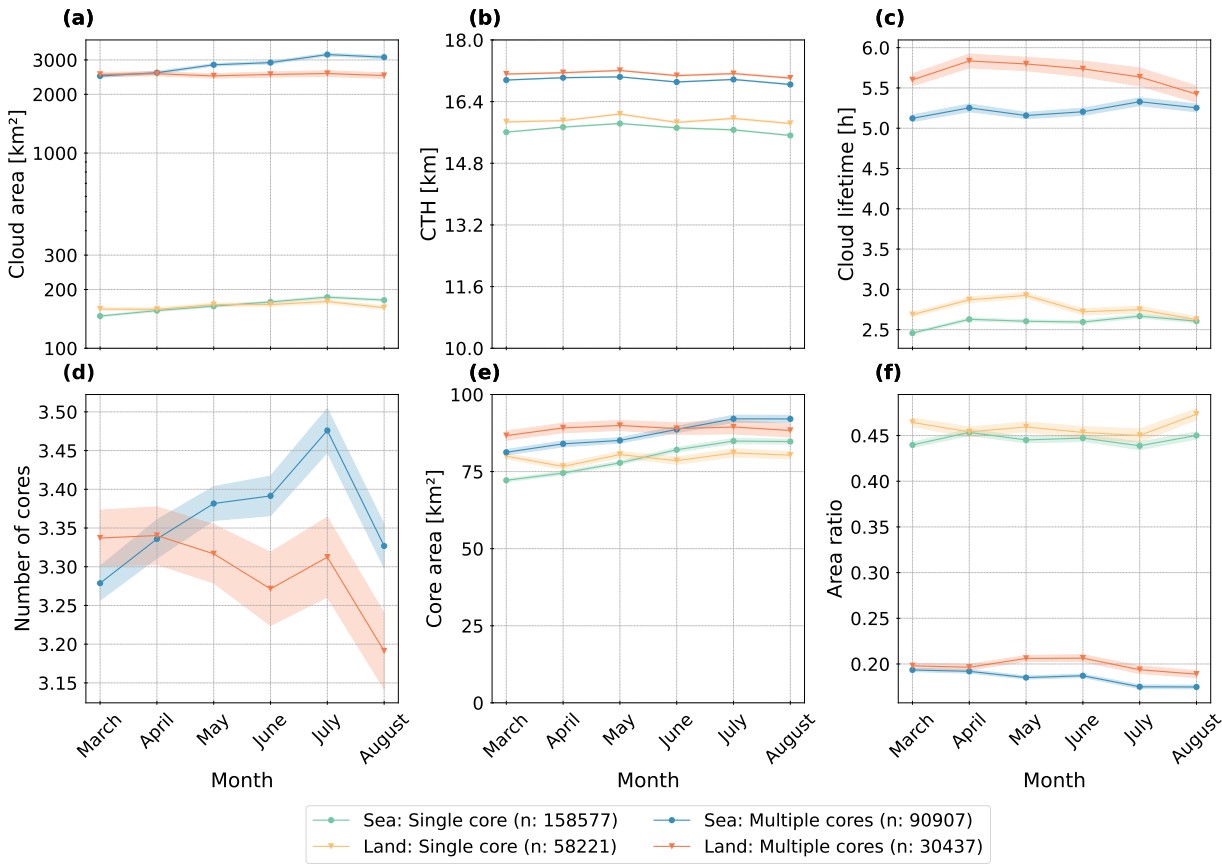

**Figure 12.** Monthly variability of cloud and core statistics between March and August for single-core and multi-core clouds grouped by the surface type for **(a)** the cloud area, **(b)** the CTH, **(c)** the cloud lifetime, **(d)** the number of cores (only multi-core clouds), **(e)** the core area, and **(f)** the area ratio between the cloud and the core. Line plots show the mean value with a confidence interval of 95 %.

in area than single-core clouds - ranging from 22 % (single-core) to 52 % (10 and more cores) (Figure 13, b). For the vertical growth, we observe average values between 5–8 km. Single-core clouds tend to grow higher than multi-core clouds, with only
minor differences between land and sea (Figure 13, c).

We use Spearman's rank correlation coefficient R to examine relationships between life-cycle metrics and cloud/core properties for all cloud tracks, single-core, and multi-core clouds. Overall, cloud and core properties show predominantly positive correlations. The strongest correlation across all datasets appears between CTH and core height. Additional strong correlations include the number of cores with the cloud area and the core height. In contrast, cloud lifetime shows weak to moderate correla-
tions (Figure 14, a). For single-core clouds, the correlation between cloud area and core height weakens, while the link between core area and cloud area strengthens (Figure 14, b). Multi-core clouds exhibit similar patterns to all cloud tracks, though with slightly weaker correlations (Figure 14, c). The number of cores correlates positively with area growth but negatively with both

**Table 4.** Comparison of mean values for cloud (cloud area, CTH, cloud lifetime) and core (number of cores, core area, area ratio) properties between March–May (MAM) and June–August (JJA) grouped by the surface type (Sea, Land). We calculate the effect size measured by Cohen's D to assess the difference between distributions over sea and land.

|  | March–May | | | June–August | | |
|---|---|---|---|---|---|---|
|  | Sea | Land | Cohen's D | Sea | Land | Cohen's D |
| Cloud area | 1605.673 | 1742.597 | 0.017 | 2634.334 | 1681.849 | 0.107 |
| CTH | 16.061 | 16.229 | 0.145 | 15.982 | 16.147 | 0.138 |
| Cloud lifetime | 4.327 | 4.900 | 0.01 | 4.356 | 4.231 | 0.003 |
| Number of cores | 2.128 | 2.166 | 0.009 | 2.459 | 2.062 | 0.136 |
| Core area | 125.118 | 128.249 | 0.013 | 143.647 | 131.638 | 0.035 |
| Area ratio | 0.398 | 0.408 | 0.032 | 0.374 | 0.407 | 0.115 |

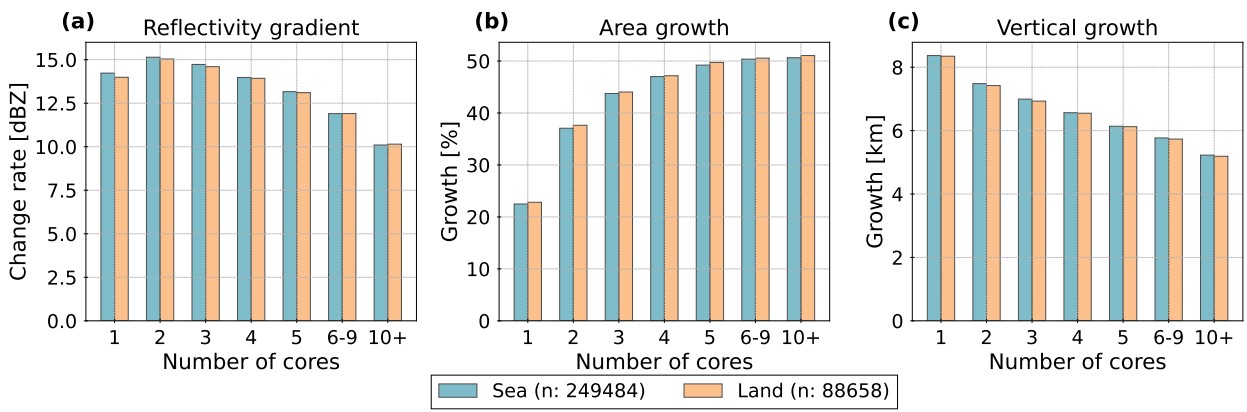

**Figure 13.** Life-cycle statistics grouped by the surface type and the number of associated cores for **(a)** the radar reflectivity gradient at 10 km height, **(b)** the relative area growth, and **(c)** the vertical growth of the cloud.

reflectivity gradient and vertical growth. Reflectivity gradient shows a weak positive correlation with area growth ($< 0.25$) and a stronger one with vertical growth (up to 0.5 for multi-core clouds). Area growth and vertical growth are negatively correlated, with medium to strong correlations. The relationship between the reflectivity gradient and other cloud properties is weak, and its direction differs by cloud type — positive for single-core, negative for multi-core clouds. Area growth correlates positively with cloud and core properties, especially the cloud lifetime. Vertical growth shows a negative correlation with cloud lifetime and mixed, generally weak correlations with cloud area, CTH, and core metrics.

Figure 15 presents the average timing (post-detection) of life-cycle statistics, grouped by surface type and core count. Here, we compute the average time after detection when the clouds reach their maximum for the reflectivity gradient, anvil growth,

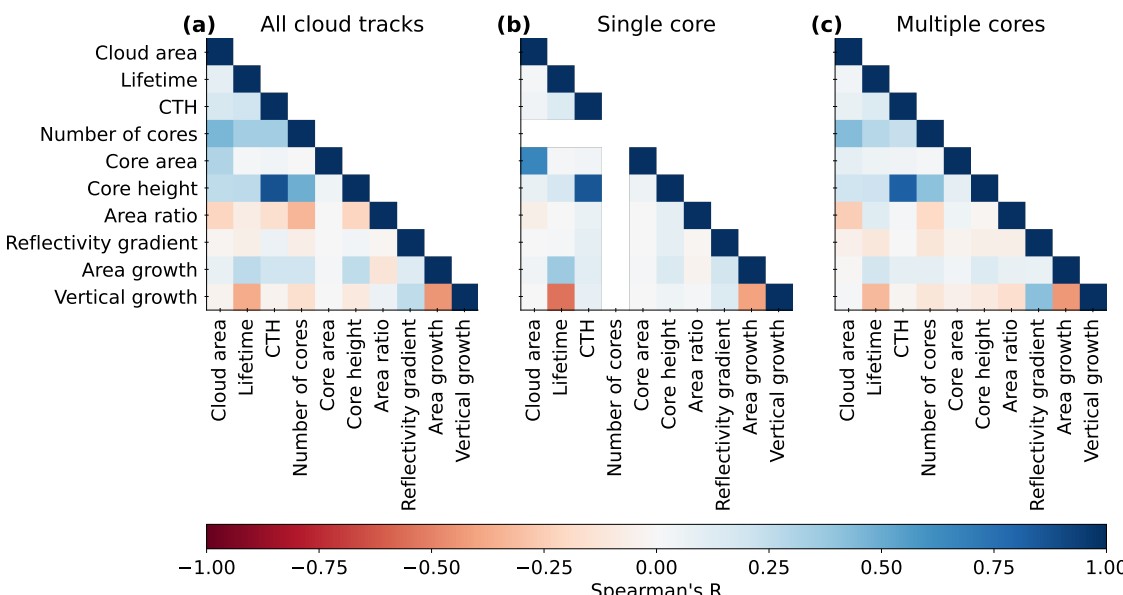

**Figure 14.** Correlation matrix for the life-cycle statistics, cloud and core properties. We calculate Spearman's R to quantify the correlation coefficient on a scale between -1 and 1 for **(a)** the whole dataset (n = 338,142), **(b)** clouds with a single core (n = 216,798), and **(c)** clouds with multiple cores (n = 121,344).

vertical growth, number of cores, and core area. Moreover, we derive the time of dissipation which marks the end of the cloud life-cycle.

- Reflectivity gradient: Peaks around 0.85 h on average, with little difference for the median between land and sea. However, the distribution broadens over both surface types with increasing core numbers. The arithmetic mean increases up
to 2.5–3 h for clouds with 10 and more cores (Figure 15, a).

- Area growth: Occurs at 1.64 h on average. Single-core clouds peak earlier (< 1.5 h), while multi-core clouds range from 1.7 h (2 cores) to 4 h (10 and more cores). For clouds with 3–5 cores, we find a predominantly similar distribution with only slight variations of the arithmetic mean (Figure 15, b).

- Vertical growth: Peaks around 1.19 h. This occurs earlier for single-core and oceanic clouds. For clouds with 4 (5) or
more cores over land (sea), the distributions are similar and show only a weak variability (Figure 15, c).

- Core area & core number: The maximum core area typically occurs 1.25 h after detection, while the core number peaks slightly later at 1.37 h. Surface type has little effect on the time of the maximum core area, though the timing increases with more cores (Figure 15, d). Clouds over land - especially those with 2–5 cores - reach their maximum number of cores later than oceanic clouds. Despite this observation, we find less of a linear timing increase compared to other
life-cycle statistics (Figure 15, e).

– Cloud dissipation: Clouds over land generally dissipate later than oceanic ones. Lifetime extends further with more cores, and value variability is also higher over land. Single-core clouds typically dissipate within 1–3 hours, whereas multi-core clouds last longer, with broader distributions and higher mean lifetimes (Figure 15, f).

Notably, the analysis shows that vertical growth may peak after the reflectivity gradient but before area growth. The times stretch for multi-core clouds, while single-core systems exhibit more compact timelines. However, outliers may distort observed mean values. Hence, the consecutive order of the life-cycle statistics may be affected by a high variability in the distribution. Across all cloud tracks, core number and core area tend to peak between vertical and area growth maxima. However, the distributions show a high variability, especially for multi-core clouds and clouds over land. While we observe life-cycle statistics occurring on average later for clouds over land, the differences induced by the surface type remain overall low.

### 4.3.2 Temporal variability of life-cycle statistics

Figure 16 illustrates the diurnal patterns of the reflectivity gradient, the area growth, and the vertical growth for single-core and multi-core clouds, grouped by surface type. Similar to results from Sect. 4.2.1, the diurnal cycle is more pronounced over land than over the ocean. The reflectivity gradient exhibits short-term fluctuations with noticeable nocturnal peaks around 20:00–21:00 UTC and 00:00–01:00 UTC, followed by decreases. During the day, peaks occur between 09:00–12:00 UTC and around 16:00 UTC, with a negative dip around noon (land) and 18:00 UTC (both land and sea). Over the ocean, the reflectivity gradient is generally higher and shows a slightly weaker variability than over land. Over land, multi-core clouds exhibit stronger gradients than single-core clouds. Distinct land-based negative peaks occur around 03:00–06:00, 08:00, and 11:00–15:00 UTC. Over the ocean, we find a weaker nocturnal peak at 01:00 UTC and a gradual increase between 06:00–20:00 UTC. Overall diurnal variability ranges from 0.5 dBZ (ocean) to 1 dBZ (land), or roughly 8–16 % of the mean gradient range (10–16 dBZ) (Figure 16, a, d). Multi-core clouds show significantly greater area growth (42–48 %) than single-core clouds (20–27 %). Over the ocean, we see sporadic daytime peaks that occur around 15:00, 20:00, and 01:00 UTC. Over land, area growth increases steadily for single-core clouds in the morning and peaks between 12:00–14:00 UTC. For multi-core clouds, we find several sporadic peaks during the day, similar to clouds over the ocean. Evening peaks appear around 18:00 and 22:00 UTC for multi-core clouds, and 20:00 UTC for single-core clouds. Diurnal variability remains low, ranging up to 5 % over both land and sea (Figure 16, b, e). Diurnal variation in vertical growth is weak over the ocean with values below 0.5 km. Over land, morning peaks are evident, particularly for single-core clouds, while all cloud types show a distinct dip around noon. Afternoon and evening values rise again, with multiple peaks (e.g., 17:00, 21:00, 06:00 UTC) and troughs (18:00, 22:00, 03:00 UTC), typically within a 1–1.5 km variability range. Over both surface types, single-core clouds reach higher altitudes (7.5–8.75 km) than multi-core clouds (6.25–7.5 km) (Figure 16, c, f).

Figure 17 presents monthly changes for the reflectivity gradient, the area growth, and the vertical growth between March and August 2019 for single- and multi-core clouds over land and sea. Overall, reflectivity gradients increase from March to August. For single-core clouds, the increase is more pronounced over the ocean (from 14.3 to 15.8 dBZ) than over land (13.5 to 14.25 dBZ). For multi-core clouds, values over land rise more (14.3 to 14.6 dBZ) than over the ocean (14.5 to 14.6 dBZ).

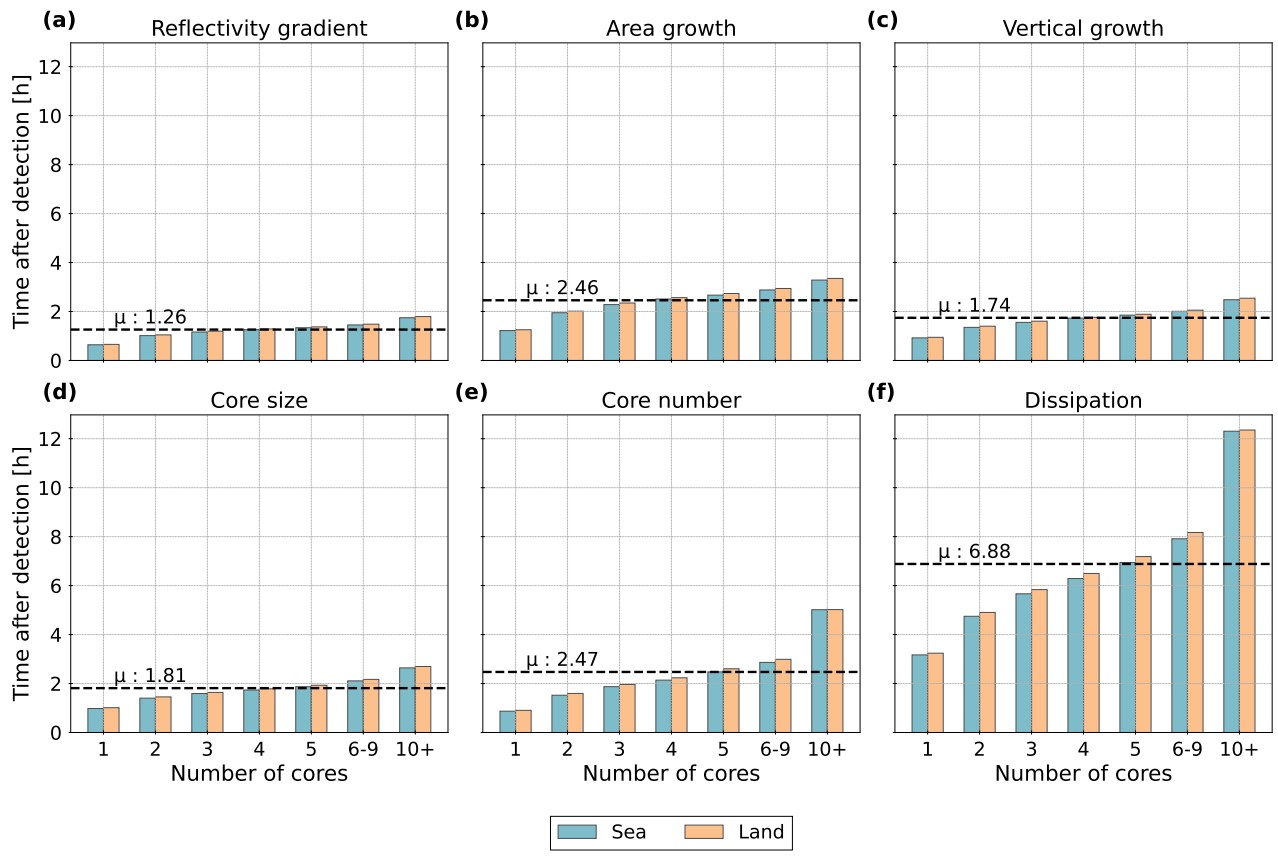

**Figure 15.** Distribution of the time dependency (x-axis) for the life-cycle statistics. The boxplot diagrams show the time after detection (in h) when the detected clouds reach on average the maximum value for **(a)** the radar reflectivity gradient at 10 km height, **(b)** the vertical growth, **(c)** the anvil growth, **(d)** the core area, and **(e)** the core number. Moreover, we show the average **(f)** dissipation time. Clouds are grouped by the surface type and number of cores (y-axis). The boxplot contains the median (bold blue or orange lines) and the arithmetic mean (blue or orange diamonds). The grey vertical lines show the mean time dependency averaged over all clouds tracks.

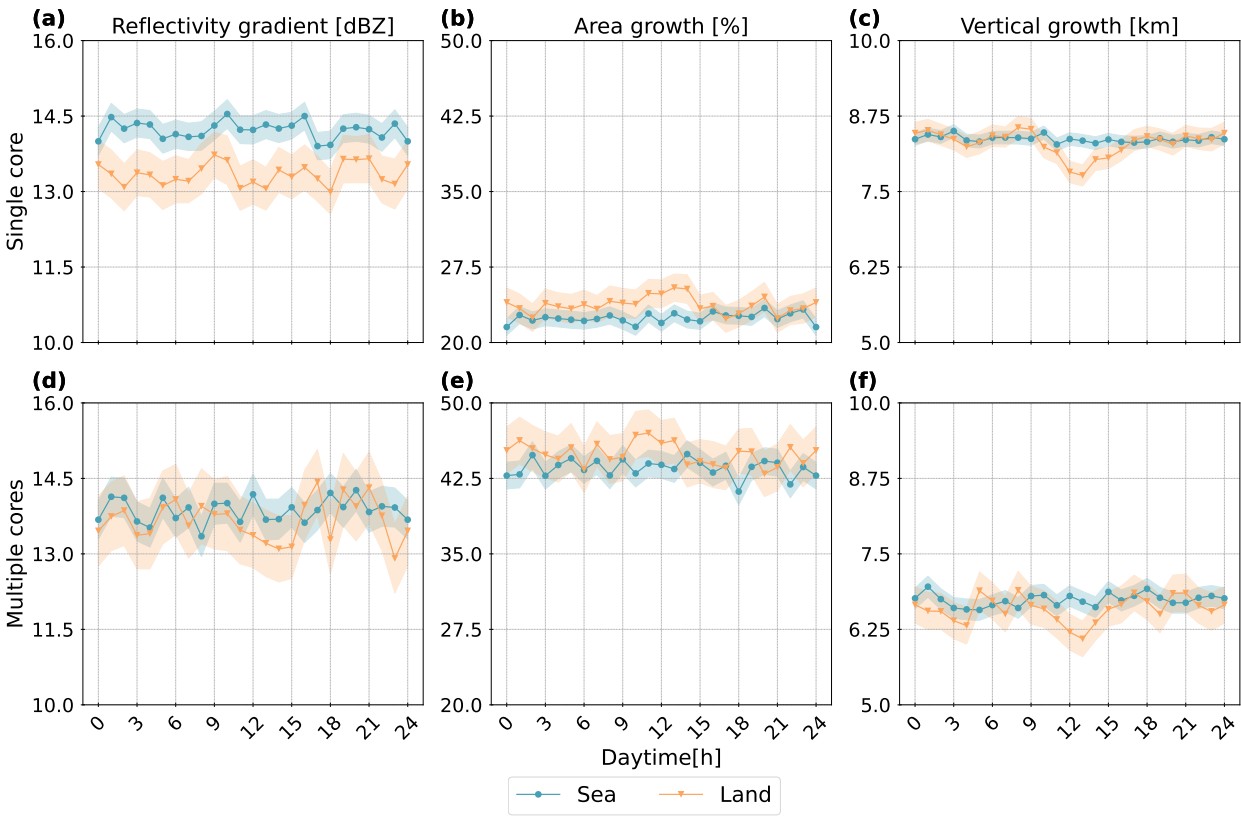

**Figure 16.** Diurnal cycle for the life-cycle statistics. We display the hourly changes (in UTC) regarding **(a)** & **(d)** the maximum cooling, **(b)** & **(e)** the cloud area growth, and **(c)** & **(f)** the cloud vertical growth for single- (1 core) and multi-core (>1 core) clouds over sea and land. Line plots show the mean value with a confidence interval of 95 %.

A notable dip occurs across all cloud types in April. Month-to-month variability is high (Figure 17, a, d). Between March and June, area growth is higher over land, peaking in May for single-core and in June for multi-core clouds. After June, area growth becomes higher over the ocean. Over the period, single-core clouds over land show a net decline (around 3 %), while values increase slightly over the ocean (around 5 %) and for multi-core clouds over land (around 3 %). Monthly changes appear to be nonlinear and fluctuate considerably (Figure 17, b, e). The vertical growth of single-core clouds peaks in March, drops sharply in April. From May onward, it rises again over land and decreases over the ocean. The total change ranges between 0.5–1 km. Early in the period, oceanic clouds have a higher vertical growth than clouds over land. From June onward, this pattern reverses (Figure 17, c, f). Throughout the study period, single-core clouds consistently show higher reflectivity gradients and vertical growth, while multi-core clouds exhibit greater area growth. Though the surface type may influence these statistics, the observed effect in our study remains relatively small. In contrast, the number of cores plays a more substantial role in shaping the cloud life-cycle.

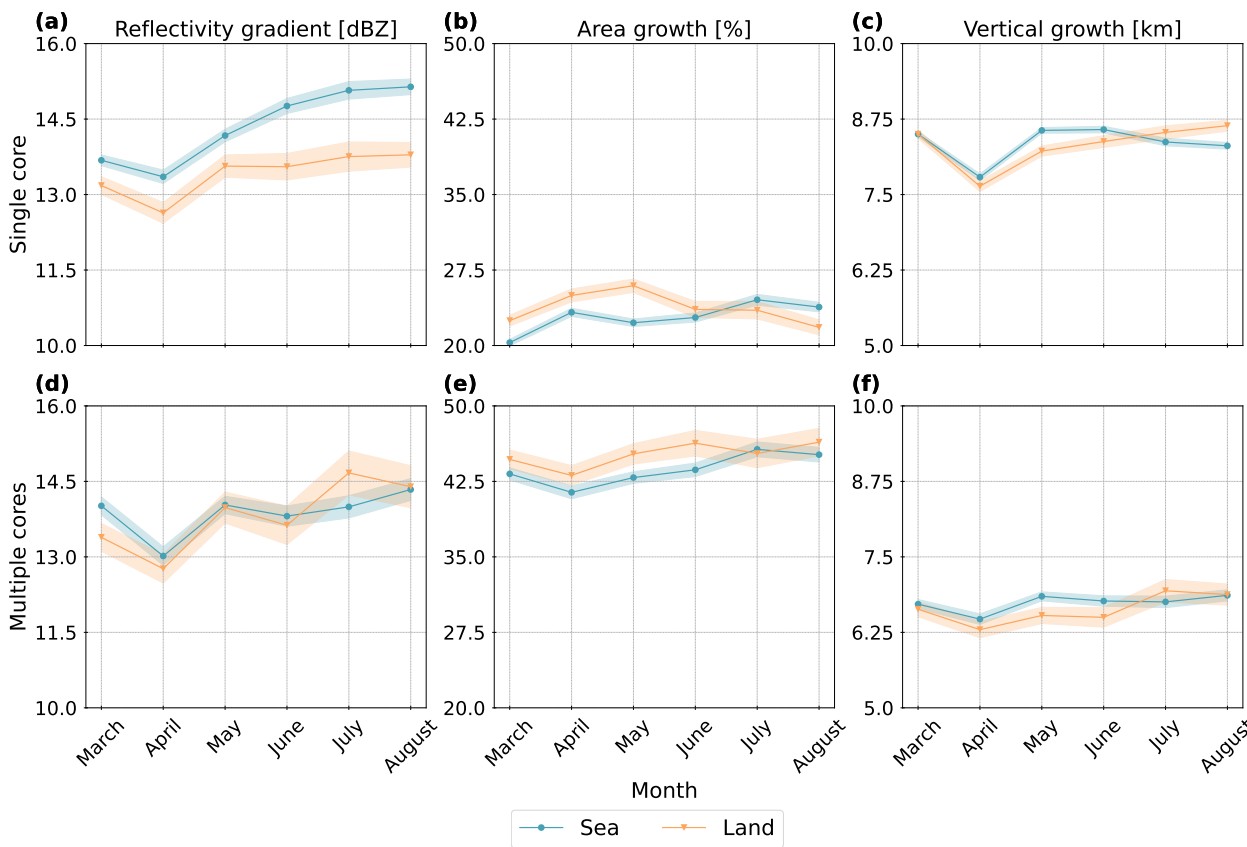

**Figure 17.** Monthly changes for the life-cycle statistics. In **(a)** & **(d)**, we show the maximum cooling, in **(b)** & **(e)** the anvil area growth, and in **(c)** & **(f)** the vertical growth for clouds with a single (1 core) core or multiple (>1 core) cores. Line plots show the mean value with a confidence interval of 95 %.

## 5  Discussion

### 5.1  Summary of results

In this study, we use a ML–based 3D extrapolation of radar reflectivity, derived from 2D satellite imagery, to characterise single- and multi-core convective clouds in the tropics. Our analysis shows that the majority of detected clouds are short-lived (under three hours) and dominated by a single core (65 %). Compared to multi-core clouds, these single-core systems tend to have a smaller cloud area, lower reflectivity at 10 km, and reduced CTH. Their cores are also typically smaller, shorter-lived, and lower in altitude — though they exhibit a higher area ratio between cloud and core area. In contrast, multi-core systems a larger in area, persist longer, reach a greater CTH and core height, and form larger individual cores (Section 4.1). Correlation analysis reveals a high interdependence among cloud and core properties. The strongest positive relationships are found between CTH and core height, and between the number of cores and both cloud area and core height (Figure 14).

Over land, the diurnal cycle exhibits a midday-to-afternoon peak (12:00–16:00 UTC) for the cloud and core area, the reflectivity at 10 km, the core height, and area ratio, with a secondary overnight peak for the cloud and core lifetimes. Over the ocean, the diurnal cycle is less pronounced but still shows afternoon, evening, and nighttime maxima. These patterns are especially distinct for single-core clouds (Section 4.2) and are largely consistent with prior tropical observations (e.g., Chen et al. (2021); Takahashi et al. (2017); Gupta et al. (2024)). Derived life-cycle statistics indicate that peaks for the reflectivity gradient

typically precede vertical growth and subsequent area expansion — mainly around local noon and early evening over continental Africa. In oceanic regions, a less consistent pattern emerges, with a high variability throughout the day (Section 4.3.2). Compared to Wilcox et al. (2023), we observe weaker diurnal amplitudes over both land and ocean; however, the differences between single- and multi-core clouds are more pronounced than those between land- and sea-based convection.

We also find that changes in cloud area, core number, and cloud height often evolve in line with an idealised convective life

cycle described in Sect. 3.6. Longer-lived clouds tend to exhibit more cores and larger maximum core areas. Multi-core systems reach their peak core number and core size later in their life cycle than single-core clouds (Section 4.3.1). The reflectivity gradients correlate positively with vertical growth and negatively with area expansion, reflecting transitions from development to maturity, as noted by Hu et al. (2021). Single-core clouds display stronger vertical ascent and higher reflectivity gradients, though most correlations are weak — aside from a strong negative relationship between cloud lifetime and vertical growth,

and a moderate positive link between lifetime and area growth. For multi-core systems, average area growth is about 20–30 % higher than for single-core clouds. Reflectivity gradient and vertical growth are negatively correlated with most other properties, while the area growth shows positive correlations with the core count, core area, and core height. However, correlations differ between single-core and multi-core systems (Figure 13, Figure 14). Finally, our results underscore how differences in core structure and number may influence convective cloud development and the associated life-cycle.

## 5.2  Influences on convective clouds in the tropics

Compared to previous studies by Takahashi et al. (2023) and Hu et al. (2021), our analysis identifies a significantly higher number of potentially convective cloud tracks. The derived cloud characteristics align well with aircraft observations (Zipser and LeMone, 1980) and precipitation-based studies (Zipser et al., 2006). Over tropical Africa, our core distribution results are consistent with those derived for geostationary satellite data (Jones et al., 2024) or the CloudSat CPR (Deng et al., 2016), both

of which found a high prevalence of clouds with one to three cores. Similarly, Pilewskie and L'Ecuyer (2022) reported that one-third to half of convective systems observed globally by the CloudSat CPR contain a single core. For the tropics, however, our results are in closer agreement. In line with these findings, we observe that cloud area generally increases with the number of cores. However, this relationship exhibits substantial variability, especially in multi-core systems (Section 4.3.1).

Our results show that convective cloud properties over land typically peak during the day, while over the ocean, we observe

two peaks during daytime and at night (Sect. 4.2.1). These findings align with the diurnal cycle of tropical convection (Vondou, 2012; Takahashi et al., 2023). A nocturnal enhancement over the ocean may be linked to the diurnal cycle of free-tropospheric humidity, which peaks overnight and supports convection (Wall et al., 2020). After sunrise, solar heating may stabilize the atmosphere. A weakening land breeze may lead to the dissipation of night-time clusters (Houze Jr., 2004). While these diurnal

patterns may be reflected by the cloud properties (Sections 4.2.1), differences in the daytime of the first detection for the cloud tracks appear weaker (Figure 9). Throughout the day, we observe several peaks for the reflectivity gradient, followed by phases of vertical and horizontal cloud growth (Section 4.3.2). Notably, as the number of cores increases, both reflectivity gradient and vertical growth decline, while area growth becomes more pronounced (Section 4.3.1). This finding may correspond to a higher reflectivity at 10 km and broader spatial extent seen for multi-core systems (Section 4.1). Our observations may point to a self-sustaining mechanism where cores are regenerated in response to diurnal heating (e.g., Deng et al. (2016); Hartmann et al. (2018); Takahashi et al. (2017)). However, we did not explicitly investigate this process.

Seasonal variability in tropical convection has been highlighted in past studies. For example, multi-core systems often persist overnight during the onset of the West African Monsoon (Futyan and Genio, 2007). During this period, convection may frequently initiate over high terrain and propagate downslope at night under katabatic flow (Nicholson, 2018). While our results may be influenced by interannual variability, as the dataset spans only one year and does not capture a full annual cycle, we may observe temporal changes in cloud and core properties between March and August. For instance, we find an increase of the cloud area, cloud lifetime, number of cores, and core area over the ocean. Over land, these properties slightly decrease (Section 4.2.2). Reflectivity gradients increase along the period, in particular over the ocean, while vertical and area growth vary more considerably (Section 4.3.2). Between June and August, we detect overall a lower proportion of cloud tracks than between March and May. However, we see a shift regarding the distribution of cloud tracks over land and sea along the period: While we detect a higher proportion of cloud tracks over continental Africa in March and April, the number of detected clouds over the ocean exceeds those over land from May to August. The differences between monthly averages over both surface types remain small. Here, our results may diverge from studies that report more pronounced spatial and seasonal variations for convective clouds over land and sea (e.g., Takahashi and Luo (2012); Wilcox et al. (2023)). More striking than these surface-type induced differences in mean cloud properties are the contrasts between single- and multi-core clouds. Longer-lived, multi-core systems often exhibit repeated phases of growth (Takahashi et al., 2017). Consistent with Taylor et al. (2017), we observe slightly larger cores, especially between March and May, and enhanced area growth for continental convection (Sections 4.2.2 and 4.3.2).

## 5.3 Limitations and future challenges

Our analysis is based on ML-derived 3D radar reflectivity fields. While the results may help to extend the data availability of global radar reflectivities, they possibly contribute to mitigating limitations in approximating the cloud vertical extent from geostationary satellite observations. However, we point out several important limitations. The input data for the ML model are based on observations from the Cloud Profiling Radar (CPR) aboard the CloudSat satellite, which has known limitations in detecting ice clouds due to its tendency to underestimate the height of upper-level outflow (Wang et al., 2014). Additionally, signal attenuation near the surface caused by topography reduces the CPR's sensitivity to shallow convection. As a result, our analysis underrepresents both shallow convective and cirrus cloud types. Emerging satellite systems, such as the Flexible Combined Imager on the Meteosat Third Generation (MTG) platform (Holmlund et al., 2021) and the enhanced CPR on the EarthCARE mission (Eisinger et al., 2024), offer improved spatial and temporal resolution. These instruments are expected to enhance the detection and characterisation of the convective cloud life-cycle. Moroever, our study does not account for

several potentially important influences on convection, such as aerosol interactions, vertical wind shear, and entrainment rates (Masunaga and Luo, 2016). Incorporating these factors in future analyses could lead to a more comprehensive understanding of convective processes. Our study focuses on a domain within the tropical band from 30° W to 30° E and 30° N to 30° S. As extratropical influences may blur the statistics at the domain's northern and southern edges, it may be beneficial to explore the intra-tropical variability and the role of large-scale dynamics beyond this region to distinguish tropical from midlatitude convective processes.

To identify isolated and clustered convective systems, we employ the *tobac* framework (Sokolowsky et al., 2024). However, this object-based approach relies on manually set thresholds for the detection, introducing a degree of subjectivity that may influence the resulting cloud statistics. This includes potential limitations in the analysis of the cloud life-cycle, which itself does not provide insights on actual changes of the cloud temperature. It is important to emphasise that no universally optimal detection algorithm exists (Lakshmanan and Kain, 2010); each method has context-specific strengths and limitations depending on the intended application and geographic domain (Prein et al., 2024). While our approach may enable an approximation of the vertical cloud column, radar reflectivity alone does not substitute for measurements of vertical wind shear (Luo et al., 2008). Although vertical shear may have a smaller impact on convective processes in the tropics than in mid-latitude regions, its role still warrants further investigation (Takahashi et al., 2017). Finally, to better assess current and future convective risks, particularly those posed by multi-core systems, future work should explore their associated precipitation patterns in more depth (Atiah et al., 2023).

## 6  Conclusions

This study analyses the properties and life-cycle of convective clouds and their deep convective cores over a tropical region covering the Atlantic Ocean and West Africa. Using an ML-based extrapolation of radar reflectivities, we may enhance the number of detected clouds compared to retrievals from CloudSat CPR alone. Hence, the approach may help to close current data gaps. In this study, we aim to showcase the 3D data and their ability to track convective clouds and cores along their life-cycle. Compared to using data from only either a passive or active sensor, our perspective may allow a simultaneous coverage of cloud development in the horizontal and vertical dimension.

The results suggest that differences based on the number of cores are higher than the surface-type induced variability. Single-core clouds develop and dissipate on shorter timescales. They have a smaller cloud and core area, and lower CTH and core height than multi-core systems. The longer cloud lifetime of multi-core clouds may be associated to a later occurrence of the maximum number of cores and core area. Between single-core and multi-core clouds, we find considerable differences in the cloud life-cycle statistics regarding the changes in the radar reflectivity at 10 km height, the vertical growth, and the area growth of the cloud. While the former two are higher for clouds with a single core, multi-core cloud clusters with a larger cloud area tend to grow more along the horizontal dimension. The more cores we find, the later the maximum number of cores and the maximum core area occur. While the differences between the convective clouds over land and ocean are lower

than expected, we emphasise our analysis uses six months of data and may not represent the annual cycle of convection. Nevertheless, expanding the approach to investigate a longer time series may account for current uncertainties.

In this work, we use the number of convective cores to compare the effects of spatial clustering. However, we think that it is worth comparing these results to a quantification of convective organisation using more advanced metrics, as done in the accompanying manuscript.

*Code and data availability.* The level 2B-GEOPROF CloudSat data used in this study are available at the CloudSat Data Processing Center at CIRA/Colorado State University and can be retrieved from http://www.CloudSat.cira.colostate.edu/order-data (CloudSat Data Processing Center, 2024). The Meteosat SEVIRI level 1.5 data used in this study is freely and openly available via the EUMETSAT Data Store at https://navigator.eumetsat.int/product/EO-:EUM:DAT:MSG:HRSEVIRI (EUMETSAT Data Services, 2024). The code used in this study will be released upon publication. The dataset of convective cloud tracks created in this study is available at the following repository: https://zenodo.org/records/14724869 (Brüning, 2025b). The material used to prepare this paper, including code used to perform analysis and that needed for the preparation of figures, is archived at https://zenodo.org/records/15607483 (Brüning, 2025a).

*Author contributions.* S.B and H.T. designed the study. S.B developed the code for the analysis and visualisation. S.B. and H.T. contributed to the analysis and evaluation of cloud tracks and properties. S.B. and H.T. wrote the draft of the paper. All authors have read and agreed to the published version of the manuscript.

*Competing interests.* The authors declare that they have no conflict of interest.

*Acknowledgements.* This work was supported by the project "Big Data in Atmospheric Physics (BINARY)", funded by the Carl Zeiss Foundation (grant P2018-02-003), and the Max Planck Graduate Center with the Johannes Gutenberg University of Mainz (MPGC). We thank EUMETSAT for providing access to the Meteosat SEVIRI imager data and the Cooperative Institute for Research in the Atmosphere, CSU, for providing access to the CloudSat 2B-GEOPROF data.

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
