# Peer review of "A machine learning-based perspective on deep convective clouds and their organisation in 3D. Part I: Influence of deep convective cores on the cloud life-cycle"

_EGUsphere, 2025_

## Referee Comment (RC2)

**Summary & Evaluation**

The focus of this analysis was to apply a tracking method to machine learning-based predicted 3D radar reflectivity structures for distinguishing convective core and system lifecycle characteristics. Data used were passive remote sensing channels from SEVERI on board Meteosat and reflectivity profiles from the Cloud Profiling Radar on board CloudSat. Despite the short time sample of using only six months of data, 375,000 convective systems were detected which made it a sufficient analysis for generating statistics, at least for capturing the diurnal cycle of convection and their differences between land and ocean. It would be worth highlighting potential known limitations in extracting a seasonal cycle given that only one year of data was used for the analysis. Overall, the analysis is consistent with what is already known of the convective lifecycle and potentially adds new insight into the lifecycle features of convective cores. The paper would benefit from improving upon and clarifying the analysis section, and details for doing so are outlined in both the General and Specific comments. Finally, it would be greatly appreciated if the authors comment on any limitations regarding interpreting the lifecycle of predicted radar reflectivity structures as "truth".

**General Comments**

*Introduction*: It is best practice to choose the earliest references that support statements. The authors appear to have chosen very recent papers, which are still valid to keep, but it is recommended to find more sources that are older. In particular, I am referring to *L18* [Roca et al., 2010], *L19* [Chen et al., 2021], *L20* [Kukulies et al., 2021; Haberlie and Ashley, 2018], *L22* [Prein et al., 2024], *L33* [Fiolleau and Roca, 2013], *L47* [Haberlie and Ashley, 2018], *L132* [Prein et al., 2024]. A good reference for background on cloud feedbacks and climate sensitivity is Sherwood et al., (2020). Leary and Houze, (1979) goes into detail on the convective cloud lifecycle.

*Data and Method*: It would be beneficial if you just focused on describing all the data used in the Data section and then had a separate section in the Methods that described the ML algorithm that was published previously. It is unclear if the details in *L106-128* are summarizing the ML algorithm described in your previous paper. Please also clarify the following:
- It is discussed how CloudSat, but not how Seviri, data are being used. Are both CloudSat and Seviri data used to train the model?
- Could you briefly describe how you computed the average error of the model?
- How do you test the model to evaluate its performance, if at all?
- It should be clearly stated that the methodology outlined in the "Methods" section (as it currently stands) are applied to the predicted radar reflectivities outputted from the ML model.
- Descriptions of both Sections 3.2 (Detect convective core regions) and 3.3 (Extract cloud and core properties) should be clarified greatly. Details for how they should be clarified are under Specific Comments.

*Results*: The descriptions and wording in the analysis are somewhat vague and need to be clarified. For example, it is not understood how "convective activity" is defined, and this phrase

is used several times throughout the text. More clarifications to the results can be found under Specific Comments.

*Seasonal cycle*: Are there any potential limitations in your statistical analysis of the seasonal cycle given that you are only using one year of data? Can you comment on if interannual variability (e.g. ENSO) might have impacted such results?

**Specific Comments**

*L8, L360*: What is meant by "absolute" cooling?

*L42-43*: Bacmeister and Stephens, (2011) is not a good reference that investigates "the temporal evolution of MCSs. A study using geostationary satellite observations or other such measurements that capture the temporal evolution is preferred.

*L67*: Why is it that single convective cells are typically tracked using data from active remote sensing sensors? And specifically which sensors? Are different measurements typically used to track single-cell systems compared to MCSs?

*L113*: Could you please clarify what a "spatio-temporal matching scheme" is?

*L149*—"weighted mean": Weighted by what?

*L154-155*: It is unclear what "The value of each pixel is decreased towards its local minimum using a threshold of -15 dBZ" means.

*L163*: Why do you need to define a system as "elongated"? Is it under the assumption that a system should be near circular, and that it is elongated due to the merging of multiple systems? *Section 3.2*: Have you validated your core detection method? It would be interesting if you apply your core detection method to physical 2B-GEOPROF radar reflectivities and then see how it compares to cores detected using the Conv_strat_flag from 2C-PRECIP-COLUMN (Pilewskie and L'Ecuyer, 2022; data description here) for a month of data, if possible.

*L194*: Why do you expand the threshold?

*L194-195*: What is "the first criterion"?

*L195*: In the case of significant attenuation within a convective core region, the CBH may reach up to 5 km. Have you looked into this? How do you account for attenuation in core profiles?

*L195-196*—"if the vertical profile shows no convective pixels for more than 50% of the CTH": Please clarify what is meant by this statement. Firstly, how are the convective pixels defined? There is a definition for DCCs in *L190-192*, but it is not sufficient as it does not give any thresholds to the maximum "column-wise aggregated radar reflectivity" nor the "difference between the CTH and CBH" used to isolate DCCs. Secondly, what is meant by "more than 50% of the CTH"?

*L196-197*: It is not understood what is meant by "we add the mean radar reflectivity of the vertical profile to the derived vertical depth of the column". Please clarify.

*L212*: Please specify what is meant by "We employ the radar reflectivity at a fixed altitude of 10 km as a measure of convective activity". What defines "high" versus "low" convective activity?

*L219, L232-233*: It is not understood if there is a general CTH threshold applied to the clouds for isolating convective systems. As it currently reads, it seems clouds need to have a CTH of at least 10 km; however, clouds during the convective initiation (CI) often have cloud tops below 10 km. For this reason, I am not convinced that you are capturing systems during the CI stage.

*L220*: Please describe the assumptions made for using the "difference between the radar reflectivity at 10 km height at CI and the current time step" to approximate cooling? Also, what is meant by cooling, is it at the top of the atmosphere, atmospheric cooling, or with regards to the surface?

*L221-222*: These sentences would be better understood if they were merged. Also, what is meant by "difference between the CBH and CTH at CI for each time step, compared to CI"?

*L223*—"difference of the radar reflectivity": difference with respect to what exactly?

*L238*: If I am interpreting correctly, are the convective clouds that are identified actually "real", as in you would be identifying these systems in SEVIRI if you were to be just tracking them in 2D? It's just the 3D cloud fields that are predicted, hence not "real"?

*L240:* What is considered "high reflectivity"? 0 dBZ at cloud top is not necessarily a "high" radar reflectivity.

*L250-252*: From the statistics stated in the text, it does not seem intuitive that only 10% of the population contain 2-10 cores. What are the exact counts in each bin? When studying the histogram in Figure 6, it looks like single-cell systems make up more like 65% of the population, and systems with ten or more DCCs is only ~3%, meaning that there is a larger population containing 2-10 cores than what is being inferred in the text. How do these statistics compare to other studies that have assessed the statistics on the number of cores in systems?

*Figure 5*: It would be easier to distinguish cores from the rest of the cloud if there was a larger contrast between the colors. Perhaps make the cores a deeper red.

*Figure 6*—"surface type derived from a land-sea mask compared to the location at CI": I don't understand this distinction.

*L268*—"0.5 dBZ higher over the ocean than over land": How might this tie into the notion of differences in intensity over land versus ocean? Does this suggest oceanic convection is more intense of land, which differs from our general understanding of tropical convection?

*L272-273*: The sentence beginning with "The core lifetime…" is somewhat challenging to interpret. Please clarify.

*L276-277*—"especially for convective clouds over land": It appears that both regions show this, not just land.

*L286-287*: Have other studies shown that continental clouds have a larger anvil area and lower reflectivity compared to clouds over the ocean for the region and time period you are studying?

*L287-288*: Please clarify this explanation—differences in cloud properties would cause a local thermal instability—is that how it is to be read?

*L291-294*: Please include references here, and are you describing the diurnal cycle for this specific region?

*L298*: Figure 9 a-d does not show the eccentricity of clouds.

*L304-305*: Second maximum in what, exactly? Where is the first maximum? Also, please use a more scientific word than "powerful" in "afternoon peak is consistently more powerful". Also, what afternoon peak are you referring to?

*L315, L371*: How do you define convective activity? Is this shown in Figure 11?
*L315-316*: Are you explaining that this is occurring over time, and witnessed over both land and ocean and for all cores?

*L316-317*: Do you mean to imply that the number of DCCs increases as the months progress? There is a sharp drop though between July to August over sea.

*L318-319*: Perhaps use wording other than "less distinct" to make your point clearer. It seems like sea has more cores than land starting in May.

*L322-323*: Where do you get the anvil extent being larger over land than ocean?

*L332-333*: Again, has this definition of cooling been used previously in literature? Please explain the assumption.

*L352*: Did you quantify "the cooling and area growth appear earlier during the relative cloud lifetime" to determine it? It is not clear based on the figure.

*L357-358*: For cooling, there is a morning peak at 0100 over both land and ocean. Is it statistically larger than the mean for the next several hours? There are dips at 6 pm over both land and ocean, and an additional dip at 12 pm over land. Again, are these statistically different and indicative of the diurnal cycle? The diurnal cycle of vertical growth seems to be much more pronounced over land, but not noticeable over ocean. Please make this distinction.

*L359-360*: Which convective characteristics are you referring to?

*L361-363*: Is this suggesting that as the number of DCCs increases, the total cooling and vertical growth per DCC is less than that of the initiation phase?

*L365*: How about the peaks in the early morning?

*L367*: Aside from the dip in July over land.

*L368-369*: Why do you think the cooling and vertical growth are higher for clouds with a single core?

*L371-372*: Are you referring to the larger variability in area and vertical growth of convection over land? In other words, the seasonal cycle of convection is more pronounced over the ocean, but the diurnal cycle of convection is more pronounced over land?

*Section 5.1*: This section is great, I appreciate the connection between your results and previous literature. Could you also add a few sentences on discerning the characteristics between isolated and multi-cellular convection within previous literature, and how it is consistent with what you are finding?

*L405-407*: Had you considered also using CALIPSO observations to capture thin ice clouds?

*L429-431*: Convection as a self-maintenance mechanism has been suggested in previous literature, but which signatures in your results are pointing to this conclusion?

*L432-433*: What do you mean by "weaker convective activity"?

**Technical Corrections**

*L12-13*: The way that this sentence reads is a bit confusing. I'd suggest "…more intense over land than ocean during both seasons, despite an increase in convective activity over the ocean during summer".

*L58*—"Early studies": Change to "Some studies" as a study provided in the next sentence was published earlier than Masunaga and Luo, (2016).

*L70*—"Contrasting": Change to "Contrastingly" or "Alternatively"

*L93*—"diverging convective development": "Diverging" is not the best descriptor here, so rephrase to say that convection develops differently in the extra-tropics compared to tropics.

*L95-96*: sentence beginning with "Clouds…" seems out of place in this paragraph.

*L161*: Does "width" refer to the vertical cloud thickness? Please clarify that here and in the workflow figure (Figure 1).

*L204*: ", e.g."

*L206-209*: Although important, this information is already addressed in the introduction, so can be removed to make the paper more succinct.

*L240*: Morocco

*L271*: Replace "connected" with "associated with"

*L273*: Replace "come along" with "associated with" or an alternate phrase

*L273*: By more extensive, you mean larger horizontal area, not more extensive in the vertical direction? Please clarify.

*L274*: Please modify the phrase "increases stronger".

*L300*: Does "temporal shift of the afternoon peak" mean that the peak occurs later in the day?

*Figure 9*: Add units to the "Cloud lifetime" axis.

*Figure 10*: Label axis "core eccentricity" for consistency.

*L371*: Can delete sentence beginning with "Clustered clouds…" because this was implied from the previous sentence.

*Figure 15*: Make the y-axes ranges consistent between single and multiple DCCs so it is easier to compare them.

*L378*: In addition to specific horizontal or vertical dimension, you can also include that there is a limitation in the temporal dimension if trying to exclusively use CloudSat data.

**References**

Leary, C. A., and R. A. Houze, Jr., 1980: The contribution of mesoscale motions to the mass and heat fluxes of an intense tropical convective system. *J. Atmos. Sci.,* **37**, 784-796.

Pilewskie, J. A., & L'Ecuyer, T. S. (2022). The global nature of early-afternoon and late-night convection through the eyes of the A-Train. *Journal of Geophysical Research: Atmospheres*, 127, e2022JD036438. https://doi.org/10.1029/2022JD036438

Sherwood, S. C., Webb, M. J., Annan, J. D., Armour, K. C., Forster, P. M., Hargreaves, J. C., et al. (2020). An assessment of Earth's climate sensitivity using multiple lines of evidence. *Reviews of Geophysics*, 58, e2019RG000678. https://doi.org/10.1029/2019RG000678.

---

## Author Comment (AC1)

**Final response**
**(Manuscript "EGUSPHERE-2025-374")**
**Reviewer 1**

Sarah Brüning and Holger Tost

May 23, 2025

**1   Introductory remarks**

We would like to thank the reviewer for taking your time to write the constructive and detailed feedback. In the following sections, we address all comments. For every comment, you can find (1) the comment, (2) the author's response (both with lines from initial manuscript), and (3) the author's changes in the revised manuscript. The lines given for **author's changes in the manuscript** refer to the lines in the **revised manuscript**.

Based on all referee comments, we focus the revision on improving the clarity of the written expression for, i.e, the description of our methods and the results. As suggested in your review, we aim to include a more detailed acknowledge about the subjectivity of our object-based detection approach and derived cloud life-cycle. While the narrative and main points of the manuscript did not change, we aim to focus on directly describing observed results, in particular regarding the connection between convective cores and cloud properties. Based on both referee comments, we agree the term "seasonal" may be misleading for the depicted period of only six months. Instead, we describe "monthly" changes, which we observed during the period. Moreover, we changed the abbreviation "DCC" to "cores" to align with our manuscript for Part 2 of the study ("EGUSPHERE-2025-376").
We changed the figure labels in the following way:

- 1 -> 1

- 2 -> 3

- 3 -> 2

- 4 -> 4

- 5 -> 5

- 6 -> 6

- 7 -> 7

- 8 -> 8

- 9 -> 10

- 10 -> 11

- 11 -> 12

- 12 -> Removed

- 13 -> 13

- 14 -> 15

- 15 -> 16

- 16 -> 17

- New: 9 (Distribution of cloud tracks grouped by the surface type: month, daytime of first detection)

- New: 14 (Correlation matrix for cloud properties, core properties, and life-cycle statistics for all cloud tracks, clouds with a single core, and clouds with multiple cores)

**2 General comments**

- **Comment**: The paper's perspective might be broadly categorized as an "object-based" scientific worldview. The basic idea is that atmospheric dynamics are best understood as interactions between discrete components – clouds, MCSs, DCCs – and therefore the geometry and spatiotempral distribution of such objects is of key importance in our scientific understanding of weather and climate. Although concepts like MCSs and DCCs are useful and important, in my opinion it is crucial to remember that such concepts are not fundamental and represent a subjective, imprecise, and qualitative description of the atmosphere. The reality is that the atmosphere is made up of continuous three- dimensional fields of matter and energy. This may seem like a pedantic philosophical point on my part, but the implication is substantial: quantitative object-based results such as those presented here are likely strongly dependent on the subjective definitions and thresholds used (specific examples are below). The issue is not that a given threshold might be "wrong," but rather that any threshold will be imperfect given that the basic object being defined (e.g. an MCS) is largely subjective in the first place. Because the objects under consideration (MCSs, DCCs) are fundamentally qualitative concepts, any study must introduce subjective thresholds and definitions to obtain quantitative statistics. Some metrics, such as cloud area, will obviously depend strongly on the threshold used to define cloud. Many other metrics, even some that seem at first glance less subjective, will also be sensitive to thresholds. For example, any cloud with an eccentricity higher than .75 is split into two clouds if the cloud has a "neck", for example if the cloud is hourglass-shaped (Sect. 3.1.2). This is performed in order to avoid "incorrect label assignments" (L139). Statistics such as number of cores per cloud, mean eccentricity, mean cloud size, cloud/core area ratio, etc. will all strongly depend on whether this step is performed or if the threshold of .75 was changed. It should be stressed that the problem is not that one method might be "incorrect" as implied by L139. Rather, the act of dividing an atmospheric volume into discrete clouds is a subjective process that imposes an artificial discrete framework on a continuous field. Discrete categories do not exist in the real atmosphere. To address the above observation I do not suggest any major revision of the methodology, but I would like an acknowledgement about the subjectivity of the chosen thresholds and at least some discussion about the inherent subjectivity and imprecision of dividing a continuous field into discrete objects.

- **Author's response**: We thank you for your comment and agree with you about the issues that may be connected with concepts that divide the atmosphere into qualitative categories. In the revised version, we try to point out more clearly the subjectivity of our object-based object with fixed thresholds and the derived life-cycle statistics. Also, we will add a more detailed description of the uncertainties and limitations to provide a better acknowledgment of this inherent subjectivity.

- **Author's changes in the manuscript**: Throughout the text, and in particular in Sect. 3 (Methods) and Sect. 5 (Discussion), we address the subjectivity of the approach more directly. You may find examples, e.g., in lines 136-140: "The workflow to identify possibly convective trajectories consists of three steps: detecting cloud features by their centroid's position, segmenting the associated cloud field for each centroid, and linking segmented objects through time (Figure 1, a–c). Moreover, we aim to separate cloud clusters that are only connected by a few pixels in the horizontal and vertical dimensions (Oreopoulos et al., 2017). The workflow of this object-based approach is depicted in Fig. 1 and will be explained in the following paragraphs.", lines 148-153: To identify potential cloud structures, we apply a fixed threshold of –15 dBZ to distinguish signals of hydrometeors from background noise in the radar reflectivity data (Marchand et al., 2008). While this threshold is only moderately restrictive — allowing for the inclusion of short-lived or weak features — it is intentionally chosen to capture the full spatio-temporal evolution of convective clouds between development and dissipation stage (Esmaili et al., 2016)", or lines 139-145: "The framework, while enabling detailed analysis of convective cores, has limitations. The predicted 3D cloud fields represent model-based approximations rather than direct

observations, reflecting patterns learned by the ML model. Additionally, using fixed thresholds in the object-based detection may oversimplify complex structures associated to clouds in the atmosphere. Nonetheless, we may employ the data to enable a large-scale, high-resolution tracking of convective systems over the tropical Atlantic and continental Africa."

- **Comment**: 6 months of data is not enough to get meaningful results on the seasonal characteristics (Sect 4.2.2, L366-L374, conclusion). Each season (spring and summer) have an N of 1, obviously not statistically robust. This is clearly seen in the 7 day means in Fig 11d, where there is clear variability due to the subseasonal meteorology that has a larger magnitude than any seasonal trend (note how non-smooth the curves are). Recommend either removing these sections or analyzing at least decades of data
  **Author's response**: We agree that the period is not sufficient to derive a statistical significant climatology of the seasonal or annual cycle of convection. In our study, we compare the two seasons (March-May, June-August) to showcase changes in the depicted period. We revise the manuscript to point out our goal more clearly (which is not an analysis of seasonality in statistical sense, but to highlight the findings achieved with our approach).
  **Author's changes in the manuscript**: Please see, e.g., Sect. 2, lines 74-79: "Our objective is to detect and analyse convective clouds and their life-cycles by a six month period between March and August 2019. This period was selected to highlight key characteristics of 3D cloud structures across different surface types within the AOI. Particular attention is given to the seasonal northward migration of the Inter-Tropical Convergence Zone (ITCZ) and the onset of the West African Monsoon (WAM). Since the WAM plays a critical role in shaping West Africa's climate and is responsible for a significant portion of the annual rainfall in the AOI, its arrival is expected to enhance the frequency of convective cloud formation (Andrews et al., 2024; Kniffka et al., 2019)." or Sect. 5.2, lines 506-510: "Seasonal variability in tropical convection has been highlighted in past studies. For example, multi-core systems often persist overnight during the onset of the West African Monsoon (Futyan and Genio, 2007). During this period, convection may frequently initiate over high terrain and propagate downslope at night under katabatic flow (Nicholson, 2018). While our results may be influenced by interannual variability, as the dataset spans only one year and does not capture a full annual cycle, we may observe temporal changes in cloud and core properties between March and August."

**3 Specific comments**

In the next section, we shortly address your specific comments. Changes will be added in the revised manuscript.

- **L77**: "subject to a spatio-temporal sampling in either dimension" is unclear but reads to me as just saying that spatial and temporal resolutions are finite... which is obvious and applies to all data, what did you mean?
  **Author's response**: In the original manuscript, we wanted to point out that active sensors provide detailed information on the vertical column, whereas passive sensors have a higher coverage of the horizontal dimensions (longitude, latitude). However, no sensor covers all dimensions while achieving a high temporal resolution and broad spatial coverage (i.e., like a global 3D sensor). We change the sentence to be more clear.
  **Author's changes in the manuscript**: Sect. 1, Lines 56-59: "Active and passive sensors contain important vertical or horizontal information, but are limited in their spatial and temporal coverage (active) or offer only an approximation of the vertical column (passive) (Masunaga and Luo, 2016; Taylor et al., 2017). To address this gap, we apply a machine learning (ML) framework to reconstruct contiguous 3D radar reflectivity fields from 2D satellite data (Brüning et al., 2024). "

- **L79-85**: I assume that Cloudsat is only used during ML training and only the geostationary data is used throughout the study. This is not clear in L79-85 and also 424, which imply your data are superior because they use multiple sensors ("Compared to using data from a single sensor, our perspective allows..."), which is not true. I don't see how I could be wrong here given that the data resolution is 15 minutes whereas Cloudsat's temporal resolution is roughly 16 days, but please clarify.
  **Author's response**: The ML model is trained using MSG SEVIRI channels to predict the vertical cross section of the CloudSat CPR containing radar reflectivities (hence, the model is validated against these cross sections). To receive pairs of MSG SEVIRI patches and the corresponding CloudSat cross-section, we search for overpasses of CloudSat along the domain in the tropical belt. Due to the symmetrical architecture of the Res-UNet, it may predict not only the reconstructed cross-section, but a contiguous 3D image. These predicted data incorporate

information from both sensors to represent the horizontal and vertical distribution of the cloud field (which comprises, for the final output of merged patches, 60°N–60°S and 60°E–60°W over 90 height levels between 2.4–24 km height). The predictions keep the temporal resolution of MSG SEVIRI (15 minutes).

**Author's changes in the manuscript**: Please see Sect. 3.1 for a more detailed description of the machine learning model, e.g., lines 111-114: "The Res-UNet is trained to reconstruct CloudSat-like 3D reflectivity volumes with a horizontal size of $100 \times 100$ pixels and a vertical size of 90 levels. The predicted radar reflectivity values range from –25 to 20 dBZ and retain the 15-minute temporal resolution of the original SEVIRI input."

- **Fig 7/8**: There is a fair bit of discussion about land/sea differences, but I find all land/sea results so close that they appear statistically insignificant. For example, the largest differences are in the reflectivity in 7e. The largest difference to my eye looks to be about a quarter dBz, although it is claimed the mean difference is 0.5 dBz (L268). Both are much smaller than the RMSE error calculated from ML model validation, which is 3dBz (Table 2).

  **Author's response**: We agree and revise the description of the results.

  **Author's changes in the manuscript**: In Sect. 4.1, we revised the description of Figures 7/8, e.g., in lines 289-293: "Land–sea differences are more pronounced for single-core clouds. Despite expectations based on previous tropical studies (Deng et al., 2016; Takahashi et al., 2017), oceanic clouds often show stronger reflectivity and larger areas — though overall surface-related differences remain small. The lower number of land-based clouds may exaggerate statistical noise.". Further discussion may be found in Sect. 5.2, e.g., lines 510-514: "However, differences between monthly averages over both surface types remain small. Here, our results may diverge from studies that report more pronounced spatial and seasonal variations for convective clouds over land and sea (e.g., Takahashi and Luo (2012); Wilcox et al. (2023)). More striking than these surface-type induced differences in mean cloud properties are the contrasts between single- and multi-core clouds. Longer-lived, multi-core systems often exhibit repeated phases of growth (Takahashi et al., 2017)."

- **Fig 9**: In Fig 9, much is made out of a "bimodal distribution" in the time direction, which is technically visible but I suspect is not really statistically significant. To me it looks the valley between the peaks barely crosses a single contour in all cases, so it's not very clear. A much more striking result is the bimodal distribution in the reflectivity direction in 9k and l, where the valley is many contours below the peaks, and there is a big difference between multiple and single DCCs. This is not really discussed much and I would be curious to hear the author's take.

  **Author's response**: We agree and revise the description of (old) Figure 9. We checked the data to investigate the distribution in the reflectivity direction in 9k & 9l and found a bug in calculating the 2D densities in our former version that led to an overestimation of high values in the density histogram. We add an updated version in the revised manuscript. Here, the radar reflectivity does not reveal a bimodal distribution, but rather depicts the overall increase for multi-core clouds compared to single-core clouds as shown in Fig. 7. At the same time, the revised code fixed the apparent bimodal distribution of the area ratio (Fig. 10 m-p). In the revised version, there is no valley but rather a high variability for single-core clouds and a lower variability with overall lower values for multi-core clouds.

  **Author's changes in the manuscript**: Sect. 4.2.1 for a revised description of the findings observed in (new) Figures 10 and 11, lines 314-337.

- **Fig 9, 10**: Perhaps the most notable feature of figs 9 and 10 is that all the contours seem to approach a frequency value close to zero at hour 0 and 24. These plots should "wrap" such that the contours line up at that location, and they might, but the minimum in frequency there really looks spurious to me. I encourage the authors to check their code for why this is happening. (Note that the implication would be that almost no clouds live past midnight.)

  **Author's response**: Actually, we found clouds living past midnight (e.g., Figure 9, a–d). Hence, we suggest the problem lies within the visualization. In the original manuscript, we used hourly bins to calculate the density distribution, but included no cyclic point. We update the figure to wrap up hourly bins at nighttime. We revise the figure and its description to be more clear.

  **Author's changes in the manuscript**: See Sect. 4.2.1 for a revised version of Figure 10 and 11, lines 314-337.

- **L301ff.**: Some points in paragraph starting at L301 seem quite exaggerated: • "a bimodal distribution over the sea and a singular afternoon peak over land:" I don't see that at all. b (land) and i (sea) have similarly significant bimodal dists (neither is that significant). c (sea), g (sea), e (sea) all have nearly zero evidence of a bimodal

distribution. • "the afternoon peak is consistently more powerful (Figure 10,c–d)" there is no peak at all in c, and in d it is only one contour level. • "Cores are more extensive and persistent over land" actually both a/b and c/d show nearly the same core area over land/sea. The "persistent" claim is more correct.

**Author's response**: We revise the results presented in this section and their description to be more clear.

**Author's changes in the manuscript**: Please see Sect. 4.2.1 for the description of Figures 10/11. E.g., lines 325-333: "The diurnal patterns of core properties (Figure 11) largely mirror those of the cloud properties. Over land, core area peaks between 12:00–18:00 UTC for both single- and multi-core clouds. Over the ocean, single-core clouds show two peaks between 00:00–06:00 and 14:00–20:00 UTC (Figure 11, a–d). The core lifetime follows a similar pattern for single-core clouds. For multi-core clouds, cores over land show two peaks, while oceanic cores point out no clear diurnal variation for multi-core clouds (Figure 11, e–h). For single-core clouds, peaks of the core lifetime resemble the core area (Figure 11, a, e). The distribution of the core height follows those of the core area (Figure 11, m–p). On average, clouds with multiple cores have higher and more variable values for core area, lifetime, and height. In contrast, the area ratio is lower and has a weaker variability for multi-core systems. For single-core clouds, we observe an afternoon peak over land and nocturnal and afternoon peaks over the ocean. Multi-core clouds show a weak diurnal variation, particularly over the ocean (Figure 11, i–l)."

- **L218ff.**: MCS "cooling:" The lifecycle of an MCS is idealized as occuring in 3 stages, the first of which ends at maximum "cloud cooling," calculated as a change between the 10km reflectivity between CI and the time step of interest (paragraph at L218). This tripped me up as the link between reflectivity and temperature changes is not at all obvious, especially when comparing to a baseline at CI where the cloud will often not even extend to 10km. I suspect the authors mean something more like "convective activity". Just substituting this term would make a lot more sense in many places, but either way please explain what is meant further. (note the paragraph at L329 and Fig 13 will need to be changed, and other brief mentions elsewhere)

  **Author's response**: We add a more detailed explanation of the statistics to describe the cloud life-cycle in Sect. 3.6. Moreover, we agree the term "cooling" may be misleading since we do not assess any cloud temperature directly in our study. Instead, we analyse the development stage of by calculating temporal changes in the radar reflectivity at 10 km height and searching for a maximum increase after the first time step of detection (References added in the manuscript). In response, rename "cooling" to "reflectivity gradient" to describe this step more clearly.

  **Author's changes in the manuscript**: See Sect. 3.6, lines 240-246 for a description how we approximate the development stage of convection by using the radar reflectivity at 10 km height: "Unlike methods that assess cloud stages using a cooling induced by temperature changes, the ML-based radar reflectivity does not provide information on temperatures. As an alternative, we approximate the life-cycle using temporal changes in radar reflectivity at 10 km height and the resulting vertical and horizontal cloud characteristics. For estimating the vertical growth of the cloud, we compute the difference between CTH and CBH (i.e., to display the height of the cloud layer) for every point in time. For the horizontal growth of the cloud, we calculate changes of the cloud area derived as proportional differences to the cloud area at the first timestep of detection." and lines 247-253: "Development stage: Building on the approach by Luo et al. (2008), we use a radar reflectivity threshold of 0 dBZ at 10 km altitude as a proxy for potential cloud-top cooling, which may be indicative of convective growth. We calculate the temporal gradient of radar reflectivity at 10 km for each cloud trajectory, identifying the time of maximum increase to mark the cloud development stage. This stage may be associated with a high cloud vertical layer and strong updrafts that support continued vertical growth (e.g., Kikuchi and Suzuki (2019); Chen et al. (2021)).The transition from development to maturity is defined by the time of maximum radar reflectivity increase (Takahashi et al., 2023; Hu et al., 2021)."

- **Fig. 13**: Fig. 13: this fig is interpreted as providing evidence for differences over land and ocean (L332, L333, L338), but to my eye the land/ocean results are almost identical, e.g. with differences in reflectivity being far smaller than the uncertainty of 3dBz (Table 2). Conclusions about land/sea differences appear unwarranted.

  **Author's response**: We agree and change the description of the figure to focus on differences between clouds with a single core and multiple cores instead of land-sea differences (which we agree to be more marginal, see e.g., Table 4 of revised manuscript).

  **Author's changes in the manuscript**: Please see Sect. 4.3.1 for the relationship between life-cycle statistics and core numbers, lines 366-376: "To analyse the cloud life-cycle (as outlined in Section 3.6), we check the point of time when three key events occur in each cloud trajectory: the maximum radar reflectivity gradient at 10

km altitude ("reflectivity gradient"), the maximum cloud area growth("area growth"), and the maximum vertical growth ("vertical growth"). Figure 13 shows the distribution of these indicators grouped by the surface type and number of cores. The average maximum reflectivity gradient ranges from 10 to 16 dBZ. Clouds with 2–3 cores show the highest gradients (14.5–16 dBZ), while the gradient for single-core clouds averages around370 14 dBZ. It decreases with further increasing core count, dropping to around 10 dBZ for clouds with 10 or more cores. Surface type has little impact overall, though values are slightly higher over the ocean for clouds with 1–3 cores (Figure 13, a). In contrast, cloud area growth is slightly higher over land. More important, clouds with multiple cores grow considerably more in area than single-core clouds - ranging from 22 % (single-core) to 52 % (10 and more cores) (Figure 13, b). For the vertical growth, we observe average values between 5–8 km. Single-core clouds tend to grow higher than multi-core clouds, with only minor differences between land and sea (Figure 13, c)."

- **Fig. 14**: Please clarify whether Fig 14 x-axes are cloud lifetime or the time from CI, it is confusing at the moment (e.g. discussion at L340). My best guess is the plot shows a PDF of the times at which x occur, where x is max area growth rate, max number of cores, etc. This is not really lifetime so please reword things. If this is true, how do most clouds have maximum core size at 0 hours? How does max cooling rate most often occur at hour 0 which should by definition have a cooling of 0?
  **Author's response**: The x-axes refer to the time derived after the first detection of the cloud (which would be the CI). We add a revised version of the figure displaying the PDF of the times each statistic (e.g., max cores, max growth) occurs, grouped by the number of cores to show these timing differences be more clearly.
  **Author's changes in the manuscript**: See Sect. 4.3.1, lines 390-415 for a revised description of Figure 15.

- **L343**: "Multiple peaks characterise the distribution for single-core clouds" not supported by plot
  **Author's response**: We agree and revise the description of the figure.
  **Author's changes in the manuscript**: See above, please find a revised description in lines 390-415.

- **L357**: "We observe a morning peak followed by a decrease in the afternoon ..." not supported by plot
  **Author's response**: We agree and revise the description of the figure.
  **Author's changes in the manuscript**: Please see Sect. 4.3.2, lines 416-435 for the description of (now) Figure 16. E.g., lines 416-426: "The reflectivity gradient exhibits short-term fluctuations with noticeable nocturnal peaks around 20:00–21:00 UTC and 00:00–01:00 UTC, followed by decreases. During the day, peaks occur between 09:00–12:00 UTC and around 16:00 UTC, with a negative dip around noon (land) and 18:00 UTC (both land and sea). Over the ocean, the reflectivity gradient is generally higher and shows a slightly weaker variability than over land. Over land, multi-core clouds exhibit stronger gradients than single-core clouds. Distinct land-based negative peaks occur around 03:00–06:00, 08:00, and 11:00–15:00 UTC. Over the ocean, we find a weaker nocturnal peak at 01:00 UTC and a gradual increase between 06:00–20:00 UTC. Overall diurnal variability ranges from 0.5 dBZ (ocean) to 1 dBZ (land), or roughly 8–16 % of the mean gradient range (10–16 dBZ) (Figure 16, a, d)."

- **L360**: "Clouds with a single core show a reflectivity difference of up to 14 dBZ between CI and MAT and a vertical growth of 5.6–7 km. With an increasing number of DCCs, the total cooling and vertical growth of each DCC decreases as prior convective activity already induced a higher average radar reflectivity. In contrast, the anvil growth is on average 15 % higher for clustered systems." unclear where this is shown. Are we still talking about fig 15?
  **Author's response**: We revise the text in the manuscript.
  **Author's changes in the manuscript**: See comment above for description of (now) Figure 16 in Sect. 4.3.2, lines 416-435.

- **Sect. 5.1**: I find this section distracting and not well tied to the results found in the present paper. The only clear link seems to be L394, discussing the seasonality which I suggested above should be removed. It comes across as a literature review rather than a review of the new findings.
  **Author's response**: Thank you for your comment. While we think the section provides additional context to our results, we aim to improve the text to better link our findings with previous studies.
  **Author's changes in the manuscript**: Please see revised Sect. 5.2, lines 482-515 (e.g., lines 482-489: "Compared to previous studies by Takahashi et al. (2023) and Hu et al. (2021), our analysis identifies a significantly higher number of potentially convective cloud tracks. The derived cloud characteristics align well with aircraft

observations (Zipser and LeMone, 1980) and precipitation-based studies (Zipser et al., 2006). Over tropical Africa, our core distribution results are consistent with those derived for geostationary satellite data (Jones et al., 2024) or the CloudSat CPR Deng et al. (2016), both of which found a high prevalence of clouds with one to three cores. Similarly, Pilewskie and L'Ecuyer (2022) reported that one-third to half of convective systems observed globally by the CloudSat CPR contain a single core. For the tropics, however, our results are in closer agreement. In line with these findings, we observe that cloud area generally increases with the number of cores. However, this relationship exhibits substantial variability, especially in multi-core systems (Section 4.3.1)."

- **L405**: "lacks sensitivity to observe ice" → "weak representation of shallow convection" Are you sure this is what you mean?
  **Author's response**: We suggest the CloudSat CPR has a limited sensitivity for thin clouds in high altitudes and shallow convection in low altitudes (radar attenuation due to topography). These limitations are reflected by our ML-based 3D predictions. We revise the description in the manuscript.
  **Author's changes in the manuscript**: Please see Sect. 5.3, lines 516-541 for a revised verison of the limitations, e.g., lines 519-523: "The input data for the ML model are based on observations from the Cloud Profiling Radar (CPR) aboard the CloudSat satellite, which has known limitations in detecting ice clouds due to its tendency to underestimate the height of upper-level outflow (Wang et al., 2014). Additionally, signal attenuation near the surface caused by topography reduces the CPR's sensitivity to shallow convection. As a result, our analysis underrepresents both shallow convective and cirrus cloud types."

- **Section 5.2**: Thank you for including a limitations section (5.2).
  **Author's response**: Thank you.
  **Author's changes in the manuscript**: Sect. 5.2 is now Sect. 5.3

- **L431**: "Isolated convective cells have a higher cooling and more extensive vertical growth than clustered clouds." See the above about cooling. This also appears to confuse vertical growth rate with cloud height, at least in how it reads to me. I don't think cloud height was explicitly considered (?) but cloud top height (Fig 7f) and mean core height (Fig d) are both largest for clouds with the largest number of cores.
  **Author's response**: The sentence refers to findings about the cloud life-cycle derived from Figures 13, 15 and 16. In the revised version of the manuscript, we aim to improve the description of the results.
  **Author's changes in the manuscript**: For the concluding remarks, please see Sect. 6, lines 543-561 (e.g., lines 549-556: "The results suggest that differences based on the number of cores are higher than the surface-type induced variability. Single-core clouds develop and dissipate on shorter timescales. They have a smaller cloud and core area, and lower CTH and core height than multi-core systems. The longer cloud lifetime of multi-core clouds may be associated to a later occurrence of the maximum number of cores and core area. Between single-core and multi-core clouds, we find considerable differences in the cloud life-cycle statistics regarding the changes in the radar reflectivity at 10 km height, the vertical growth, and the area growth of the cloud. While the former two are higher for clouds with a single core, multi-core cloud clusters with a larger cloud area tend to grow more along the horizontal dimension. The more cores we find, the later the maximum number of cores and the maximum core area occur."

- **L 433**: "[Isolated convective cells] ... show weaker convective activity" seems to be inconsistent with the above comment that they have "higher cooling".. maybe these things need to be more specifically defined
  **Author's response**: We agree, we revise the text to describe our findings more clear throughout the manuscript.
  **Author's changes in the manuscript**: See comment above (revised version of concluding remarks in Sect. 6)

- **L204-211, L375-383**: Up to the authors, but in my opinion some sections discussing the advantage of the 3D perspective considered here (**L204-211, L375-383**) seem out of place and repetitive. While true, this discussion belongs in the intro and conclusions.
  **Author's response**: Thank you for your remark, we agree this part is repetitive and remove it from the discussion.
  **Author's changes in the manuscript**: Removed from Sect. 3.4 (Extract cloud and core properties) and Sect. 5 (Discussion). The information may be found in Sect. 1, lines 54-62: "Despite decades of research, knowledge of the 3D structure of convective cores remains limited. In the absence of high- resolution, global 3D observational data, our understanding of the relationship between cores and overall cloud evolution relies heavily on 2D observations and simulations. Active and passive sensors contain important vertical or horizontal information,

but are limited in their spatial and temporal coverage (active) or offer only an approximation of the vertical column (passive) (Masunaga and Luo, 2016; Taylor et al., 2017). To address this gap, we apply a machine learning (ML) framework to reconstruct contiguous 3D radar reflectivity fields from 2D satellite data (Brüning et al., 2024). Our goal is to simultaneously capture the horizontal and vertical evolution of convective clouds and their cores. We use imagery from the Meteosat-11 SEVIRI sensor as input to the ML model, which is trained to reconstruct vertical cross sections based on CloudSat Cloud Profiling Radar (CPR) observations. This approach allows us to extrapolate a continuous 3D cloud field between 2.4 and 24 km altitude."

**4   Technical corrections**

- **L 298**: the sentence " Especially... up to 0.9" seems out of place, also the reference should be fig 10 not fig 9. Also, there are two terms used in this sentence "axes rato" and "eccentricity" and a third in Fig 10's plot label, "Core Roundness". Please standardize

  **Author's response**: Thank you for your comment, we update the figure labels and change "roundness" to "eccentricity" throughout the text.
  **Author's changes in the manuscript**: See Sect. 4.2.1, lines 325-333, for a revised description of (now) Figure 11.

- **Fig 10m,n**: can the area ratio not be clipped at .5? it should go to 1 right?

  **Author's response**: You are right, the scale is 0-1, we add an updated version of the figure in the revised manuscript.
  **Author's changes in the manuscript**: See Figure 11 i–l with the scale between 0-1 for the area ratio.

---

## Author Comment (AC2)

**Final response**
**(Manuscript "EGUSPHERE-2025-374")**
**Reviewer 2**

Sarah Brüning and Holger Tost

May 23, 2025

**1   Introductory remarks**

We would like to thank the reviewer for taking your time to write your constructive and detailed feedback. In the following sections, we address all comments. For every comment, you can find (1) the comment, (2) the author's response (both with lines from initial manuscript), and (3) the author's changes in the revised manuscript. The lines given for **author's changes in the manuscript** refer to the lines in the **revised manuscript**.

Based on all referee comments, we focus the revision on improving the clarity of the written expression for, i.e, the description of our methods and the results. As suggested in the reviews, we aim to include a more detailed acknowledge about the subjectivity of our object-based detection approach and derived cloud life-cycle. While the narrative and main points of the manuscript did not change, we aim to focus on directly describing observed results, in particular regarding the connection between convective cores and cloud properties. Based on both referee comments, we agree the term "seasonal" may be misleading for the depicted period of only six months. Instead, we describe "monthly" changes, which we observed during the period. Moreover, we changed the abbreviation "DCC" to "cores" to align with our manuscript for Part 2 of the study ("EGUSPHERE-2025-376").

We changed the figure labels in the following way:

- 1 -> 1
- 2 -> 3
- 3 -> 2
- 4 -> 4
- 5 -> 5
- 6 -> 6
- 7 -> 7
- 8 -> 8
- 9 -> 10
- 10 -> 11
- 11 -> 12
- 12 -> Removed
- 13 -> 13
- 14 -> 15

- 15 -> 16

- 16 -> 17

- New: 9 (Distribution of cloud tracks grouped by the surface type: month, daytime of first detection)

- New: 14 (Correlation matrix for cloud properties, core properties, and life-cycle statistics for all cloud tracks, clouds with a single core, and clouds with multiple cores)

**2 General comments**

- **Comment**: Introduction: It is best practice to choose the earliest references that support statements. The authors appear to have chosen very recent papers, which are still valid to keep, but it is recommended to find more sources that are older. In particular, I am referring to L18 [Roca et al., 2010], L19 [Chen et al., 2021], L20 [Kukulies et al., 2021; Haberlie and Ashley, 2018], L22 [Prein et al., 2024], L33 [Fiolleau and Roca, 2013], L47 [Haberlie and Ashley, 2018], L132 [Prein et al., 2024]. A good reference for background on cloud feedbacks and climate sensitivity is Sherwood et al., (2020). Leary and Houze, (1979) goes into detail on the convective cloud lifecycle.
  **Author's response**: Thank your for your comment, we revise the section to include more background references, like the ones you suggested.
  **Author's changes in the manuscript**: We revised the references, i.e., in Sect. 1. Examples may be found in lines 16–23: "Convective clouds play a vital role in the hydrological cycle of the Earth through their radiative forcing and feedback mechanisms (Wielicki et al., 1995). Despite growing evidence for the connection between clouds and climate warming, they remain one of the greatest sources of uncertainty in climate sensitivity assessments (e.g., Bony et al. (2015); Sherwood et al. (2020)). Additionally, convective clouds are key drivers of severe weather, particularly large-scale systems like mesoscale convective systems (MCSs), which are linked to extreme events such as hailstorms, damaging winds, and intense rainfall (e.g., Houze and Hobbs (1982); Leary and Houze (1980); Maddox (1980)). Because of their societal and environmental impacts, accurately representing convective clouds remains of particular interest", lines 26–29: "While cores drive intense precipitation, the stratiform anvil and cirrus canopy generally produce lighter rain (e.g., Houze Jr. (1989); Hartmann et al. (1984)). Core sizes typically range from 10 to 100 km, with lifespans of 1–3 hours, whereas anvils can persist for up to 10–20 hours. The idealised MCS life-cycle includes three stages: development, maturity, and dissipation (Futyan and Genio, 2007).", or lines 39-42: "For instance, passive and active sensors provide valuable insights into the temporal evolution of clouds. Passive sensors, especially those measuring infrared (IR) radiation, help identify cloud-top features (Mecikalski et al., 2010). However, they lack vertical resolution, making it difficult to distinguish between deep convection, stratiform clouds, and cirrus (Liu and Zipser, 2008)."

- **Comment**: Data and Method: It would be beneficial if you just focused on describing all the data used in the Data section and then had a separate section in the Methods that described the ML algorithm that was published previously. It is unclear if the details in L106-128 are summarizing the ML algorithm described in your previous paper. Please also clarify the following: - It is discussed how CloudSat, but not how Seviri, data are being used. Are both CloudSat and Seviri data used to train the model? - Could you briefly describe how you computed the average error of the model? - How do you test the model to evaluate its performance, if at all? - It should be clearly stated that the methodology outlined in the "Methods" section (as it currently stands) are applied to the predicted radar reflectivities outputted from the ML model. - Descriptions of both Sections 3.2 (Detect convective core regions) and 3.3 (Extract cloud and core properties) should be clarified greatly. Details for how they should be clarified are under Specific Comments.
  **Author's response**: Thank you for your suggestions. We updated Sects. 2 (Data) and 3 (Methods) accordingly. In the Sect. 2, you may find information on the satellite data sources. In the Sect. 3, we add a detailed description of the ML algorithm and its validation in Sect. 3.1 (which hopefully answers your questions as above). Moreover, we revised the description of the detection and tracking approach (Sects. 3.2 - 3.3).
  **Author's changes in the manuscript**: Please see Section 2 (Data from MSG SEVIRI & CloudSat CPR) and Section 3 (Method). For the training and evaluation of the ML model, you may find information in lines 109–118: "The model is trained on nine months of data and validated on a separate three-month period. The Res-

UNet is trained to reconstruct CloudSat-like 3D reflectivity volumes with a horizontal size of $100 \times 100$ pixels and a vertical size of 90 levels. The predicted radar reflectivity values range from –25 to 20 dBZ and retain the 15-minute temporal resolution of the original SEVIRI input. We use an L1 loss function (mean absolute error) during training to evaluate the model's performance. Notably, direct validation is possible only for the diagonal cross-section, which accounts for about 10 % of each training sample. For the three-month test period, the modified daylight-independent model achieves a root mean square error (RMSE) of 2.99 dBZ — an improvement over the original model (Table 2). This level of accuracy is comparable to the 5 dBZ precision reported for other CloudSat products (Tomkins et al., 2024)." or lines 124–128: "To further assess model performance, we compute cloud top heights (CTH) from the predicted radar reflectivity and compare them to CTH values from the CMSAF CLAAS-V002E1 dataset (Finkensieper et al., 2020). The comparison reveals that the model captures realistic spatial patterns of CTH in both tropical and mid-latitude regions. However, model accuracy tends to decline with increasing distance from the MSG SEVIRI nadir.". In Sections 3.2 (Detect convective core regions), lines 193-215, and 3.3 (Extract cloud and core properties), lines 217-227, you may find a revised description of the detection and tracking framework.

- **Comment**: Results: The descriptions and wording in the analysis are somewhat vague and need to be clarified. For example, it is not understood how "convective activity" is defined, and this phrase is used several times throughout the text. More clarifications to the results can be found under Specific Comments.
  **Author's response**: We revised the manuscript with a focus on improving the written expression and clarity of observed results.
  **Author's changes in the manuscript**: Replaced the term "convective activity" with a more direct description of the results, i.e., when talking about the number of cores associated to a cloud or the cloud life-cycle. Examples may be found, e.g., in Sect. 4.1, lines 271-273: "Figures 4 (b)–(e) illustrate the development and dissipation of cores, often lasting only a short time. Some clusters contained multiple cores, potentially indicating mesoscale convective systems (MCSs) (Takahashi et al., 2017)."

- **Comment**: Seasonal cycle: Are there any potential limitations in your statistical analysis of the seasonal cycle given that you are only using one year of data? Can you comment on if interannual variability (e.g. ENSO) might have impacted such results?
  **Author's response**: The period analysed in this study may provide insights on the 3D cloud properties between March and August 2019; however we may not derive a climatology which would indeed require the processing of at preferably decades of data. Unfortunately, processing data for a climatoligal period is out of scope for our study. Hence, our results may be affected by specific events and the mentioned interannual variability occurring during the chosen period - we will acknowledge potential limitations on the depicted period in the revised manuscript more directly. Here, we did not specifically analyse the impact of, e.g., ENSO, but we agree these might be important factors, in particular when analysing a longer (climatological) time series.
  **Author's changes in the manuscript**: We change the terminology from "seasonal" changes (since we only have a sample size of n=1 for each season) to "monthly changes" along the period (see, e.g., Sect 4.2.2 "Monthly variability of convective properties"). Limitations are, e.g., added to the discussion in Sect. 5.2, lines 502-506: "Seasonal variability in tropical convection has been highlighted in past studies. For example, multi-core systems often persist overnight during the onset of the West African Monsoon (Futyan and Genio, 2007). During this period, convection may frequently initiate over high terrain and propagate downslope at night under katabatic flow (Nicholson, 2018). While our results may be influenced by interannual variability, as the dataset spans only one year and does not capture a full annual cycle, we may observe temporal changes in cloud and core properties between March and August. "

**3 Specific comments**

In the next section, we shortly address your specific comments. Changes will be added in the revised manuscript.

- **L8, L360** : What is meant by "absolute" cooling?
  **Author's response**: Here, the original manuscript refers to changes of the radar reflectivity at 10 km height between different time steps. The term "absolute" cooling refers to the maximum difference between the radar reflectivity at 10 km height at first time step of detection and the depicted time step. We revise the terminology

since radar reflectivities may not replace temperatures; we change the text to be more accurate.

**Author's changes in the manuscript**: See the description of the cloud life-cycle in Sect. 3.6, lines 240-246: "Unlike methods that assess cloud stages using a cooling induced by temperature changes, the ML-based radar reflectivity does not provide information on temperatures. As an alternative, we approximate the life-cycle using temporal changes in radar reflectivity at 10 km height and the resulting vertical and horizontal cloud characteristics. For estimating the vertical growth of the cloud, we compute the difference between CTH and CBH (i.e., to display the height of the cloud layer) for every point in time. For the horizontal growth of the cloud, we calculate changes of the cloud area derived as proportional differences to the cloud area at the first timestep of detection."

- **L42-43**: Bacmeister and Stephens, (2011) is not a good reference that investigates "the temporal evolution of MCSs. A study using geostationary satellite observations or other such measurements that capture the temporal evolution is preferred.

  **Author's response**: We revise the references used in this section.

  **Author's changes in the manuscript**: Please see Sect. 1, lines 39-41: "For instance, passive and active sensors provide valuable insights into the temporal evolution of clouds. Passive sensors, especially those measuring infrared (IR) radiation, help identify cloud-top features (Mecikalski et al., 2010)."

- **L67**: Why is it that single convective cells are typically tracked using data from active remote sensing sensors? And specifically which sensors? Are different measurements typically used to track single-cell systems compared to MCSs?

  **Author's response**: Analysing the life-cycle of single-core with a short lifetime clouds requires data with a high temporal resolution. Moreover, to assess their vertical growth, active sensors are beneficial. However, we revise the text in this section to be more clear.

  **Author's changes in the manuscript**: We removed the sentences from Sect. 1 in the revised manuscript.

- **L113**: Could you please clarify what a "spatio-temporal matching scheme" is?

  **Author's response**: The term "matching scheme" is employed to describe how we receive training samples for the machine learning model. Since CloudSat is a polar-orbiting satellite, its spatial and temporal resolution differ from the geostationary MSG SEVIRI imagery. We propose a framework to extract the training samples (which consist of 128 × 128 pixel patches and eight channels between the near- to thermal infrared of SEVIRI imagery) by searching for locations where we detect a CloudSat overpass (in our domain between 60°N-60°S and 60°W-60°W). In the revised manuscript, we add a more detailed description of the approach (Sect. 3.1).

  **Author's changes in the manuscript**: Please see Sect. 3.1 for a revised description of the machine-learning method, e.g, lines 107-109: "Training data consist of 128 × 128 pixel patches of SEVIRI imagery that are spatially and temporally aligned with CloudSat overpasses. Each training sample includes a diagonal CPR cross-section. Due to the spatial resolution mismatch between MSG SEVIRI and CloudSat, we downsample the SEVIRI data to match the CPR's horizontal resolution."

- **L149**: —"weighted mean": Weighted by what?

  **Author's response**: We compute the centroid of a potential cloud using a weighted center-of-mass approach, using the geometric center for the detection threshold > -15 dBZ and the positions of the centroid.

  **Author's changes in the manuscript**: Sect. 3.2.1, lines 153-156: "The detection process begins by applying a Gaussian filter with a sigma value of 0.5 to smooth the input data and reduce noise (Kukulies et al., 2021). We then compute the centroid of each potential cloud using a weighted center-of-mass approach. Here, each point's weight is defined by its reflectivity value above the –15 dBZ threshold (Heikenfeld et al., 2019). These centroid positions are each assigned a unique identifier, which is maintained throughout the subsequent tracking and segmentation steps."

- **L154-155**: It is unclear what "The value of each pixel is decreased towards its local minimum using a threshold of -15 dBZ" means.

  **Author's response**: For the watershed segmentation, we set markers at the previously identified centroid positions in a binary 3D image filled otherwise with zeros. Then, the algorithm fills the volume based on the original input field (radar reflectivities) starting from these markers until reaching the defined threshold (which is -15 dBZ in our case). The result is an output image of the same shape as the original input with zeros at all grid points in which there is no cloud or updraft and the integer number of the associated feature at all grid

points belonging to that specific cloud or updraft. We revise the description to be more clear.

**Author's changes in the manuscript**: Sect. 3.2.1, description of the segmentation step in lines 157-164: "Next, we apply a 3D watershed segmentation algorithm to delineate the spatial extent of individual cloud structures associated with each centroid. In this approach, the 3D radar reflectivity field is interpreted as a topographic surface, where higher reflectivity values represent peaks and surrounding areas are segmented like catchment basins divided by ridges (Meyer, 1994). We initialize the algorithm by placing markers at the detected centroids in a binary 3D volume, where all other grid points are set to zero. From each marker, the algorithm expands through the volume, assigning reflectivity-based pixels to the corresponding cloud until the threshold of –15 dBZ is reached. The result is a labeled 3D cloud mask, where each voxel is either zero (indicating no cloud) or an integer label corresponding to a specific cloud object (Fiolleau and Roca, 2013)."

- **L163**: Why do you need to define a system as "elongated"? Is it under the assumption that a system should be near circular, and that it is elongated due to the merging of multiple systems?

  **Author's response**: We chose to separate these objects to account for uncertainties connected to the predicted 3D cloud field. Here, we aim to cut off cloud systems that are only shallow connected by few pixels to avoid the merging of (possibly) distinct systems during the labeling process (which we aim to identify by a less circular shape and a distinct dip in the width of the cloud area).

  **Author's changes in the manuscript**: Please see Sect. 3.2.2, lines 167-179 for a revised description.

- **Section 3.2**: Have you validated your core detection method? It would be interesting if you apply your core detection method to physical 2B-GEOPROF radar reflectivities and then see how it compares to cores detected using the Convstratflag from 2C-PRECIP-COLUMN (Pilewskie and L'Ecuyer, 2022; data description here) for a month of data, if possible.

  **Author's response**: Thank you for your suggestions. In our study, we did not explicitly validate the core detection method. However, derived cloud and core statistics (e.g., for the distribution of core numbers) align well with studies in similar tropical regions (please see Sect. 5.2, lines 485-492, for a discussion of results). Based on this analysis, we estimate the approach to deliver meaningful results. While the predictions of our machine learning model are based on CloudSat CPR 2B-GEOPROF data, we think the results may be comparable to studies which applied similar methods to identify cores along the cloud vertical column directly to CloudSat CPR 2B-GEOPROF radar reflectivities (e.g., Luo et al. 2008, Bacmeister and Stephens 2011, or Igel et al. 2014; please see manuscript references). While our predicted cloud field may incorporate the ML model uncertainties (see Sect. 3), the 3D data allow us to showcase one possibility to detect contiguous core regions. Hence, we may combine approaches applied to the cross sections of 2B-GEOPROF data with object-based segmentation techniques typically applied to e.g., geostationary satellite data. For instance, we do not think our approach, as it is adapted to 3D input data would be applicable to the 2B-GEOPROF radar reflectivities (which provide 2D data). Nevertheless, we think it would be interesting to compare different types of core detection methods (e.g., the approach using the Convstratflag from 2C-PRECIP-COLUMN as shown in Pilewskie and L'Ecuyer, 2022) in detail - which we rather see as an intriguing opportunity for further studies to improve robustness of convective cloud analysis. We would like to thank the reviewer for the suggested reference and will acknowledge the variability of core detection methods in our discussion.

  **Author's changes in the manuscript**: Please see Sect. 3.3 for a revised description of the core detection method, lines 196-202: "To detect convective cores, we use the previously generated labelled 3D cloud mask (Section 3.2.1), derived from the ML–based radar reflectivity data. There are different approaches to identify convective cores from radar reflectivities. These methods may comprise the detection of convective precipitation, which may be associated to core regions in hydrometeors (Haynes et al., 2009; Pilewskie and L'Ecuyer, 2022) or the analysis of the radar reflectivity employing fixed thresholds along the vertical column (Luo et al., 2008; Bacmeister and Stephens, 2011; Igel et al., 2014). In this study, we focus on combining the latter with an object-based detection algorithm to identify centroids of convective cores in the predicted 3D radar reflectivity field".

- **L194**: Why do you expand the threshold?

  **Author's response**: When detecting pixels in the vertical column that pass the criteria of > 0 dBZ, which may be used to display the location of a possible core region, we lower the threshold to avoid the segmentation of vertical holes in the core.

  **Author's changes in the manuscript**: Sect. 3.3, lines 205-208: "Specifically, we calculate the mean radar

reflectivity for each vertical cloud column, then determine the height of this core layer by counting the number of pixels with reflectivity values higher than 0 dBZ located at more than 5 km height. We aim to fill isolated gaps for otherwise vertical continuous cores by expanding the threshold from 0 dBZ to –5 dBZ in columns that contain at least one pixel higher than 0 dBZ (Igel et al., 2014; Luo et al., 2008)."

- **L194-195**: What is "the first criterion"?
  **Author's response**: Refers to the detection of pixels < 0 dBZ in each cloud vertical column. We revise the description to be more clear.
  **Author's changes in the manuscript**: Sect. 3.3, lines 202-208: "We begin by smoothing the radar reflectivity data associated with each cloud label using a 3×3×3 median filter. Core centroids are identified by locating local maxima in a combined metric that incorporates both smoothed radar reflectivity and the vertical extent of a contiguous potential core layer. Specifically, we calculate the mean radar reflectivity for each vertical cloud column, then determine the height of this core layer by counting the number of pixels with reflectivity values higher than 0 dBZ located at more than 5 km height. We aim to fill isolated gaps for otherwise vertical continuous cores by expanding the threshold from 0 dBZ to –5 dBZ in columns that contain at least one pixel higher than 0 dBZ (Igel et al., 2014; Luo et al., 2008). We apply a minimum vertical extent of 5 km for a column to be considered part of a core; otherwise, its value is set to zero."

- **L195**: In the case of significant attenuation within a convective core region, the CBH may reach up to 5 km. Have you looked into this? How do you account for attenuation in core profiles?
  **Author's response**: Thank you for your remark. We did not specifically look into this, but we did try to account for attenuation of the whole cloud by omitting CloudSat CPR height levels < 2.4 km. If we detect cores in height levels < 5 km CBH, we do not account for those in the current approach. However, revising the method to include cases where the CBH is higher due to attenuation may provide beneficial.
  **Author's changes in the manuscript**: For the updated core detection approach, see Sect. 3.3, lines 193-215.

- **L195-196**: —"if the vertical profile shows no convective pixels for more than 50 % of the CTH": Please clarify what is meant by this statement. Firstly, how are the convective pixels defined? There is a definition for DCCs in L190-192, but it is not sufficient as it does not give any thresholds to the maximum "column-wise aggregated radar reflectivity" nor the "difference between the CTH and CBH" used to isolate DCCs. Secondly, what is meant by "more than 50% of the CTH"?
  **Author's response**: Cores are identified by locating local maxima in a combined metric that incorporates both smoothed radar reflectivity and the vertical extent of a contiguous potential core layer. Here, we define a pixel as potentially "convective" (which may induce the location of a core) by a threshold of > 0 dBZ (adapted from Luo et al., 2008 and Chen et al. 2021). Specifically, we calculate the mean radar reflectivity for each vertical cloud column based on the labelled cloud mask. We revise the description to be more clear.
  **Author's changes in the manuscript**: Please see comment above, changed description in Sect. 3.3, lines 193-215.

- **L196-197**: It is not understood what is meant by "we add the mean radar reflectivity of the vertical profile to the derived vertical depth of the column". Please clarify.
  **Author's response**: See above, for assessing the vertical depth of a potential core in each vertical column of the labelled cloud mask, we count the number of pixels with reflectivity values > 0 dBZ. We add the mean radar reflectivity for each vertical cloud column. Within the resulting layer, we search for local maxima to identify the possible centroid of core regions. We revise the description to be more clear.
  **Author's changes in the manuscript**: Please see comment above for the changed description in Sect. 3.3, lines 193-215.

- **L212** : Please specify what is meant by "We employ the radar reflectivity at a fixed altitude of 10 km as a measure of convective activity". What defines "high" versus "low" convective activity?
  **Author's response**: Since the predicted ML-based radar reflectivity data does not provide direct information on the cloud temperature, which are typically used to investigate the cloud life-cycle, we choose to analyse changes in the radar reflectivity at 10 km height. A positive gradient between different timesteps - and the maximum of this gradient - are used as a measure to determine growing convective cells, possibly within the development stage. We revise the description of the extracted cloud and core properties, and the cloud life-cycle, to be more clear.

**Author's changes in the manuscript**: Changed description of the cloud life-cycle in Sect. 3.6, e.g., lines 240-246: "Unlike methods that assess cloud stages using a cooling induced by temperature changes, the ML-based radar reflectivity does not provide information on temperatures. As an alternative, we approximate the life-cycle using temporal changes in radar reflectivity at 10 km height and the resulting vertical and horizontal cloud characteristics. For estimating the vertical growth of the cloud, we compute the difference between CTH and CBH (i.e., to display the height of the cloud layer) for every point in time. For the horizontal growth of the cloud, we calculate changes of the cloud area derived as proportional differences to the cloud area at the first timestep of detection."

- **L219, L232-233** : It is not understood if there is a general CTH threshold applied to the clouds for isolating convective systems. As it currently reads, it seems clouds need to have a CTH of at least 10 km; however, clouds during the convective initiation (CI) often have cloud tops below 10 km. For this reason, I am not convinced that you are capturing systems during the CI stage.
  **Author's response**: The threshold of > 10 km is used after the linking step to identify potentially convective cloud tracks in the dataset. We do not require clouds to have a CTH > 10 km at every time step during this trajectory, but if the cloud never reaches a CTH > 10 km, we discard the track ID. However, clouds may have a lower CTH, e.g. during initiation or shortly before dissipating.
  **Author's changes in the manuscript**: Please see Sect. 3.5, lines 229-235: "We filter the cloud trajectories to exclude possibly non-convective tracks from the analysis (Figure 2). For that purpose, we require the cloud tracks to have at least one core and a radar reflectivity of higher than 0 dBZ at 10 km height for at least 15 min along the trajectory. Additionally, we apply a minimum CTH of 10 km and a maximum CBH of lower than 5 km for the cloud during at least one time step (Igel et al., 2014; Luo et al., 2008). While we do not require the convective clouds to have a CTH higher than 10 km at every time step during their trajectory, we discard the trajectories that never reach the CTH threshold. The criteria may help to identify convective clouds with an evolved cloud base and vertical height that may be typically associated to tropical convection (Li et al., 2021; Takahashi et al., 2023)."

- **L220**: Please describe the assumptions made for using the "difference between the radar reflectivity at 10 km height at CI and the current time step" to approximate cooling? Also, what is meant by cooling, is it at the top of the atmosphere, atmospheric cooling, or with regards to the surface?
  **Author's response**: We agree the term "cooling" may be misleading since we do not assess any cloud temperatures in our study. Instead, we approximate the stage of developing convective cells by calculating changes in the radar reflectivity at 10 km height along the temporal axis and searching for the maximum increase (or gradient) after the first time step of detection. We rename "cooling" to "reflectivity gradient" to describe this step more clearly. Moreover, we add a more detailed explanation of the statistics used to assess the cloud life-cycle in Sect. 3.6.
  **Author's changes in the manuscript**: See comment above, the description of the cloud life-cycle can be found in Sect. 3.6, lines 237-263.

- **L221-222** : These sentences would be better understood if they were merged. Also, what is meant by "difference between the CBH and CTH at CI for each time step, compared to CI"?
  **Author's response**: Thank you for your suggestion. We will change the text to describe the approximation of cloud life-cycle statistics.
  **Author's changes in the manuscript**: Please see Sect. 3.6, e.g., line 244: "For estimating the vertical growth of the cloud, we compute the difference between CTH and CBH (i.e., to display the height of the cloud layer) for every point in time." and

- **L223**—"difference of the radar reflectivity": difference with respect to what exactly?
  **Author's response**: Refers to changes of the radar reflectivity at 10 km height (hereafter, reflectivity gradient) derived at the first time step of the trajectory (t0) and each following time step (t1 - tn).
  **Author's changes in the manuscript**: See above for description of Sect. 3.6, lines 246-253 (Development stage): "Development stage: Building on the approach by Luo et al. (2008), we use a radar reflectivity threshold of 0 dBZ at 10 km altitude as a proxy for potential cloud-top cooling, which may be indicative of convective growth. We calculate the temporal gradient of radar reflectivity at 10 km for each cloud trajectory, identifying the time of maximum increase to mark the cloud development stage. This stage may be associated with a high cloud vertical layer and strong updrafts that support continued vertical growth (e.g., Kikuchi and Suzuki (2019);

Chen et al. (2021)).The transition from development to maturity is defined by the time of maximum radar reflectivity increase (Takahashi et al., 2023; Hu et al., 2021)."

- **L238**: If I am interpreting correctly, are the convective clouds that are identified actually "real", as in you would be identifying these systems in SEVIRI if you were to be just tracking them in 2D? It's just the 3D cloud fields that are predicted, hence not "real"?

  **Author's response**: You are right, our study is based on the ML predictions of a 3D cloud field. Hence, the detected clouds are not "real". While they are based on information observed by MSG SEVIRI and the CloudSat CPR which we extrapolate in space and time, the clouds are not "real" in the sense of being actually observed. We evaluated the ML model to provide a reasonable extrapolation. Nevertheless, the data underlie sensor-specific limitations (CloudSat CPR attenuation, resolution of SEVIRI data) and uncertainties connected to the ML algorithm (RMSE focuses on mean, blurs out edges). We revise the text describe the data more detailed.

  **Author's changes in the manuscript**: See, e.g., Sect. 3.1, lines 141-145: "The framework, while enabling detailed analysis of convective cores, has limitations. The predicted 3D cloud fields represent model-based approximations rather than direct observations, reflecting patterns learned by the ML model. Additionally, using fixed thresholds in the object-based detection may oversimplify complex structures associated to clouds in the atmosphere. Nonetheless, we may employ the data to enable a large-scale, high-resolution tracking of convective systems over the tropical Atlantic and continental Africa."

- **L240**: What is considered "high reflectivity"? 0 dBZ at cloud top is not necessarily a "high" radar reflectivity.

  **Author's response**: We employ a threshold of -15 dBZ to differentiate background noise from actual signals from hydrometeors, thus we consider values > 0 dBZ to be more often connected to convective clouds (and the value higher, in relative terms).

  **Author's changes in the manuscript**: Revised description in Sect. 4.1, Lines 268f.: "While regions over Morocco and Mauritania showed radar reflectivity > 0 dBZ, no vertically continuous convective systems were identified there."

- **L250-252**: From the statistics stated in the text, it does not seem intuitive that only 10% of the population contain 2-10 cores. What are the exact counts in each bin? When studying the histogram in Figure 6, it looks like single-cell systems make up more like 65% of the population, and systems with ten or more DCCs is only 3%, meaning that there is a larger population containing 2-10 cores than what is being inferred in the text. How do these statistics compare to other studies that have assessed the statistics on the number of cores in systems?

  **Author's response**: We check the reported values and revise the figure to contain proportions rather than frequencies. Compared to other studies in the tropics, we receive similar results regarding the the distribution of the core numbers. We will add this to the text in the revised manuscript.

  **Author's changes in the manuscript**: Please see Sect. 4.1, lines 277-283: "Single-core clouds make up roughly 65 % of all trajectories (Figure 6, a), with the frequency decreasing as core count increases. Only about 5 % of clouds have 10 or more cores. Most clouds (80 %) have lifespans between 0–6 hours (Figure 6, b). Surface type distribution reveals that 65 % of clouds form over the ocean and 35 % over land—about a 10 % shift toward ocean compared to land-sea coverage (Figure 6, c). Among single-core clouds, 70 % occur over the ocean, while for multi-core clouds, the figure is 75 %. This imbalance — 249,484 oceanic clouds vs. 88,658 continental — may reflect differences in tropical landmass distribution and the eastward propagation of convective systems. Oceans may also offer more favorable conditions for multi-core development (Cui et al., 2021)." and Sect. 5.2, lines 484-489: "The derived cloud characteristics align well with aircraft observations (Zipser and LeMone, 1980) and precipitation-based studies (Zipser et al., 2006). Over tropical Africa, our core distribution results are consistent with those derived for geostationary satellite data (Jones et al., 2024) or the CloudSat CPR Deng et al. (2016), both of which found a high prevalence of clouds with one to three cores. Similarly, Pilewskie and L'Ecuyer (2022) reported that one-third to half of convective systems observed globally by the CloudSat CPR contain a single core. For the tropics, however, our results are in closer agreement. In line with these findings, we observe that cloud area generally increases with the number of cores. However, this relationship exhibits substantial variability, especially in multi-core systems (Section 4.3.1)"

- **Figure 5**: It would be easier to distinguish cores from the rest of the cloud if there was a larger contrast between the colors. Perhaps make the cores a deeper red.

  **Author's response**: Thanks for the suggestion, we add an updated figure in the revised manuscript with lighter

yellow color for the cloud and darker red color for the cores.
**Author's changes in the manuscript**: See Sect. 4.1, Figure 5 (b).

- **Figure 6** —"surface type derived from a land-sea mask compared to the location at CI": I don't understand this
distinction.
**Author's response**: Here, we compare the actual land-sea proportions in the domain from a binary mask vs. the
land-sea proportions derived from the locations of the cloud trajectories. We revise the description to be more
clear.
**Author's changes in the manuscript**: Sect. 4.1, Figure 6 (c) description: "(c) the surface type derived from
a land-sea mask compared to the modal locations of detected clouds with a single core or multiple cores"; and
Sect. 3.4, line 224: "We also record the cloud's travel distance and assign a surface type using a binary land-sea
mask and the modal value for the locations of the cloud trajectory within this land-sea mask."

- **L268**: —"0.5 dBZ higher over the ocean than over land": How might this tie into the notion of differences in
intensity over land versus ocean? Does this suggest oceanic convection is more intense of land, which differs
from our general understanding of tropical convection?
**Author's response**: We found that averages over land and sea may contradict the general understanding of
convective properties implying an overall slightly more intense convection over oceans. However, the differences
are small and the data is skewed towards clouds located over the ocean - the uncertainties of the skewed
distribution and short period may affect the analysis.
**Author's changes in the manuscript**: Sect. 4.1, lines 284-292: "We assess how the 3D cloud properties
described in Table 3 may vary with core count and surface type. Our findings show that single-core clouds
have shorter lifetimes and travel distances than multi-core systems (Figure 7, a–b). Eccentricity exhibits a weak
variation across all groups, mostly ranging between 0.6–0.7 (Figure 7, c). Cloud area increases significantly
with core count, especially for clouds with 10 and more cores (Figure 7, d). CTH is 10–20 % greater over
land, yet radar reflectivity at 10 km height and cloud area are slightly higher over the ocean (Figure 7, d–f).
CTH increases from 15.5 km for single-core clouds to 17.25 km for multi-core ones (Figure 7, f). Land–sea
differences are more pronounced for single-core clouds. Despite expectations based on previous tropical studies
(Deng et al., 2016; Takahashi et al., 2017), oceanic clouds often show stronger reflectivity and larger areas —
though overall surface-related differences remain small. The lower number of land-based clouds may exaggerate
statistical noise."

- **L272-273**: The sentence beginning with "The core lifetime…" is somewhat challenging to interpret. Please
clarify.
**Author's response**: We rewrite the section to be more clear.
**Author's changes in the manuscript**: Sect. 4.1, lines 293-294: "The analysis of core properties (Figure 8)
shows average core lifetimes of 0.3–0.4 hours for single-core clouds, increasing to about 0.8 hours for clouds
with more than 10 cores (Figure 8, a). "

- **L276-277**: —"especially for convective clouds over land": It appears that both regions show this, not just land.
**Author's response**: We agree and change the description of the figure in the revised manuscript.
**Author's changes in the manuscript**: Lines 295-297: "Core area is slightly larger for single-core clouds than
for those with 2–9 cores but increases considerably for clouds with 10 and more cores. For single-core clouds,
we detect a larger core area over the ocean, while cores for multi- core clouds are larger over land (Figure 8, c)."

- **L286-287** : Have other studies shown that continental clouds have a larger anvil area and lower reflectivity
compared to clouds over the ocean for the region and time period you are studying?
**Author's response**: We checked the figure and revised its; in fact, the area and reflectivity are higher over the
ocean in the revised version. While our findings for the land-sea differences are contradictory to previous results
in the tropics, differences are small and may result from the uncertainties connected to the approach (Sect. 5.3)
or the imbalance of cloud tracks over land and sea (Figure 9).
**Author's changes in the manuscript**: Sect. 4.2.1, lines 319-324: "Despite similar diurnal patterns for the
cloud lifetime and radar reflectivity, multi-core clouds consistently exhibit higher mean values than single-core
clouds. These differences may reflect environmental contrasts between land and ocean. As suggested by Cui et
al. (2021), local circulations over land in the tropics often trigger afternoon convection, producing the observed

peaks in Figure 10 (f), (h), and (j). In contrast, more constant ocean temperatures may suppress strong diurnal variations (Figure 10, a, c, g)."

- **L287-288**: Please clarify this explanation—differences in cloud properties would cause a local thermal instability—is that how it is to be read?

  **Author's response**: Here, we want to point out that a local thermal instability may affect cloud properties - we revise the text and clarify the description.

  **Author's changes in the manuscript**: See Sect. 4.2.1, lines 321-324: "As suggested by Cui et al. (2021), local circulations over land in the tropics often trigger afternoon convection, producing the observed peaks in Figure 10 (f), (h), and (j). In contrast, more constant ocean temperatures may suppress strong diurnal variations (Figure 10, a, c, g)."

- **L291-294** Please include references here, and are you describing the diurnal cycle for this specific region?

  **Author's response**: We revise the description of the figure and add an evaluation of the results in the discussion.

  **Author's changes in the manuscript**: See Sect. 4.2.1, lines 314-337, for the diurnal cycle of tracked cloud properties and a discussion in Sect. 5.2, e.g., lines 490-497: "Our results show that convective cloud properties over land typically peak during the day, while over the ocean, we observe two peaks during daytime and at night (Sect. 4.2.1). These findings align with the diurnal cycle of tropical convection (Vondou, 2012; Takahashi et al., 2023). A nocturnal enhancement over the ocean may be linked to the diurnal cycle of free-tropospheric humidity, which peaks overnight and supports convection (Wall et al., 2020). After sunrise, solar heating may stabilize the atmosphere. A weakening land breeze may lead to the dissipation of night-time clusters (Houze Jr., 2004). While these diurnal patterns may be reflected by the cloud properties (Sections 4.2.1), differences in the daytime of the first detection for the cloud tracks appear weaker (Figure 9). Throughout the day, we observe several peaks for the reflectivity gradient, followed by phases of vertical and horizontal cloud growth (Section 4.3.2)."

- **L298**: Figure 9 a-d does not show the eccentricity of clouds?

  **Author's response**: You are right, we revise the description.

  **Author's changes in the manuscript**: Sect. 4.2.1, lines 315-324: "Figure 10 illustrates these variations in cloud lifetime (a–d), cloud area (e–h), and radar reflectivity at 10 km height (i–l). Over land, single-core clouds show an afternoon peak (12:00–16:00 UTC) in both radar reflectivity and cloud area, while cloud lifetime displays two peaks: one at night and one in the morning (Figure 10, b, d). Over the ocean, the diurnal cycle is weaker or less distinct. Cloud lifetime lacks a clear diurnal peak (Figure 10, a, c), whereas cloud area and reflectivity show nocturnal and daytime peaks (Figure 10, e, i, k). Despite similar diurnal patterns for the cloud lifetime and radar reflectivity, multi-core clouds consistently exhibit higher mean values than single-core clouds. These differences may reflect environmental contrasts between land and ocean. As suggested by Cui et al. (2021), local circulations over land in the tropics often trigger afternoon convection, producing the observed peaks in Figure 10 (f), (h), and (j). In contrast, more constant ocean temperatures may suppress strong diurnal variations (Figure 10, a, c, g)."

- **L304-305**: Second maximum in what, exactly? Where is the first maximum? Also, please use a more scientific word than "powerful" in "afternoon peak is consistently more powerful". Also, what afternoon peak are you referring to?

  **Author's response**: Here, we refer to the diurnal cycle of the core area and core lifetime over land and over the ocean. We revise the description of the figure to be more clear.

  **Author's changes in the manuscript**: Please see Sect. 4.2.1, lines 325-333: "The diurnal patterns of core properties (Figure 11) largely mirror those of the cloud properties. Over land, core area peaks between 12:00–18:00 UTC for both single- and multi-core clouds. Over the ocean, single-core clouds show two peaks between 00:00–06:00 and 14:00–20:00 UTC (Figure 11, a–d). The core lifetime follows a similar pattern for single-core clouds. For multi-core clouds, cores over land show two peaks, while oceanic cores point out no clear diurnal variation for multi-core clouds (Figure 11, e–h). For single-core clouds, peaks of the core lifetime resemble the core area (Figure 11, a, e). The distribution of the core height follows those of the core area (Figure 11, m–p). On average, clouds with multiple cores have higher and more variable values for core area, lifetime, and height. In contrast, the area ratio is lower and has a weaker variability for multi-core systems. For single-core clouds, we observe an afternoon peak over land and nocturnal and afternoon peaks over the ocean. Multi-core clouds show a weak diurnal variation, particularly over the ocean (Figure 11, i–l)."

- **L315, L371**: How do you define convective activity? Is this shown in Figure 11?
  **Author's response**: The term "convective activity" here refers to the frequency we detected potentially convective clouds, in particular in regards to those with multiple cores. We change the text to be more clear in the revised manuscript and avoid unclear phrases.
  **Author's changes in the manuscript**: Throughout the manuscript, we replaced the phrase "convective activity" by a more direct description of the results. Examples may be found in Sect. 4.2.2, e.g, lines 339-344: "For different months, the value variability may considerably influence the development of convective clouds and their core structures within the tropics (Andrews et al., 2024). We explore these changes in Figure 12 by comparing monthly averages of the cloud area, CTH, cloud lifetime, number of cores, core area, and area ratio over land and sea for single-core and multi-core clouds. From March to August, the cloud area shows a gradual increase for single- and multi-core cloud systems over the ocean. For clouds over land, the cloud area slightly decreases (Figure 12, a)."

- **L315-316**: Are you explaining that this is occurring over time, and witnessed over both land and ocean and for all cores?
  **Author's response**: Refers to an increase of the cloud area, core area, and number of cores (until July) which may be seen over (partly) land and sea. However, we revise the Section to be more clear.
  **Author's changes in the manuscript**: See comment above, we revise the description of (now) Figure 12 in Sect 4.2.2, lines 339-344: "From March to August, cloud area shows a gradual increase, especially for oceanic clouds — both single- and multi-core systems (Figure 12, a). In contrast, CTH generally decreases, though with noticeable month-to-month fluctuations rather than a consistent downward trend (Figure 12, b). Cloud lifetime remains relatively stable for single-core clouds, while multi-core clouds exhibit a slight decline in lifetime over the ocean. Over land, lifetime rises from March to April, dips in May, and increases again in June—returning to near-initial values by August. The number of cores per cloud increases over the ocean from March to July, followed by a sharp drop in August (Figure 12, c). Initially higher over land, core counts shift in favor of oceanic clouds after April. Land-based systems show a decrease from March to June, a peak in July, and another decline in August (Figure 12, d). Core area steadily increases over the ocean but fluctuates more over land (Figure 12, e). The area ratio between core and cloud shows a slight decrease for multi-core clouds throughout the period, while remaining higher and more variable for single-core clouds (Figure 12, f)."

- **L316-317**: Do you mean to imply that the number of DCCs increases as the months progress? There is a sharp drop though between July to August over sea
  **Author's response**: You are right, we revise the text to more accurate regarding the non-linear monthly changes between March and August.
  **Author's changes in the manuscript**: See comment above, description in Sect. 4.2.2., lines 339-344.

- **L318-319**: Perhaps use wording other than "less distinct" to make your point clearer. It seems like sea has more cores than land starting in May.
  **Author's response**: Thank you for your comment, we revise the text in the revised manuscript to be more clear.
  **Author's changes in the manuscript**: See comment above, description in Sect. 4.2.2., lines 339-344.

- **L322-323**: Where do you get the anvil extent being larger over land than ocean?
  **Author's response**: We agree the description got mixed up, we revise the text to be more accurate.
  **Author's changes in the manuscript**: Removed figure from the revised manuscript. Instead, we added a statistical evaluation of differences between land and sea in Sect. 4.2.2, lines 355-364: "To quantify the effect of these changes, we compare average values across two periods: March–May (MAM) and June-August (JJA). Metrics include the cloud area, CTH, cloud lifetime, number of cores, core area, and area ratio (Table 4). We calculate Cohen's D to measure effect sizes, with thresholds defined as small ($< 0.2$), medium ($0.2$–$0.5$), and large ($> 0.8$) (Cohen, 2013). Over the ocean, cloud area, number of cores, and core area are higher in JJA, while CTH, cloud lifetime, and area ratio are greater in MAM. A similar pattern emerges over land, except cloud area and number of cores are higher in MAM. Overall, observed differences between the two seasons and over land and sea remain weak. Most effect sizes are small, indicating high internal variability rather than distinct temporal trends within the period. These results highlight the importance of analysing longer time periods to account for the inherent variability and imbalance between cloud tracks over land and sea (Figures 4 and 9), which may influence the representativeness of the findings."

- **L332-333**: Again, has this definition of cooling been used previously in literature? Please explain the assumption.
  **Author's response**: See comment above, we change the term "cooling" to "reflectivity gradient" (as we do not consider cloud temperature directly). The method for estimating the cloud life-cycle is described in Sect. 3.6 in the revised manuscript. Here, we explain how we we adapt our approach from a reflectivity-based assessment of the development stage using a threshold of 0 dBZ at 10 km height as a proxy to identify a potential cooling at the cloud top for convective cells derived from CloudSat.
  **Author's changes in the manuscript**: See Sect. 3.6, lines 247-252: "Development stage: Building on the approach by Luo et al. (2008), we use a radar reflectivity threshold of 0 dBZ at 10 km altitude as a proxy for potential cloud-top cooling, which may be indicative of convective growth. We calculate the temporal gradient of radar reflectivity at 10 km for each cloud trajectory, identifying the time of maximum increase to mark the cloud development stage. This stage may be associated with a high cloud vertical layer and strong updrafts that support continued vertical growth (e.g., Kikuchi and Suzuki (2019); Chen et al. (2021)).The transition from development to maturity is defined by the time of maximum radar reflectivity increase (Takahashi et al., 2023; Hu et al., 2021)".

- **L352**: Did you quantify "the cooling and area growth appear earlier during the relative cloud lifetime" to determine it? It is not clear based on the figure.
  **Author's response**: We agree this description is unclear and revise the text.
  **Author's changes in the manuscript**: See Sect. 4.3.1, lines 390-409 for a description of Figure 15 (time dependency for the life-cycle statistics based on average cloud lifetime after detection) and lines 410-415 for a summary: "Notably, the analysis shows that vertical growth may peak after the reflectivity gradient but before area growth. The times stretch for multi-core clouds, while single-core systems exhibit more compact timelines. However, outliers may distort observed mean values. Hence, the consecutive order of the life-cycle statistics may be affected by a high variability in the distribution. Across all cloud tracks, core number and core area tend to peak between vertical and area growth maxima. However, the distributions show a high variability, especially for multi-core clouds and clouds over land. While we observe life-cycle statistics occurring on average later for clouds over land, the differences induced by the surface type remain overall low."

- **L357-358**: For cooling, there is a morning peak at 0100 over both land and ocean. Is it statistically larger than the mean for the next several hours? There are dips at 6 pm over both land and ocean, and an additional dip at 12 pm over land. Again, are these statistically different and indicative of the diurnal cycle? The diurnal cycle of vertical growth seems to be much more pronounced over land, but not noticeable over ocean. Please make this distinction
  **Author's response**: While we see a more distinct peak in the night, the diurnal variations remain weak; however, we did not apply statistical tests for the diurnal variations as show high fluctuations and overall differences appear to be small. We revise the description of the figures to be more clear.
  **Author's changes in the manuscript**: Sect. 4.3.2, lines 418-426: "Figure 16 illustrates the diurnal patterns of the reflectivity gradient, the area growth, and the vertical growth for single-core and multi-core clouds, grouped by surface type. Similar to results from Sect. 4.2.1, the diurnal cycle is more pronounced over land than over the ocean. The reflectivity gradient exhibits short-term fluctuations with noticeable nocturnal peaks around 20:00–21:00 UTC and 00:00–01:00 UTC, followed by decreases. During the day, peaks occur between 09:00–12:00 UTC and around 16:00 UTC, with a negative dip around noon (land) and 18:00 UTC (both land and sea). Over the ocean, the reflectivity gradient is generally higher and shows a slightly weaker variability than over land. Over land, multi-core clouds exhibit stronger gradients than single-core clouds. Distinct land-based negative peaks occur around 03:00–06:00, 08:00, and 11:00–15:00 UTC. Over the ocean, we find a weaker nocturnal peak at 01:00 UTC and a gradual increase between 06:00–20:00 UTC. Overall diurnal variability ranges from 0.5 dBZ (ocean) to 1 dBZ (land), or roughly 8–16 % of the mean gradient range (10–16 dBZ) (Figure 16, a, d)."

- **L359-360**: Which convective characteristics are you referring to?
  **Author's response**: The sentence here refers to the area growth.
  **Author's changes in the manuscript**: See Sect. 4.3.2, lines 424-431: "Multi-core clouds show significantly greater area growth (42–48 %) than single-core clouds (20–27 %). Over the ocean, we see sporadic daytime peaks that occur around 15:00, 20:00, and 01:00 UTC. Over land, area growth increases steadily for single-core clouds in the morning and peaks between 12:00–14:00 UTC. For multi-core clouds, we find several sporadic

peaks during the day, similar to clouds over the ocean. Evening peaks appear around 18:00 and 22:00 UTC for multi-core clouds, and 20:00 UTC for single-core clouds. Diurnal variability remains low, ranging up to 5 % over both land and sea (Figure 16, b, e)."

- **L361-363**: Is this suggesting that as the number of DCCs increases, the total cooling and vertical growth per DCC is less than that of the initiation phase?
  **Author's response**: That is what we would assume from the comparison of life-cycle statistics for single-core and multi-core clouds; however we did not examine statistical differences. In response, we change the description to be more clear.
  **Author's changes in the manuscript**: Please see Sect. 4.2.2, lines 447-459: "Throughout the study period, single-core clouds consistently show higher reflectivity gradients and vertical growth, while multi-core clouds exhibit greater area growth. Though the surface type may influence these statistics, the observed effect in our study remains relatively small. In contrast, the number of cores plays a more substantial role in shaping the cloud life-cycle." and Sect. 5.1, lines 470-476: "We also find that changes in cloud area, core number, and cloud height often evolve in line with an idealised convective life cycle described in Sect. 3.6. Longer-lived clouds tend to exhibit more cores and larger maximum core areas. Multi-core systems reach their peak core number and core size later in their life cycle than single-core clouds (Section 4.3.1). The reflectivity gradients correlate positively with vertical growth and negatively with area expansion, reflecting transitions from development to maturity, as noted by Hu et al. (2021). Single-core clouds display stronger vertical ascent and higher reflectivity gradients, though most correlations are weak — aside from a strong negative relationship between cloud lifetime and vertical growth, and a moderate positive link between lifetime and area growth."

- **L365**: How about the peaks in the early morning?
  **Author's response**: We agree there are multiple peaks in the morning which may be seen for all clouds. We revise the description of the figure to be more detailed.
  **Author's changes in the manuscript**: See Sect. 4.2.2, lines 424-431: "Multi-core clouds show significantly greater area growth (42–48 %) than single-core clouds (20–27 %). Over the ocean, we see sporadic daytime peaks that occur around 15:00, 20:00, and 01:00 UTC. Over land, area growth increases steadily for single-core clouds in the morning and peaks between 12:00–14:00 UTC. For multi-core clouds, we find several sporadic peaks during the day, similar to clouds over the ocean. Evening peaks appear around 18:00 and 22:00 UTC for multi-core clouds, and 20:00 UTC for single-core clouds. Diurnal variability remains low, ranging up to 5 % over both land and sea (Figure 16, b, e)."

- **L367**: Aside from the dip in July over land.
  **Author's response**: We agree and revise the description.
  **Author's changes in the manuscript**: See Sect. 4.2.2, lines 440-445: "Between March and June, area growth is higher over land, peaking in May for single-core and in June for multi-core clouds. After June, area growth becomes higher over the ocean. Over the period, single-core clouds over land show a net decline (around 3 %), while values increase slightly over the ocean (around 5 %) and for multi-core clouds over land (around 3 %). Monthly changes appear to be nonlinear and fluctuate considerably (Figure 17, b, e)."

- **L368-369**: Why do you think the cooling and vertical growth are higher for clouds with a single core?
  **Author's response**: We assume the higher gradient of the radar reflectivity and the higher vertical growth may be connected to the difference in the average radar reflectivity for single-core and multi-core clouds. For multi-core clouds the effect of individual cores may weaker as outliers may be more blurred compared to single-core clouds.
  **Author's changes in the manuscript**: See Sect. 5.2, lines 497-501: "Throughout the day, we observe several peaks for the reflectivity gradient, followed by phases of vertical and horizontal cloud growth (Section 4.3.2). Notably, as the number of cores increases, both reflectivity gradient and vertical growth decline, while area growth becomes more pronounced (Section 4.3.1). This finding may correspond to a higher reflectivity at 10 km and broader spatial extent seen for multi-core systems (Section 4.1). Our observations may point to a self-sustaining mechanism where cores are regenerated in response to diurnal heating (e.g., Deng et al. (2016); Hartmann et al. (2018); Takahashi et al. (2017)). However, we did not explicitly investigate this process."

- **L371-372**: Are you referring to the larger variability in area and vertical growth of convection over land? In other words, the seasonal cycle of convection is more pronounced over the ocean, but the diurnal cycle of

convection is more pronounced over land?

**Author's response**: We checked our data, the updated figures suggest that the variability over land is higher for the seasonal and diurnal cycle. We revise the text to describe the observed results more directly.

**Author's changes in the manuscript**: Sect. 4.2.2, lines 447-451: "Throughout the study period, single-core clouds consistently show higher reflectivity gradients and vertical growth, while multi-core clouds exhibit greater area growth. Though the surface type may influence these statistics, the observed effect in our study remains relatively small. In contrast, the number of cores plays a more substantial role in shaping the cloud life-cycle."

- **Section 5.1**: This section is great, I appreciate the connection between your results and previous literature. Could you also add a few sentences on discerning the characteristics between isolated and multi-cellular convection within previous literature, and how it is consistent with what you are finding?

  **Author's response**: Thank you for your comment. We add a further comparison of studies dealing with single- and multi-core clouds in the tropics in the revised manuscript.

  **Author's changes in the manuscript**: Please see revised Sect. 5.2, lines 483-489: "Compared to previous studies by Takahashi et al. (2023) and Hu et al. (2021), our analysis identifies a significantly higher number of potentially convective cloud tracks. The derived cloud characteristics align well with aircraft observations (Zipser and LeMone, 1980) and precipitation-based studies (Zipser et al., 2006). Over tropical Africa, our core distribution results are consistent with those derived for geostationary satellite data (Jones et al., 2024) or the CloudSat CPR Deng et al. (2016), both of which found a high prevalence of clouds with one to three cores. Similarly, Pilewskie and L'Ecuyer (2022) reported that one-third to half of convective systems observed globally by the CloudSat CPR contain a single core. For the tropics, however, our results are in closer agreement. In line with these findings, we observe that cloud area generally increases with the number of cores. However, this relationship exhibits substantial variability, especially in multi-core systems (Section 4.3.1)."

- **L405-407**: Had you considered also using CALIPSO observations to capture thin ice clouds?

  **Author's response**: We are aware that integrating CALIPSO into the machine learning approach may help to identify ice clouds. So far, in the interest of time and resources, we did not get to try the idea of using either CALIPSO or the combined CloudSat-CALIPSO products to predict a, possibly, more accurate 3D cloud field. However, we think it might be interesting for future studies.

  **Author's changes in the manuscript**: In Sect. 5.3, we describe some limitations of the study and point out how the integration of more recent sensors could help to enhance the results, e.g., in lines 523-529: "Emerging satellite systems, such as the Flexible Combined Imager on the Meteosat Third Generation (MTG) platform (Holmlund et al., 2021) and the enhanced CPR on the EarthCARE mission (Eisinger et al., 2024), offer improved spatial and temporal resolution. These instruments are expected to enhance the detection and characterisation of the convective cloud life-cycle. Moroever, our study does not account for several potentially important influences on convection, such as aerosol interactions, vertical wind shear, and entrainment rates (Masunaga and Luo, 2016). Incorporating these factors in future analyses could lead to a more comprehensive understanding of convective processes."

- **L429-431**: Convection as a self-maintenance mechanism has been suggested in previous literature, but which signatures in your results are pointing to this conclusion?

  **Author's response**: We did not specifically address this, but results found in Sect. 4.3 (i.e., changes in the cloud life-cycle for single-core vs multi-core clouds) may point towards such a mechanism for multi-core clouds; we revise the manuscript to be more clear.

  **Author's changes in the manuscript**: See Sect. 5.2, lines 496-501: "Throughout the day, we observe several peaks for the reflectivity gradient, followed by phases of vertical and horizontal cloud growth (Section 4.3.2). Notably, as the number of cores increases, both reflectivity gradient and vertical growth decline, while area growth becomes more pronounced (Section 4.3.1). This finding may correspond to a higher reflectivity at 10 km and broader spatial extent seen for multi-core systems (Section 4.1). Our observations may point to a self-sustaining mechanism where cores are regenerated in response to diurnal heating (e.g., Deng et al. (2016); Hartmann et al. (2018); Takahashi et al. (2017)). However, we did not explicitly investigate this process."

- **L432-433**: What do you mean by "weaker convective activity"?

  **Author's response**: The sentence refers to the occurrence of less convective cloud tracks and clouds with fewer cores. We will revise the text to improve the written expression and provide a more clear description of the

findings.

**Author's changes in the manuscript**: We removed the phrase "convective activity" here (see Introductory Remarks). Please see the concluding remarks in Sect. 6, lines 549-558: "The results suggest that differences based on the number of cores are higher than the surface-type induced variability. Single- core clouds develop and dissipate on shorter timescales. They have a smaller cloud and core area, and lower CTH and core height than multi-core systems. The longer cloud lifetime of multi-core clouds may be associated to a later occurrence of the maximum number of cores and core area. Between single-core and multi-core clouds, we find considerable differences in the cloud life-cycle statistics regarding the changes in the radar reflectivity at 10 km height, the vertical growth, and the area growth of the cloud. While the former two are higher for clouds with a single core, multi-core cloud clusters with a larger cloud area tend to grow more along the horizontal dimension. The more cores we find, the later the maximum number of cores and the maximum core area occur. While the differences between the convective clouds over land and ocean are lower than expected, we emphasise our analysis uses six months of data and may not represent the annual cycle of convection. Nevertheless, expanding the approach to investigate a longer time series may account for current uncertainties."

**4 Technical corrections**

- **L12-13**: The way that this sentence reads is a bit confusing. I'd suggest "... more intense over land than ocean during both seasons, despite an increase in convective activity over the ocean during summer".
  **Author's response**: Thank you for your suggestion, we will change the text.
  **Author's changes in the manuscript**: Revised the abstract to describe the results more directly and focus less on "seasonal" differences (which are hardly statistical robust).

- **L58**—"Early studies": Change to "Some studies" as a study provided in the next sentence was published earlier than Masunaga and Luo, (2016).
  **Author's response**: Thank you for your suggestion, we change the sentences.
  **Author's changes in the manuscript**: Revised Sect. 1, lines 46-54: "Early studies relied on manual tracking, but automated detection algorithms now enable the processing of large datasets. Most of these algorithms are centroid-based, linking cloud objects across time steps (Prein et al., 2024). One of the earliest and most influential tools is TITAN Dixon and Wiener (1993), later adapted into TINT Raut et al. (2021), which is optimized for tracking fast-evolving storm cells. TOOCAN, developed by Fiolleau and Roca (2013), focuses specifically on convective cores and associated anvils in MCSs. More recently, general-purpose tools such as PyFLEXTRKR (Feng et al., 2023) and tobac (Heikenfeld et al., 2019) have emerged. PyFLEXTRKR offers flexible 2D tracking, while tobac supports 4D analysis, enabling a more comprehensive view of convective systems."

- **L70**: —"Contrasting": Change to "Contrastingly" or "Alternatively"
  **Author's response**: We change the phrase.
  **Author's changes in the manuscript**: See comment above, we revised the text in Sect. 1, lines 46-54.

- **L93** —"diverging convective development": "Diverging" is not the best descriptor here, so rephrase to say that convection develops differently in the extra-tropics compared to tropics.
  **Author's response**: We agree and change the expression.
  **Author's changes in the manuscript**: We removed the sentence in the revised manuscript.

- **L95-96**: sentence beginning with "Clouds..." seems out of place in this paragraph.
  **Author's response**: We will revise the section.
  **Author's changes in the manuscript**: We removed the sentence from Sect. 2, lines 72-75: "The area of interest (AOI) for this study spans a tropical region over central and western Africa, extending from 30° N to 30° S and 30° W to 30° E. This region is characterised by environmental conditions that contribute to the development of convective clouds (Takahashi et al., 2023). Our objective is to detect and analyse convective clouds and their life-cycles by a six month period between March and August 2019."

- **L161**: Does "width" refer to the vertical cloud thickness? Please clarify that here and in the workflow figure (Figure 1).

**Author's response**: Refers to the number of pixels that comprise the horizontal cloud area, we will change the figure and its description.

**Author's changes in the manuscript**: See Figure 1 (b) and Sect. 3.2.2, lines 170-175: "he shape of each cloud is characterised using the best-fitting ellipse (Ganetis et al., 2018). We then calculate the aspect ratio — i.e., the ratio of the major to minor axis lengths. If the major axis is more than 75 % longer than the minor axis, we classify the cloud as elongated (Cui et al., 2021). The orientation of the major axis provides the direction of elongation, which guides the search for potential split locations. Next, we examine the aggregated 2D cloud area along this direction and analyse the area distribution to detect change points."

- **L204**: ", e.g."

  **Author's response**: Will be changed.

  **Author's changes in the manuscript**: We removed the sentence from the manuscript as it seemed to be repetitive to the introduction in Sect. 1, lines 54-66.

- **L206-209**: Although important, this information is already addressed in the introduction, so can be removed to make the paper more succinct

  **Author's response**: Thanks for the suggestion, we remove the sentences.

  **Author's changes in the manuscript**: See above, we removed the sentences as they may be found in the introduction, lines 54-66.

- **L240**: Morocco

  **Author's response**: Will be corrected.

  **Author's changes in the manuscript**: Sect 4.1, line 269, "While regions over Morocco and Mauritania showed radar reflectivity > 0 dBZ, no vertically continuous convective systems were identified there."

- **L271**: Replace "connected" with "associated with"

  **Author's response**: Will be changed.

  **Author's changes in the manuscript**: Sect. 4.1, lines 293-294: "The analysis of core properties (Figure 8) shows average core lifetimes of 0.3–0.4 hours for single-core clouds, increasing to about 0.8 hours for clouds with more than 10 cores (Figure 8, a). "

- **L273**: Replace "come along" with "associated with" or an alternate phrase.

  **Author's response**: Will be changed.

  **Author's changes in the manuscript**: Sect. 4.1, line 297-299: "Core height and distance between cores both increase with core count (Figure 8, d–e). The largest distances, which may indicate the least compact core morphology, occur for clouds with 10 and more cores (Figure 8, e)."

- **L273**: By more extensive, you mean larger horizontal area, not more extensive in the vertical direction? Please clarify.

  **Author's response**: Yes, the sentence refers to a larger horizontal area. We change the text to be more clear.

  **Author's changes in the manuscript**: Sect. 4.1, lines 295-297: "Core area is slightly larger for single-core clouds than for those with 2–9 cores but increases considerably for clouds with 10 and more cores. For single-core clouds, we detect a larger core area over the ocean, while cores for multi-core clouds are larger over land (Figure 8, c)."

- **L274**: Please modify the phrase "increases stronger".

  **Author's response**: We will change the text.

  **Author's changes in the manuscript**: See comment above, lines 295-297: "Core area is slightly larger for single-core clouds than for those with 2–9 cores but increases considerably for clouds with 10 and more cores. For single-core clouds, we detect a larger core area over the ocean, while cores for multi-core clouds are larger over land (Figure 8, c)."

- **L300**: Does "temporal shift of the afternoon peak" mean that the peak occurs later in the day?

  **Author's response**: Yes, that's we wanted to state. We change the text to be more clear.

  **Author's changes in the manuscript**: We revised the description of Figure (now) 10 in Sect. 4.2.1, lines 314-324: "We analyse the diurnal cycle of cloud properties for single-core and multi-core clouds over land and ocean by computing a 2D density distribution displaying hourly changes of the cloud properties. Figure 10 illustrates

these variations in cloud lifetime (a–d), cloud area (e–h), and radar reflectivity at 10 km height (i–l). Over land, single-core clouds show an afternoon peak (12:00–16:00 UTC) in both radar reflectivity and cloud area, while cloud lifetime displays two peaks: one at night and one in the morning (Figure 10, b, d). Over the ocean, the diurnal cycle is weaker or less distinct. Cloud lifetime lacks a clear diurnal peak (Figure 10, a, c), whereas cloud area and reflectivity show nocturnal and daytime peaks (Figure 10, e, i, k). Despite similar diurnal patterns for the cloud lifetime and radar reflectivity, multi-core clouds consistently exhibit higher mean values than single-core clouds. These differences may reflect environmental contrasts between land and ocean. As suggested by Cui et al. (2021), local circulations over land in the tropics often trigger afternoon convection, producing the observed peaks in Figure 10 (f), (h), and (j). In contrast, more constant ocean temperatures may suppress strong diurnal variations (Figure 10, a, c, g)."

- **Figure 9**: Add units to the "Cloud lifetime" axis.
  **Author's response**: We will add units, will be changed to "Cloud lifetime [h]".
  **Author's changes in the manuscript**: See (now) Figure 10, a-d: "Cloud lifetime [h])" in Sect. 4.2.1

- **Figure 10**: Label axis "core eccentricity" for consistency.
  **Author's response**: We will change the label.
  **Author's changes in the manuscript**: Sect. 4.2.1, (now) Figure 11, replaced core eccentricity with core height in this figure

- **L371**: Can delete sentence beginning with "Clustered clouds..." because this was implied from the previous sentence.
  **Author's response**: We agree and remove the sentence in the revised manuscript.
  **Author's changes in the manuscript**: See Sect. 4.3.2, lines 447-450: "Throughout the study period, single-core clouds consistently show higher reflectivity gradients and vertical growth, while multi-core clouds exhibit greater area growth. Though the surface type may influence these statistics, the observed effect in our study remains relatively small. In contrast, the number of cores plays a more substantial role in shaping the cloud life-cycle."

- **Figure 15**: Make the y-axes ranges consistent between single and multiple DCCs so it is easier to compare them.
  **Author's response**: We change the axes to cover the same range.
  **Author's changes in the manuscript**: Sect. 4.3.2, (now) Figure 16 and 17 have consistent y-axes for the three life-cycle statistics (10-16 dBZ, 20-50 %, 5-10 km)

- **L378**: In addition to specific horizontal or vertical dimension, you can also include that there is a limitation in the temporal dimension if trying to exclusively use CloudSat data.
  **Author's response**: Thank you for your suggestion, we add some details in the text.
  **Author's changes in the manuscript**: See Sect. 1, lines 56-58: "Active and passive sensors contain important vertical or horizontal information, but are limited in their spatial and temporal coverage (active) or offer only an approximation of the vertical column (passive) (Masunaga and Luo, 2016; Taylor et al., 2017)."

---

## Referee Report (RR1)

The authors have done a thorough job in addressing the comments. They have clarified the methodology, as well as improved the description of the results. I suggest that this paper can be accepted as is, with a few suggested edits that can be completed during the galley proof stage:

- L38: Make note that "cold peaks in brightness temperature" is specific to passive satellite observations
- L162: "voxel" is a typo
- L228: remove extra comma
- L489: put "Deng et al. (2016)" reference in parentheses
- L513-514: Please refine the sentence—are you comparing the results to earlier months?

---

## Author Response (AR2)

**Final response**
**(Manuscript "EGUSPHERE-2025-374")**

Sarah Brüning and Holger Tost

July 8, 2025

**1 Introduction**

We would like to thank the reviewers again for their feedback. In this response, we address the suggested edits from the report of Referee 2. For every comment, you can find (1) the comment, (2) the author's response (both with lines from submitted manuscript), and (3) the author's changes in the revised manuscript. The lines given for **author's changes in the manuscript** refer to the lines in the **revised manuscript**.

**2 Specific comments**

:

- **L38**: L38: Make note that "cold peaks in brightness temperature" is specific to passive satellite observations.
  **Author's response**: We revise the sentence to add more detail.
  **Author's changes in the manuscript**: Lines 38f.: "When derived from passive satellite observations, cores are typically characterised by cold peaks in brightness temperature, surrounded by warmer anvil regions."

- **L162**: L162: "voxel" is a typo.
  **Author's response**: Will be changed.
  **Author's changes in the manuscript**: Lines 162f.: "The result is a labeled 3D cloud mask, where each pixel is either zero (indicating no cloud) or an integer label corresponding to a specific cloud object (Fiolleau and Roca, 2013)."

- **L228**: L228: remove extra comma.
  **Author's response**: Will be removed.
  **Author's changes in the manuscript**: Lines 228-230: "The core area and height are derived from the column-wise maximum horizontal extent and vertical extent of the previously identified cores, similar to the cloud area and CTH. These metrics may help characterise the structural properties of detected cloud systems."

- **L489**: L489: put "Deng et al. (2016)" reference in parentheses.
  **Author's response**: Will be changed.
  **Author's changes in the manuscript**: Lines 488-490: "Over tropical Africa, our core distribution results are consistent with those derived for geostationary satellite data (Jones et al., 2024) or the CloudSat CPR (Deng et al., 2016), both of which found a high prevalence of clouds with one to three cores."

- **L513-514**: L513-514: Please refine the sentence—are you comparing the results to earlier months.
  **Author's response**: Yes, here we want to compare the distribution of cloud tracks over land and sea along the period (i.e., July nd August to earlier months of the period). We revise the sentence to be more clear.
  **Author's changes in the manuscript**: Lines 513-516: "Between June and August, we detect overall a lower proportion of cloud tracks than between March and May. However, we see a shift regarding the distribution of cloud tracks over land and sea along the period: While we detect a higher proportion of cloud tracks over continental Africa in March and April, the number of detected clouds over the ocean exceeds those over land from May to August."